# An Equivalence Between Static and Dynamic Regret Minimization

**Andrew Jacobsen**[*]
Università degli Studi di Milano
Politecnico di Milano
contact@andrew-jacobsen.com

**Francesco Orabona**
KAUST
francesco@orabona.com

## Abstract

We study the problem of dynamic regret minimization in online convex optimization, in which the objective is to minimize the difference between the cumulative loss of an algorithm and that of an arbitrary sequence of comparators. While the literature on this topic is very rich, a unifying framework for the analysis and design of these algorithms is still missing. In this paper we show that *for linear losses, dynamic regret minimization is equivalent to static regret minimization in an extended decision space*. Using this simple observation, we show that there is a frontier of lower bounds trading off penalties due to the variance of the losses and penalties due to variability of the comparator sequence, and provide a framework for achieving any of the guarantees along this frontier. As a result, we also prove for the first time that adapting to the squared path-length of an arbitrary sequence of comparators to achieve regret $R_T(\boldsymbol{u}_1, \ldots, \boldsymbol{u}_T) \leq \mathcal{O}(\sqrt{T \sum_t \|\boldsymbol{u}_t - \boldsymbol{u}_{t+1}\|^2})$ is impossible. However, using our framework we introduce an alternative notion of variability based on a locally-smoothed comparator sequence $\bar{\boldsymbol{u}}_1, \ldots, \bar{\boldsymbol{u}}_T$, and provide an algorithm guaranteeing dynamic regret of the form $R_T(\boldsymbol{u}_1, \ldots, \boldsymbol{u}_T) \leq \widetilde{\mathcal{O}}(\sqrt{T \sum_i \|\bar{\boldsymbol{u}}_i - \bar{\boldsymbol{u}}_{i+1}\|^2})$, while still matching in the worst case the usual path-length dependencies up to polylogarithmic terms.

## 1 Introduction

This paper introduces new techniques for *Online Convex Optimization* (OCO), a framework for designing and analyzing algorithms which learn on-the-fly from a stream of data [14, 51, 5, 31, 6]. Formally, consider $T$ rounds of interaction between the learner and their environment. In each round, the learner chooses $\boldsymbol{w}_t \in \mathcal{W}$ from a convex feasible set $\mathcal{W} \subseteq \mathbb{R}^d$, the environment reveals a $G$-Lipschitz convex loss function $\ell_t : \mathcal{W} \to \mathbb{R}$, and the learner incurs a loss of $\ell_t(\boldsymbol{w}_t)$. The classic objective in this setting is to minimize the learner's *regret* relative to any fixed benchmark $\boldsymbol{u} \in \mathcal{W}$:

$$R_T(\boldsymbol{u}) := \sum_{t=1}^{T} (\ell_t(\boldsymbol{w}_t) - \ell_t(\boldsymbol{u})) \,.$$

In this paper, we study the more general problem of minimizing the learner's regret relative to any *sequence* of benchmarks $\boldsymbol{u}_1, \ldots, \boldsymbol{u}_T \in \mathcal{W}$ [17, 18]:

$$R_T(\boldsymbol{u}_1, \ldots, \boldsymbol{u}_T) := \sum_{t=1}^{T} (\ell_t(\boldsymbol{w}_t) - \ell_t(\boldsymbol{u}_t)) \,.$$

---

[*]Work done while visiting Optimal Lab at KAUST.

38th Conference on Neural Information Processing Systems (NeurIPS 2024).

This objective is typically referred to as *dynamic* regret, to distinguish it from the special case where the comparator sequence is fixed $\boldsymbol{u}_1 = \cdots = \boldsymbol{u}_T$ (referred to as *static* regret). We focus in particular on the special case of *Online Linear Optimization* (OLO), in which $\ell_t(\boldsymbol{w}) = \langle \boldsymbol{g}_t, \boldsymbol{w} \rangle$ for some $\boldsymbol{g}_t \in \mathbb{R}^d$. Note that OCO problems can always be reduced to OLO via the well-known inequality $\ell_t(\boldsymbol{w}_t) - \ell_t(\boldsymbol{u}) \leq \langle \boldsymbol{g}_t, \boldsymbol{w}_t - \boldsymbol{u} \rangle$ for $\boldsymbol{g}_t \in \partial \ell_t(\boldsymbol{w}_t)$, where $\partial \ell_t(\boldsymbol{w}_t)$ is the subdifferential set of $\ell_t$ at $\boldsymbol{w}_t$ [see, *e.g.*, 36], so throughout this paper we will focus on the OLO setting.

Intuitively, if the sequence of comparators $\boldsymbol{u}_1, \ldots, \boldsymbol{u}_T$ varies too much, it should be impossible to achieve low dynamic regret. On the other hand, we know it is possible to achieve sublinear regret if the sequence of comparators is constant, *i.e.*, $\boldsymbol{u}_1 = \cdots = \boldsymbol{u}_T$, because this is simply the static case. Hence, we need a way to quantify the complexity, or *variability*, of the comparator sequence. The most commonly used notion of complexity in this regard is the *path-length* of the comparator sequence [17, 18], defined as

$$P_T^{\|\cdot\|} := \sum_{t=2}^{T} \|\boldsymbol{u}_t - \boldsymbol{u}_{t-1}\| .$$

It is possible to show that Online Gradient Descent has a dynamic regret of $\mathcal{O}((D + P_T^{\|\cdot\|})G\sqrt{T})$ in *bounded* domains, where $D$ is an upper bound on the diameter of the feasible set and $G$ is the Lipschitz constant of the losses [51]. This bound was improved to $\mathcal{O}(\sqrt{DP_T^{\|\cdot\|}}G\sqrt{T})$ and shown to be minimax optimal by Zhang et al. [46].

Notice that the path-length bounds scale with a rather pessimistic constant of $D = \sup_{w,w' \in \mathcal{W}} \|\boldsymbol{w} - \boldsymbol{w}'\|$. A better bound would instead scale with the *squared* path-length:

$$P_T^{\|\cdot\|^2} := \sum_{t=1}^{T-1} \|\boldsymbol{u}_t - \boldsymbol{u}_{t-1}\|^2,$$

which can be significantly smaller[2] than the penalty in the bound above: $P_T^{\|\cdot\|^2} \leq DP_T^{\|\cdot\|}$. However, guarantees scaling with $P_T^{\|\cdot\|^2}$ are not well understood in general compared with the more common $P_T^{\|\cdot\|}$ bounds, and have only been obtained by restricting the comparator sequence to $\boldsymbol{u}_t = \operatorname{argmin}_{\boldsymbol{w} \in \mathcal{W}} \ell_t(\boldsymbol{w})$ or under additional assumptions such as strong-convexity [44, 45, 7].

In this paper, we focus on the challenging case that the domain is *unbounded*, where recent works have achieved the dynamic regret $\widetilde{\mathcal{O}}\big(\sqrt{\max_{t,t'} \|\boldsymbol{u}_t - \boldsymbol{u}_{t'}\| P_T^{\|\cdot\|} T}\big)$ in the worst case [20, 25, 21, 47]. Of particular interest, Jacobsen and Cutkosky [20], Zhang et al. [47] achieve bounds of the form

$$R_T(\boldsymbol{u}_1, \ldots, \boldsymbol{u}_T) \leq \widetilde{\mathcal{O}}\left( \sqrt{P_T^{\|\cdot\|} \sum_{t=1}^{T} \|\boldsymbol{g}_t\|^2 \|\boldsymbol{u}_t - \bar{\boldsymbol{u}}\|} \right), \tag{1}$$

which avoids the pessimistic multiplicative penalty of $\max_{t,t'} \|\boldsymbol{u}_t - \boldsymbol{u}_{t'}\|$, but results in a coupling between the gradient and variability penalties. It is unclear if it is possible to obtain a guarantee which cleanly separates the variability and variance penalties, to achieve dynamic regret scaling as $R_T(\boldsymbol{u}_1, \ldots, \boldsymbol{u}_T) \leq \mathcal{O}\left( \sqrt{P_T^{\|\cdot\|^2} \sum_{t=1}^{T} \|\boldsymbol{g}_t\|^2} \right)$. In fact, it is not clear in general how to reason about potential trade-offs that may result from adapting to different measures of variability.

**Contributions.** In this paper, we show how to reformulate the dynamic regret miniminization problem as an *equivalent* static regret problem (Section 2). This equivalence allows us to use results for the static regret setting to prove both upper and lower bounds for dynamic regret.

In our first application of this equivalence, we show that the ideal guarantee scaling the with squared path-length $\mathcal{O}\big(\sqrt{P_T^{\|\cdot\|^2} \sum_{t=1}^{T} \|\boldsymbol{g}_t\|^2}\big)$ is **not** possible in general (Section 3). We do this by proving

---

[2]Note that the bound of Zhang et al. [46] trivially implies a squared path-length dependence since $P_T^{\|\cdot\|} \leq \sqrt{TP_T^{\|\cdot\|^2}}$, so one can obtain a bound of $\mathcal{O}(\sqrt{DP_T^{\|\cdot\|}}G\sqrt{T}) \leq \mathcal{O}(D^{1/2}(P_T^{\|\cdot\|^2})^{1/4}GT^{3/4})$. However, this bound is not interesting because it does not remove the dependence on $D$ and it is never better than the existing $\sqrt{DP_T^{\|\cdot\|}}G\sqrt{T}$ bound.

a novel lower bound showing that there is a fundamental trade-off between the penalties incurred due to comparator variability and penalties incurred due to loss variance, leading to a new frontier of dynamic regret lower bounds.

Our second application is to provide a framework for achieving any of the variance/variability trade-offs along the lower bound frontier, up to polylogarithmic terms (Section 4). Our framework allows us to develop dynamic regret algorithms by simply choosing suitable dual-norm pairs $(\|\cdot\|, \|\cdot\|_*)$ in the static regret problem. Along with our matching lower bound, this framework provides a concrete way to reason about different measures of comparator variability and the trade-offs they entail, and to design algorithms achieving those trade-offs.

While our lower bound demonstrates that the ideal squared path-length guarantee cannot be achieved, using our framework we show that it is possible to achieve an alternative guarantee that scales with

$$\bar{P}^{\|\cdot\|^2}(\boldsymbol{u}_1, \ldots, \boldsymbol{u}_T) \approx \sum_i \left\| \bar{\boldsymbol{u}}_i^{(\tau)} - \bar{\boldsymbol{u}}_{i+1}^{(\tau)} \right\|_2^2,$$

where $\bar{\boldsymbol{u}}_i^{(\tau)}$ is a *local average* of the comparator sequence at a timescale of $\tau$ (see Section 4.1). Similar to $P_T^{\|\cdot\|^2}$, this variability measure maintains the property that it matches the worst-case guarantees based on path-length up to polylogarithmic terms, *i.e.*, $\bar{P}_T^{\|\cdot\|^2} \leq \widetilde{\mathcal{O}}(\max_{t,t'} \|\boldsymbol{u}_t - \boldsymbol{u}_{t'}\| P_T^{\|\cdot\|})$. These are the first guarantees for general OCO that fully decouple the variance and variability penalties for dynamic regret without explicitly incurring pessimistic $\max_{t,t'} \|\boldsymbol{u}_t - \boldsymbol{u}_{t'}\|$ penalties.

**Related Work.**  Our approach is inspired by the Haar OLR algorithm of Zhang et al. [47]. In that work, dynamic regret is approached by interpreting the comparator sequence as a high-dimensional "signal" which is decomposed into a frequency domain representation using a dictionary of features. Then, for each feature vector in the dictionary, a 1-dimensional parameter-free [32, 28] algorithm is used to learn how well that feature correlates with the losses. This allows one to compete with an arbitrary comparator sequence, so long as it can be represented in terms of the chosen dictionary of features. We take a similar but slightly more general approach. Our framework also represents the comparator sequence as a high-dimensional signal, but we instead use this signal to define an *equivalent static regret problem*, a perspective that allows us design algorithms for dynamic regret by simply choosing suitable dual-norm pairs.

Other prior works in the general OCO setting have also studied various alternative forms of variability such as the temporal variability $\sum_{t=1}^{T-1} \sup_{\boldsymbol{w} \in \mathcal{W}} |\ell_t(\boldsymbol{w}) - \ell_{t+1}(\boldsymbol{w})|$ [3, 22, 4] or deviation of the comparator from a given dynamical model $\sum_{t=1}^{T-1} \|\boldsymbol{u}_t - \Phi_t(\boldsymbol{u}_{t-1})\|$ [16]. Alternative variance penalties have also been studied in the dynamic setting, such as the small-loss penalties $\sum_{t=1}^{T} \ell_t(\boldsymbol{u}_t)$ or gradient variation penalties $\sum_{t=1}^{T} \sup_{\boldsymbol{w} \in \mathcal{W}} \|\nabla \ell_t(\boldsymbol{w}) - \nabla \ell_{t+1}(\boldsymbol{w})\|$ [15, 49, 21, 50]. It is also possible to achieve a smaller regret with stronger assumptions on the losses [2]. It is important to note however that almost all prior works, with the exception of Jacobsen and Cutkosky [20], Luo et al. [25] and Zhang et al. [47], study dynamic regret only in the easier bounded domain setting.

There is also an often ignored connection between measures of comparator variability and the function classes studied in non-parametric regression. In particular, considering the case that the losses are $\ell_t(x) = (x - u_t)^2$, then the sequence of comparators $u_1, \ldots, u_T$ with bounded path length $C_T$ and bounded squared path length $(C_T')^2$ corresponds to the sequence with discrete total variation bounded by $C_T$ and the discrete Sobolev class with bound $C_T'$, respectively. In this setting, the minimax rates are known [24, 35] and Koolen et al. [24] obtain the minimax regret for the Sobolev classes, while Baby and Wang [1] for both classes with slightly stronger assumptions. However, these results are not directly related to this paper because we consider linear losses.

**Notations.**  We will use the following definitions and notations. The elements of a matrix $\boldsymbol{A} \in \mathbb{R}^{n \times m}$ are denoted by $A_{ij}$ for $i = 1, \ldots, n$ and $j = 1, \ldots, m$. Similarly, the elements of a vector $\boldsymbol{u} \in \mathbb{R}^d$ are $u_i$ for $i = 1, \ldots, d$. The Kronecker product of matrices $\boldsymbol{A} \in \mathbb{R}^{m \times n}$ and $\boldsymbol{B} \in \mathbb{R}^{p \times q}$ is the block matrix defined by

$$\boldsymbol{A} \otimes \boldsymbol{B} := \begin{pmatrix} A_{1,1}\boldsymbol{B} & \cdots & A_{1,n}\boldsymbol{B} \\ \vdots & \ddots & \vdots \\ A_{m,1}\boldsymbol{B} & \cdots & A_{m,n}\boldsymbol{B} \end{pmatrix}.$$

---

**Algorithm 1:** Dynamic-to-Static Reduction

---
**Input** *Domain* $\mathcal{W} \subseteq \mathbb{R}^d$, *Online Learning Algorithm* $\mathcal{A}$ *with domain* $\mathcal{W}^T$
**for** $t = 1 : T$ **do**

> Get $\widetilde{\boldsymbol{w}}_t = (\boldsymbol{w}_t^{(1)}, \ldots, \boldsymbol{w}_t^{(T)}) \in \mathcal{W}^T$ from $\mathcal{A}$
> Play $\boldsymbol{w}_t = \boldsymbol{w}_t^{(t)} \in \mathcal{W}$ and observe $\boldsymbol{g}_t \in \partial \ell_t(\boldsymbol{w}_t)$
> Pass $\widetilde{\boldsymbol{g}}_t = \mathbf{e}_t \otimes \boldsymbol{g}_t = (\boldsymbol{0}_d^\top, \ldots, \boldsymbol{0}_d^\top, \underbrace{\boldsymbol{g}_t^\top}_{\text{indices } i \in [d(t-1)+1, dt]}, \boldsymbol{0}_d^\top, \ldots)^\top \in \mathcal{W}^T$ to $\mathcal{A}$

**end**

---

We let $\mathbf{e}_t$ denote the $t^{\text{th}}$ standard basis vector of $\mathbb{R}^T$ and $\boldsymbol{I}_d$ is the $d \times d$ identity matrix. For a square matrix $\boldsymbol{A}$, $\text{Diag}(\boldsymbol{A})$ is the diagonal matrix that contains the elements of the diagonal of $\boldsymbol{A}$. For a positive definite matrix $\boldsymbol{M}$, we define the weighted norm $\|\boldsymbol{x}\|_{\boldsymbol{M}} := \sqrt{\langle \boldsymbol{x}, \boldsymbol{M}\boldsymbol{x} \rangle}$. For a matrix $\boldsymbol{A} \in \mathbb{R}^{m \times n}$, we denote its Frobenius norm by $\|\boldsymbol{A}\|_F := \sqrt{\sum_{i=1}^m \sum_{j=1}^n A_{i,j}^2}$. The vec operator is the mapping defined by stacking the columns of a matrix $\boldsymbol{A}$ in a vector. We will denote by $\|\boldsymbol{A}\|_{p,p}$ the entry-wise $p$-norm of $\boldsymbol{A}$, *i.e.*, $\|\boldsymbol{A}\|_{p,p} := \|\text{vec}(\boldsymbol{A})\|_p$.

## 2 A dynamic-to-static reduction

In this section, we present a general reduction from dynamic regret to static regret. The key idea is to embed the comparator sequence in a high dimensional space $\mathcal{W}^T$, where $T$ is the number of rounds, so that competing with a *fixed* comparator $\widetilde{\boldsymbol{u}} \in \mathcal{W}^T$ in this high-dimensional space is equivalent to competing with a *sequence* of comparators in the original space $\mathcal{W}$. In this way, we can reduce the problem of minimizing the dynamic regret to the one of minimizing the static regret.

Our reduction is shown in Algorithm 1. We simply embed the linear losses $\boldsymbol{g}_t$ in a high-dimensional space by setting

$$\widetilde{\boldsymbol{g}}_t = \mathbf{e}_t \otimes \boldsymbol{g}_t = (\boldsymbol{0}_d^\top, \ldots, \boldsymbol{0}_d^\top, \underbrace{\boldsymbol{g}_t^\top}_{\text{Indices } \in [d(t-1)+1, dt]}, \boldsymbol{0}_d^\top, \ldots, \boldsymbol{0}_d^\top)^\top, \tag{2}$$

where $\mathbf{e}_t \in \mathbb{R}^T$ is the $t^{\text{th}}$ standard basis vector of $\mathbb{R}^T$ and $\boldsymbol{0}_d \in \mathbb{R}^d$ denotes the vector of zeros. We pass these losses to the online learning algorithm $\mathcal{A}$, which predicts with a vector $\widetilde{\boldsymbol{w}}_t \in \mathcal{W}^T$. Finally, we set $\boldsymbol{w}_t \in \mathbb{R}^d$ equal to the $t^{\text{th}}$ "component" of $\widetilde{\boldsymbol{w}}_t$, and play $\boldsymbol{w}_t$.

We show that the dynamic regret of the resulting algorithm will be equal to the static regret of the algorithm $\mathcal{A}$. In particular, for any sequence $\vec{\boldsymbol{u}} = (\boldsymbol{u}_1, \ldots, \boldsymbol{u}_T)$ in $\mathcal{W}$ we will denote the concatenation of $\vec{\boldsymbol{u}}$ into a single vector in $\mathcal{W}^T$ as

$$\widetilde{\boldsymbol{u}} = \sum_{t=1}^T \mathbf{e}_t \otimes \boldsymbol{u}_t = (\boldsymbol{u}_1^\top, \ldots, \boldsymbol{u}_T^\top)^\top. \tag{3}$$

Then, the following proposition shows that the dynamic regret of Algorithm 1 *w.r.t* any sequence $\vec{\boldsymbol{u}} = (\boldsymbol{u}_1, \ldots, \boldsymbol{u}_T)$ is *equal* to the static regret of $\mathcal{A}$ *w.r.t* $\widetilde{\boldsymbol{u}}$.

**Proposition 1.** *Let* $\mathcal{W} \subseteq \mathbb{R}^d$ *and let* $\mathcal{A}$ *be an online learning algorithm with domain* $\mathcal{W}^T$. *Then, for any sequence* $\vec{\boldsymbol{u}} = (\boldsymbol{u}_1, \ldots, \boldsymbol{u}_T) \in \mathcal{W}^T$, *Algorithm 1 guarantees*

$$R_T(\vec{\boldsymbol{u}}) = \sum_{t=1}^T \langle \boldsymbol{g}_t, \boldsymbol{w}_t - \boldsymbol{u}_t \rangle = \sum_{t=1}^T \langle \widetilde{\boldsymbol{g}}_t, \widetilde{\boldsymbol{w}}_t - \widetilde{\boldsymbol{u}} \rangle =: R_T^{\text{Seq}}(\widetilde{\boldsymbol{u}}).$$

*Proof.* The proof is immediate from Equations (2) and (3). In fact, observe that the cumulative loss of the comparator sequence is precisely

$$\sum_{t=1}^T \langle \boldsymbol{g}_t, \boldsymbol{u}_t \rangle = \left\langle \begin{pmatrix} \boldsymbol{g}_1 \\ \vdots \\ \boldsymbol{g}_T \end{pmatrix}, \begin{pmatrix} \boldsymbol{u}_1 \\ \vdots \\ \boldsymbol{u}_T \end{pmatrix} \right\rangle = \left\langle \sum_{t=1}^T \mathbf{e}_t \otimes \boldsymbol{g}_t, \sum_{t=1}^T \mathbf{e}_t \otimes \boldsymbol{u}_t \right\rangle = \left\langle \sum_{t=1}^T \widetilde{\boldsymbol{g}}_t, \widetilde{\boldsymbol{u}} \right\rangle.$$

We get a similar relationship for the algorithm's cumulative loss. Hence, we have $R_T(\vec{\boldsymbol{u}}) = \sum_{t=1}^T \langle \boldsymbol{g}_t, \boldsymbol{w}_t - \boldsymbol{u}_t \rangle = \sum_{t=1}^T \langle \widetilde{\boldsymbol{g}}_t, \widetilde{\boldsymbol{w}}_t - \widetilde{\boldsymbol{u}} \rangle = R_T^{\text{Seq}}(\widetilde{\boldsymbol{u}})$. $\qquad \square$

**Remark 1.** *It is important to note that the regret equivalence holds in the context of **linear losses**. However, our reduction can still be leveraged for arbitrary convex losses by first applying the standard reduction to OLO:* $\sum_{t=1}^{T} \ell_t(\boldsymbol{w}_t) - \ell_t(\boldsymbol{u}_t) \leq \sum_{t=1}^{T} \langle \boldsymbol{g}_t, \boldsymbol{w}_t - \boldsymbol{u}_t \rangle = R_T^{\mathrm{Seq}}(\widetilde{\boldsymbol{u}})$ *for* $\boldsymbol{g}_t \in \partial \ell_t(\boldsymbol{w}_t)$.

While our reduction is exceptionally simple, its utility should not be understated. Proposition 1 is a regret *equivalence* — we lose nothing by taking this perspective, yet it allows us to immediately apply all the usual techniques and approaches from the static regret setting. For instance, given any dual norm pair $(\|\cdot\|, \|\cdot\|_*)$, it is well-understood how to develop algorithms which adapt simultaneously to the comparator norm $\|\widetilde{\boldsymbol{u}}\|$ and to the gradient variance $\sum_{t=1}^{T} \|\widetilde{\boldsymbol{g}}_t\|_*^2$ to guarantee

$$R_T(\boldsymbol{u}_1, \ldots, \boldsymbol{u}_T) = R_T^{\mathrm{Seq}}(\widetilde{\boldsymbol{u}}) \leq \widetilde{\mathcal{O}}\left( \|\widetilde{\boldsymbol{u}}\| \sqrt{\sum_{t=1}^{T} \|\widetilde{\boldsymbol{g}}_t\|_*^2} \right).$$

Such algorithms are commonly referred to as "parameter-free", or "comparator adaptive", because they achieve this adaptation by completely removing the parameter that depends on the unknown comparator $\widetilde{\boldsymbol{u}}$ [*e.g.*, 26, 27, 32, 9, 12, 20, 8, 21, 48]. In this way, we have effectively reduced the problem of minimizing dynamic regret to the problem of selecting a dual-norm pair $(\|\cdot\|, \|\cdot\|_*)$ that meaningfully measures the "difficulty" of the sequence in $\widetilde{\boldsymbol{u}}$ and the losses $\widetilde{\boldsymbol{g}}_t$. In particular, $(\|\cdot\|, \|\cdot\|_*)$ should be chosen with the following considerations in mind:

1. $\|\widetilde{\boldsymbol{u}}\|$ should produce a meaningful measure of variability of the comparator sequence $\boldsymbol{u}_1, \ldots, \boldsymbol{u}_T$. For instance, we will show in Proposition 2 that the squared path-length arises from a particular weighted norm applied to $\widetilde{\boldsymbol{u}}$.

2. $\|\widetilde{\boldsymbol{g}}_t\|_*$ should not "blow up". Ideally $\|\widetilde{\boldsymbol{g}}_t\|_*$ should match the magnitude of the true losses $\boldsymbol{g}_t$ up to polylog factors.

3. $(\|\cdot\|, \|\cdot\|_*)$ should be chosen with computational considerations in mind. For instance, to apply an FTRL-based algorithm to the losses $\widetilde{\boldsymbol{g}}_t \in \mathbb{R}^{dT}$, efficient implementation will typically require $\|\cdot\|_*$ to have sparse subgradients. In general, an ideal dual-norm pair should facilitate updating only $\mathcal{O}(\log T)$ variables at a time, so as to match the $\mathcal{O}(d \log T)$ per-step computation enjoyed by existing dynamic regret algorithms. We will see one such example in Section 4.1.

In the next section, we show that there is in fact a fundamental trade-off between the penalties induced by the dual-norm pair $(\|\cdot\|, \|\cdot\|_*)$, creating a tension between the first two considerations.

## 3  Lower bounds for unconstrained dynamic regret

In the static regret setting, there is a well-known trade-off between the way in which we measure the complexity of the comparator $\boldsymbol{u}$ and the way in which we measure the complexity of the linear losses $\boldsymbol{g}_t$. For example, in Online Mirror Descent [29, 42] one can get a regret guarantee that depends on the maximum diameter of the feasible set with respect to a norm $\|\cdot\|$, while the linear losses are measured using the dual norm $\|\cdot\|_*$. The equivalence in Proposition 1 suggests that a similar tension exists for the dynamic regret.

Given the structure of our reduction, it makes sense to focus on the weighted norms $\|\cdot\|_{\boldsymbol{M}}$ and $\|\cdot\|_{\boldsymbol{M}^{-1}}$, where $\boldsymbol{M}$ is a symmetric positive definite matrix. In particular, the next theorem shows that there is a fundamental trade-off between a *variability penalty* $\|\widetilde{\boldsymbol{u}}\|_{\boldsymbol{M}}$ and a *variance penalty* $G^2 \operatorname{Tr}(\boldsymbol{M}^{-1})$ related to the losses. The proof is provided in Appendix A.1 and it is based on a lower bound to the tail of Rademacher chaos of order 2.

**Theorem 1.** *Let the number of rounds $T \geq T_0$, where $T_0$ is a universal constant. Let $\mathcal{A}$ be an online learning algorithm, and suppose $\mathcal{A}$ guarantees $R_T(0) \leq G\epsilon_T$ for any sequence of linear losses $g_1, \ldots, g_T \in \mathbb{R}$ satisfying $|g_t| \leq G$. Let $\boldsymbol{M}^{-1} \in \mathbb{R}^{T \times T}$ be any symmetric positive definite matrix, denote $\widetilde{\boldsymbol{M}}^{-1} := \boldsymbol{M}^{-1} - \mathrm{Diag}\left(\boldsymbol{M}^{-1}\right)$ and $V_T := \mathrm{Tr}(\boldsymbol{M}^{-1}) + \|\widetilde{\boldsymbol{M}}^{-1}\|_F$. Suppose that $\|\widetilde{\boldsymbol{M}}^{-1}\|_F^2 \geq \frac{T}{2}\max_i \sum_j (\widetilde{M}_{ij}^{-1})^2$. Then, for any $P$ satisfying $T_0 \leq \log_2 \frac{\sqrt{PV_T}}{2\epsilon_T} \leq T$, there is a sequence of losses $g_1, \ldots, g_T \in \mathbb{R}$, and $\widetilde{\boldsymbol{u}} = (u_1, \ldots, u_T)^\top \in \mathbb{R}^T$ satisfying $\|\widetilde{\boldsymbol{u}}\|_M = \sqrt{P}$ such that we have*

$$R_T(u_1, \ldots, u_T) \geq \Omega\left( G\epsilon_T + G\sqrt{P\left[\mathrm{Tr}(\boldsymbol{M}^{-1}) + \left\|\widetilde{\boldsymbol{M}}^{-1}\right\|_F \log_2^{\frac{1}{2}} \frac{\sqrt{PV_T}}{2\epsilon_T}\right]}\right) .$$

Let us first briefly discuss the conditions on $\boldsymbol{M}$. First, note that the restriction that $\boldsymbol{M}$ be positive definite and symmetric simply specifies that $\|\cdot\|_M$ defines a valid norm. The condition on $\|\widetilde{\boldsymbol{M}}^{-1}\|_F = \|\boldsymbol{M}^{-1} - \mathrm{Diag}\left(\boldsymbol{M}^{-1}\right)\|_F$ is less straight forward to interpret, but it essentially states that the total "variance" of $\widetilde{\boldsymbol{M}}^{-1}$ is at least as much as that of any of its columns. on a technical level this assumption leads to the restriction that $P$ satisfies $\log_2\left(\sqrt{PV_T}/2\epsilon_T\right) \leq T$. This is a natural restriction which encodes the fact that if $P$ is too large relative to $T$ (*i.e.*, when $\log_2(\sqrt{PV}/2\epsilon_T) \geq T$), one can ensure "low" regret by simply playing $\boldsymbol{w}_t = \boldsymbol{0}$ on every round:

$$R_T(\vec{\boldsymbol{u}}) = -\sum_{t=1}^T \langle \boldsymbol{g}_t, \boldsymbol{u}_t \rangle \leq \max_t \|\boldsymbol{u}_t\| \, GT \leq G\max_t \|\boldsymbol{u}_t\| \log_2\left(\sqrt{PV_T}/2\epsilon\right),$$

and hence the only lower bounds in such settings are trivial ones and it suffices to consider only $P$ satisfying $\log\left(\sqrt{PV_T}/2\epsilon\right) \leq T$. We will see in Proposition 2 that the matrix that produces the squared path-length satisfies this condition, and it can be seen that symmetric matrices with equal column sums (as is the case in Proposition 3) satisfy this condition as well.

The result of Theorem 1 shows that there is a frontier of lower bounds which trade off penalties related to variability of the comparator sequence and penalties related to the variance of the subgradients. That is, one can not guarantee a small variability penalty in all situations without also accepting a large subgradient variance penalty. The next proposition shows that i) the squared path-length can be represented by a particular choice of the weighted norm $\|\widetilde{\boldsymbol{u}}\|_M$, and ii) the fundamental tension between $\|\widetilde{\boldsymbol{u}}\|_M$ and its corresponding variance penalty $\mathrm{Tr}(\boldsymbol{M}^{-1})$ prevents any algorithm from attaining the ideal variability dependence of $\|\widetilde{\boldsymbol{u}}\|_M = \left(\sqrt{\sum_{t=1}^{T-1} \|\boldsymbol{u}_t - \boldsymbol{u}_{t+1}\|^2}\right)$. In fact, the corresponding variance penalty is $G^2 \mathrm{Tr}(\boldsymbol{M}^{-1}) = \mathcal{O}(G^2 T^2)$, resulting in a vacuous guarantee. Proof of the proposition can be found in Appendix A.2.

**Proposition 2.** *(Adapting to Squared Path-length Requires Superlinear Regret) Define the finite-difference operator $\boldsymbol{\Sigma} \in \mathbb{R}^T$ as the matrix with entries*

$$\Sigma_{ij} = \begin{cases} 1 & \text{if } i = j \\ -1 & \text{if } i = j - 1 \\ 0 & \text{otherwise} \end{cases}.$$

*Let $\mathbf{S} = \boldsymbol{\Sigma}^\top \boldsymbol{\Sigma}$ and $\boldsymbol{M} = \mathbf{S} \otimes \boldsymbol{I}_d$. Then, $\boldsymbol{M}$ satisfies the assumptions of Theorem 1 and*

$$\|\widetilde{\boldsymbol{u}}\|_M^2 = \|\boldsymbol{u}_T\|_2^2 + \sum_{t=1}^{T-1} \|\boldsymbol{u}_t - \boldsymbol{u}_{t+1}\|_2^2 \qquad \text{and} \qquad \mathrm{Tr}\left(\boldsymbol{M}^{-1}\right) = \frac{T(T+1)}{2} .$$

Proposition 2 shows that adapting to the squared path-length of an arbitrary comparator sequence *necessarily requires* incurring a linear penalty, so adapting to the squared path-length is impossible without facing a vacuous guarantee. However, we will show in Section 4.1 that it is possible to adapt to a measure of variability which is similar in spirit to the squared path-length, yet only incurs a $\mathrm{Tr}(\boldsymbol{M}^{-1}) = \mathcal{O}(\log T)$ variance penalty.

**Remark 2.** *The matrix $\boldsymbol{M}$ in Proposition 2 uniquely exposes the the squared path-length up to the bias term $\|\boldsymbol{u}_T\|^2$. Such a bias term must appear because in the static regret setting, wherein $\boldsymbol{u}_1 = \ldots = \boldsymbol{u}_T = \boldsymbol{u}$, the variability measure $\|\cdot\|_M$ must still reduce to a dependence on $\|\boldsymbol{u}\|$, otherwise*

---
**Algorithm 2:** Dynamic regret OLO through 1-dimensional reduction [9]
---
**Input** *1-d Parameter-free OLO algorithm $\mathcal{A}$, positive definite symmetric matrix $\boldsymbol{M} \in \mathbb{R}^{dT \times dT}$*
**Initialize** $\widetilde{\boldsymbol{\theta}}_1 = \widetilde{\boldsymbol{v}}_1 = \mathbf{0} \in \mathbb{R}^{dT}$, $V_1 = 0$
**for** $t = 1 : T$ **do**
$\quad$ Get $\beta_t \in \mathbb{R}$ from $\mathcal{A}$
$\quad$ Play $\widetilde{\boldsymbol{w}}_t = \beta_t \widetilde{\boldsymbol{v}}_t$ and observe $\widetilde{\boldsymbol{g}}_t$
$\quad$ Send $\langle \widetilde{\boldsymbol{v}}_t, \widetilde{\boldsymbol{g}}_t \rangle$ to $\mathcal{A}$ as the $t^{\text{th}}$ loss

$\quad$ Set $\widetilde{\boldsymbol{\theta}}_{t+1} = \widetilde{\boldsymbol{\theta}}_t - \boldsymbol{M}^{-1} \widetilde{\boldsymbol{g}}_t$
$\quad$ Set $V_{t+1} = V_t + \|\widetilde{\boldsymbol{g}}_t\|_{\boldsymbol{M}^{-1}}^2$
$\quad$ Set $\widetilde{\boldsymbol{v}}_{t+1} = \frac{\widetilde{\boldsymbol{\theta}}_{t+1}}{\sqrt{V_{t+1}}} \left[ 1 \wedge \frac{\sqrt{V_{t+1}}}{\|\widetilde{\boldsymbol{\theta}}_{t+1}\|_{\boldsymbol{M}^{-1}}} \right]$ $\quad$ // (Projected) Scale-free FTRL update
**end**
---

*the guarantee would violate existing $\widetilde{\Omega}(\|\boldsymbol{u}\| \sqrt{T})$ lower bounds for static regret. More generally, we show in Appendix A.3 that any other choice of bias would similarly lead to $\text{Tr}(\boldsymbol{M}^{-1}) \geq \Omega(T^2)$, so Proposition 2 along with our lower bound in Theorem 1 are sufficient to show that adapting to squared path-length requires accepting a vacuous guarantee.*

## 4 Dynamic regret for unconstrained OLO via weighted norms

So far, we've seen that there exists a frontier of lower bounds trading off a variability penalty, measured by $\|\widetilde{\boldsymbol{u}}\|_{\boldsymbol{M}}$, and a loss variance penalty, measured by $\text{Tr}(\boldsymbol{M}^{-1})$, and that the tension between these two quantities makes it impossible to adapt to the squared path-length of the comparator sequence without accepting a vacuous regret guarantee. A natural next question is whether there are choices of $\boldsymbol{M}$ which lead to a more favorable trade-off of these two quantities. In this section, we provide a simple framework for achieving lower bounds along the frontier described by Theorem 1, and an instance which successfully achieves an improved variance/variability trade-off. The guarantees on the lower bound frontier can be achieved using any parameter-free algorithm along with the 1-dimensional reduction of Cutkosky and Orabona [9] to extend the algorithm to dual-norm pair $(\|\cdot\|_{\boldsymbol{M}}, \|\cdot\|_{\boldsymbol{M}^{-1}})$. The generic procedure is summarized in Algorithm 2 for convenience.

**Theorem 2.** *Let $\mathbf{S} \in \mathbb{R}^{T \times T}$ be a symmetric positive definite matrix, $\boldsymbol{M} = \mathbf{S} \otimes \boldsymbol{I}_d$, and $\epsilon > 0$. There is an algorithm $\mathcal{A}$ such that for any $\boldsymbol{g}_1, \ldots, \boldsymbol{g}_T \in \mathbb{R}^d$ satisfying $\|\boldsymbol{g}_t\|_2 \leq G$ for all $t$ and any sequence $\vec{\boldsymbol{u}} = (\boldsymbol{u}_1, \ldots, \boldsymbol{u}_T) \in \mathbb{R}^{dT}$, the dynamic regret is bounded as*

$$R_T(\vec{\boldsymbol{u}}) \leq \mathcal{O}\left( \mathfrak{G}\epsilon + \|\widetilde{\boldsymbol{u}}\|_{\boldsymbol{M}} \left[ \sqrt{V_T \log\left( \frac{\|\widetilde{\boldsymbol{u}}\|_{\boldsymbol{M}} \sqrt{V_T}}{\mathfrak{G}\epsilon} + 1 \right)} \vee \mathfrak{G} \log\left( \frac{\|\widetilde{\boldsymbol{u}}\|_{\boldsymbol{M}} \sqrt{V_T}}{\epsilon \mathfrak{G}} \right) \right] \right),$$

*where $V_T = \sum_{t=1}^T \|\widetilde{\boldsymbol{g}}_t\|_{\boldsymbol{M}^{-1}}^2$ and $\mathfrak{G} = G \|\mathbf{S}^{-1}\|_{\infty, \infty}$.*

For the proof, we will need the following technical lemma.

**Lemma 1.** *Let $\mathbf{S} \in \mathbb{R}^{T \times T}$ be a symmetric positive definite matrix and let $\boldsymbol{M} = \mathbf{S} \otimes \boldsymbol{I}_d$. For $t = 1, \ldots, T$, let $\boldsymbol{g}_t \in \mathbb{R}^d$ and let $\widetilde{\boldsymbol{g}}_t = \mathbf{e}_t \otimes \boldsymbol{g}_t$. Then, we have $\|\widetilde{\boldsymbol{g}}_t\|_{\boldsymbol{M}}^2 = \|\boldsymbol{g}_t\|_2^2 S_{tt}$.*

*Proof.* Using the mixed-product property $(A \otimes B)(C \otimes D) = AC \otimes BD$ and the transpose property $(A \otimes B)^\top = A^\top \otimes B^\top$ of the Kronecker product, we have that

$$\langle \widetilde{\boldsymbol{g}}_t, \boldsymbol{M}\widetilde{\boldsymbol{g}}_t \rangle = \langle \mathbf{e}_t \otimes \boldsymbol{g}_t, [\mathbf{S} \otimes \boldsymbol{I}_d] \, \mathbf{e}_t \otimes \boldsymbol{g}_t \rangle = \langle \mathbf{e}_t \otimes \boldsymbol{g}_t, \mathbf{S}\mathbf{e}_t \otimes \boldsymbol{g}_t \rangle = (\mathbf{e}_t^\top \otimes \boldsymbol{g}_t^\top)(\mathbf{S}\mathbf{e}_t \otimes \boldsymbol{g}_t)$$
$$= \mathbf{e}_t^\top \mathbf{S}\mathbf{e}_t \otimes \boldsymbol{g}_t^\top \boldsymbol{g}_t = S_{tt} \|\boldsymbol{g}_t\|^2 \ . \qquad \square$$

*Proof of Theorem 2.* Applying Proposition 1, we have $R_T(\vec{\boldsymbol{u}}) = \sum_{t=1}^T \langle \widetilde{\boldsymbol{g}}_t, \widetilde{\boldsymbol{w}}_t - \widetilde{\boldsymbol{u}} \rangle = R_T^{\text{Seq}}(\widetilde{\boldsymbol{u}})$. Since $\boldsymbol{M}$ is symmetric and positive definite, $(\|\cdot\|_{\boldsymbol{M}}, \|\cdot\|_{\boldsymbol{M}^{-1}})$ is a valid dual-norm pair. By Lemma 1,

we have $\|\widetilde{g}_t\|_{M^{-1}}^2 = \|g_t\|_2^2 S_{tt}^{-1} \leq G^2 \|\mathbf{S}^{-1}\|_{\infty,\infty} := \mathfrak{G}^2$. Hence, let $\mathcal{A}$ be any algorithm which guarantees a parameter-free regret *w.r.t.* $(\|\cdot\|, \|\cdot\|_*)$ on losses satisfying $\|\widetilde{g}_t\|_{M^{-1}} \leq \mathfrak{G}$. Note that any parameter-free algorithm can be extended to handle arbitrary dual-norm pairs by leveraging the one-dimensional reduction of Cutkosky and Orabona [9, Section 3], that reduces the OLO problem to a unconstrained 1d problem plus an OLO problem in the unitary ball defined by the primal norm. For instance, applying Jacobsen and Cutkosky [20, Algorithm 1] with the one-dimensional reduction one can easily show (see details in Appendix B.1)

$$R_T(\vec{u}) \leq \mathcal{O}\left(\mathfrak{G}\epsilon + \|\widetilde{u}\|_M \left[\sqrt{V_T \log\left(\frac{\|\widetilde{u}\|_M \sqrt{V_T}\Lambda_T}{\mathfrak{G}\epsilon} + 1\right)} \vee \mathfrak{G}\log\left(\frac{\|\widetilde{u}\|_M \sqrt{V_T}\Lambda_T}{\epsilon\mathfrak{G}}\right)\right]\right),$$

where $V_T = \sum_{t=1}^T \|\widetilde{g}_t\|_{M^{-1}}^2$ and $\Lambda_T = \log^2(\sum_{t=1}^T \|\widetilde{g}_t\|_{M^{-1}}^2 / \mathfrak{G}^2) \leq \mathcal{O}(\log^2 T)$. $\qquad\square$

Note in particular that by Lemma 1, we have $\sum_{t=1}^T \|\widetilde{g}_t\|_{M^{-1}}^2 = \sum_{t=1}^T S_{tt}^{-1} \|g_t\|^2 \leq G \sum_{t=1}^T S_{tt}^{-1} = G\operatorname{Tr}(\mathbf{S}^{-1})$, so this bound matches the lower bound from Section 3, up to polylogarithmic terms.[3] Thus, any valid choice of $M$ will be on the lower bound frontier of Section 3.

## 4.1 Trading-off Variance and Variability

Leveraging the algorithm characterized by Theorem 2, we now show that it is indeed possible to choose $M$ such that $\sum_{t=1}^T \|\widetilde{g}_t\|_{M^{-1}}^2$ is only $\mathcal{O}(\log(T) \sum_{t=1}^T \|g_t\|^2)$, in exchange for a variability penalty which is still similar in spirit to the squared path-length.

Inspired by the Haar OLR algorithm of [47], we apply Theorem 2 using $\mathbf{S} = \mathbf{H}_n \mathbf{H}_n^\top$, where $\mathbf{H}_n$ is the unnormalized Haar basis matrix of order $n = \lceil \log_2 T \rceil$. The Haar wavelet transform and its basis matrix are common tools in the signal processing literature; we recall the basic definitions and facts for convenience in Appendix B.2. With this choice, we have the following bounds on $\|\widetilde{u}\|_M$ and $\|\widetilde{g}_t\|_{M^{-1}}^2$. The proof can be found in Appendix B.3.

**Proposition 3.** *Let $n = \log_2 T$ and $\mathbf{H}_n$ be the unnormalized Haar basis matrix of order $n$. For any $\tau \in \{2^i : i = 0, \ldots, \log_2 T\}$, let $N_\tau = T/\tau$ and let $\mathcal{I}_1^{(\tau)}, \ldots, \mathcal{I}_{N_\tau}^{(\tau)}$ be a partition of $[T]$ into intervals of length $\tau$. Define the average comparator in interval $\mathcal{I}_i^{(\tau)}$ to be $\bar{u}_i^{(\tau)} = \frac{1}{\tau}\sum_{t\in\mathcal{I}_i^{(\tau)}} u_t$, and define the* squared path-length at time-scale $\tau < T$ *to be*

$$\bar{P}(\vec{u}, \tau) := \sum_{i=1}^{N_\tau/2} \left\|\bar{u}_{2i-1}^{(\tau)} - \bar{u}_{2i}^{(\tau)}\right\|_2^2,$$

*and $\bar{P}(\vec{u}, T) = \left\|\bar{u}_1^{(T)}\right\|_2^2 = \|\bar{u}\|_2^2$. Then, setting $\mathbf{S} = [\mathbf{H}_n \mathbf{H}_n^\top]^{-1}$ and $M = \mathbf{S} \otimes I_d$, we have*

$$\|\widetilde{u}\|_M^2 \leq \|\bar{u}\|_2^2 + \frac{1}{4}\sum_{i=0}^{\log_2(T)} \bar{P}(\vec{u}, 2^i) \leq \|\bar{u}\|_2^2 + \frac{1}{4}\log(T)\max_\tau \bar{P}(\vec{u}, \tau),$$

$$\|\widetilde{g}_t\|_{M^{-1}}^2 = \|g_t\|_2^2(1 + \log T).$$

Summarizing, by applying Algorithm 1 with $\mathbf{S} = [\mathbf{H}_n \mathbf{H}_n^\top]^{-1}$ we ensure regret

$$R_T(\vec{u}) \leq \widetilde{\mathcal{O}}\left(\sqrt{\left(\|\bar{u}\|_2^2 + \max_\tau \sum_{i=1}^{N_\tau/2} \left\|\bar{u}_{2i+1}^{(\tau)} - \bar{u}_{2i}^{(\tau)}\right\|_2^2\right) \sum_{t=1}^T \|g_t\|_2^2}\right).$$

This is the first *fully decoupled* guarantee for general dynamic regret which incurs no pessimistic multiplicative penalties of the form $\max_{t,t'} \|u_t - u_{t'}\|$. That is, the terms depending on the comparators and the terms depending on the gradients appear in separate sums. Moreover, observe that

---

[3]Note that the lower bound is stated for $d = 1$, in which case $\operatorname{Tr}(\mathbf{S}^{-1}) = \operatorname{Tr}(M^{-1})$.

this measure of variability can immediately be related to the more standard (first-order/non-squared) path-length using the local averaging lemma of Zhang et al. [47] (Lemma D.7). We have

$$\|\widetilde{\boldsymbol{u}}\|_{\boldsymbol{M}}^2 \le \|\bar{\boldsymbol{u}}\|_2^2 + \frac{\log_2 T}{4} \max_\tau \sum_{i=1}^{N_\tau/2} \left\| \bar{\boldsymbol{u}}_{2i-1}^{(\tau)} - \bar{\boldsymbol{u}}_{2i}^{(\tau)} \right\|_2^2 \le \widetilde{\mathcal{O}} \left( \bar{D}^2 + \max_\tau \bar{D} \sum_{i=1}^{N_\tau/2} \left\| \bar{\boldsymbol{u}}_{2i-1}^{(\tau)} - \bar{\boldsymbol{u}}_{2i}^{(\tau)} \right\|_2 \right)$$

$$\le \widetilde{\mathcal{O}} \left( \bar{D}^2 + \bar{D} \sum_{t=1}^{T-1} \|\boldsymbol{u}_t - \boldsymbol{u}_{t+1}\|_2 \right) \le \widetilde{\mathcal{O}} \left( \bar{D}^2 + \bar{D} P_T \right),$$

where $\bar{D} = \max_{\tau,i} \left\| \bar{\boldsymbol{u}}_i^{(\tau)} - \bar{\boldsymbol{u}}_{i+1}^{(\tau)} \right\| \le \max_{i,j} \|\boldsymbol{u}_i - \boldsymbol{u}_j\|$. Thus, applying Algorithm 1 with dual-norm pair $(\|\cdot\|_{\mathbf{H}_n^{-\top}\mathbf{H}_n^\top}, \|\cdot\|_{\mathbf{H}_n \mathbf{H}_n^\top})$ still guarantees worst-case regret

$$R_T(\vec{\boldsymbol{u}}) \le \widetilde{\mathcal{O}} \left( \|\widetilde{\boldsymbol{u}}\|_{\mathbf{H}_n^{-\top}\mathbf{H}_n^{-1}} \sqrt{\textstyle\sum_{t=1}^T \|\widetilde{\boldsymbol{g}}_t\|_{\mathbf{H}_n \mathbf{H}_n^\top}^2} \right) \le \widetilde{\mathcal{O}} \left( \sqrt{\left( \|\bar{\boldsymbol{u}}\|_2^2 + \bar{D} P_T \right) \textstyle\sum_{t=1}^T \|\boldsymbol{g}_t\|_2^2} \right),$$

which matches the guarantees of prior works, up to polylogarithmic terms.

Importantly, with $\boldsymbol{M} = \boldsymbol{H}_n^{-\top} \boldsymbol{H}_n^{-1} \otimes \boldsymbol{I}_d$ the dual-norm pair $(\|\cdot\|_{\boldsymbol{M}}, \|\cdot\|_{\boldsymbol{M}^{-1}})$ leads to updates that can be implemented efficiently, in requiring only $O(\log T)$ variables to be updated on each round. This is because the Haar basis matrices are *locally supported* — the columns of $\boldsymbol{H}_n = \begin{pmatrix} \boldsymbol{h}^{(1)} & \dots & \boldsymbol{h}^{(T)} \end{pmatrix} \in \mathbb{R}^{T \times T}$ form an orthogonal basis with the property that for any $t$, $[\boldsymbol{h}^{(i)}]_t \ne 0$ for only $1 + \log_2 T$ indices $i$ (see Proposition 5). Hence, $(\boldsymbol{H}^\top \otimes \boldsymbol{I}_d)\widetilde{\boldsymbol{g}}_t = (\boldsymbol{H}^\top \otimes \boldsymbol{I}_d)(\mathbf{e}_t \otimes \boldsymbol{g}_t) = (\boldsymbol{H}^\top \mathbf{e}_t) \otimes \boldsymbol{g}_t$, is a block vector with only $1 + \log_2 T$ active blocks, requiring that we update only $O(d \log T)$ indices to maintain each of the variables needed to implement Algorithm 2. We provide the full details of this computation in Appendix B.4, which we summarize below in Proposition 4.

**Proposition 4.** *The algorithm characterized by applying Theorem 2 with $\boldsymbol{S} = [\boldsymbol{H}_n \boldsymbol{H}_n^\top]^{-1}$ can be implemented with $\mathcal{O}(d \log T)$ per-round computation.*

## 5 Recovering Variance-Variability Coupling Guarantees

Our main focus throughout the paper has been on designing algorithms that achieve a regret bounds of the form $R_T(\vec{\boldsymbol{u}}) \le O \left( \sqrt{f(\boldsymbol{u}_1, \dots, \boldsymbol{u}_T) V(\boldsymbol{g}_1, \dots, \boldsymbol{g}_T)} \right)$ for some functions $f$ and $V$, which cleanly separates the penalties associated with difficult *loss* sequences from the penalties associated with difficult *comparator* sequences. However, the first works to achieve unconstrained dynamic regret guarantees uncovered guarantees of a slightly different form, containing a *gradient-comparator correlation* penalty:

$$R_T(\vec{\boldsymbol{u}}) \le \widetilde{O} \left( \sqrt{\sum_{t=1}^{T-1} \|\boldsymbol{u}_t - \boldsymbol{u}_{t+1}\| \underbrace{\sum_{t=1}^T \|\boldsymbol{g}_t\|^2 \|\boldsymbol{u}_t - \bar{\boldsymbol{u}}\|}_{\text{Variance/Variability coupling}}} \right), \tag{4}$$

for some reference point $\bar{\boldsymbol{u}}$ [20, 47]. Guarantees of this form allow some degree of coupling between the variability and variance penalties. This can be appealing in certain situations. For instance, guarantees of the form above have the appealing property that the variance penalty completely disappears on any rounds where the comparator $\boldsymbol{u}_t$ matches the reference point $\bar{\boldsymbol{u}}$. This can be a very powerful property when one has *a priori* access to a benchmark model (represented by $\bar{\boldsymbol{u}}$) which can be expected to predict well *on average*, so that we accumulate the variance penalties only when facing atypical/unexpected conditions.

The prior works achieving a coupling guarantee do so using rather mysterious means. For instance, the guarantee of Jacobsen and Cutkosky [20] achieves the coupling guarantee almost by coincidence, as it appears in response to a composite regularizer they add to the update to cancel out certain unstable terms in the analysis, and the analysis of Zhang et al. [47] recovers a guarantee of a similar form using a rather difficult analysis of the frequency-domain representation of $\widetilde{\boldsymbol{u}}$ after projecting onto the Haar basis vectors. So far there is no unifying explanation of the principles leading to these sorts of guarantees.

Our equivalence in Proposition 1 instead shows that guarantees of the form Equation (4) can instead be understood through the lens of reward-regret duality, a standard tool used to design algorithms in the static regret setting. The reward-regret duality states that in order to guarantee regret of the form $R_T(\boldsymbol{u}) \leq f(\boldsymbol{u})$ for all $\boldsymbol{u} \in \mathcal{W}$, it suffices to design an algorithm that guarantees $-\sum_{t=1}^T \langle \boldsymbol{g}_t, \boldsymbol{w}_t \rangle \geq f^*(-\sum_{t=1}^T \boldsymbol{g}_t)$ for any $\boldsymbol{g}_1, \ldots, \boldsymbol{g}_T$. Using Proposition 1, we immediately have the following analogous design principle for dynamic regret. Proof is deferred to Appendix C.1.

**Theorem 3.** *Let* $\mathrm{Wealth}_T := -\sum_{t=1}^T \langle \widetilde{\boldsymbol{g}}_t, \widetilde{\boldsymbol{w}}_t \rangle$ *denote the "wealth" of an algorithm* $\mathcal{A}$ *and let* $(f, f^*)$ *be a Fenchel conjugate pair. Then* $\mathcal{A}$ *guarantees* $\mathrm{Wealth}_T \geq f_T^*\left(-\sum_{t=1}^T \widetilde{\boldsymbol{g}}_t\right)$ *for any sequence* $\widetilde{\boldsymbol{g}}_1, \ldots, \widetilde{\boldsymbol{g}}_T$ *if and only if* $R_T(\vec{\boldsymbol{u}}) \leq f_T(\widetilde{\boldsymbol{u}})$ *for any sequence* $\vec{\boldsymbol{u}} = (\boldsymbol{u}_1, \ldots, \boldsymbol{u}_T)$ *in* $\mathcal{W}$, *where* $\widetilde{\boldsymbol{u}} = (\boldsymbol{u}_1^\top, \ldots, \boldsymbol{u}_T^\top)^\top$ *is the concatenation of the sequence* $\vec{\boldsymbol{u}}$ *into a vector.*

So, suppose we would like to design an algorithm that guarantees for any sequence $\vec{\boldsymbol{u}} = (\boldsymbol{u}_1, \ldots, \boldsymbol{u}_T)$ and any $\vec{\boldsymbol{g}} = (\boldsymbol{g}_1, \ldots, \boldsymbol{g}_T)$ regret of the form

$$R_T(\vec{\boldsymbol{u}}) \leq \sqrt{f_T(\widetilde{\boldsymbol{u}}) V_T(\widetilde{\boldsymbol{u}})},$$

for some $f_T(\widetilde{\boldsymbol{u}})$ and $V_T(\widetilde{\boldsymbol{u}}) = V_T(\widetilde{\boldsymbol{u}}; \vec{\boldsymbol{g}})$. Then, since $\sqrt{ab} = \min_{\eta \geq 0} \frac{a}{2\eta} + \frac{\eta}{2} b$, any such algorithm must have $R_T(\vec{\boldsymbol{u}}) \leq \frac{f_T(\widetilde{\boldsymbol{u}})}{2\eta} + \frac{\eta}{2} V_T(\widetilde{\boldsymbol{u}})$ for every $\eta \geq 0$. So, via Proposition 1 and the the reward-regret duality of Theorem 3, we have that the desired guarantee is equivalent to guaranteeing for all $\eta \geq 0$ a wealth lower bound of

$$\mathrm{Wealth}_t = -\sum_{t=1}^T \langle \widetilde{\boldsymbol{g}}_t, \widetilde{\boldsymbol{w}}_t \rangle \geq \left[ \frac{f_T(\cdot)}{2\eta} + \frac{\eta}{2} V_T(\cdot) \right]^* \left( -\widetilde{\boldsymbol{g}}_{1:T} \right) = \frac{f_T^*\left( -2\eta \widetilde{\boldsymbol{g}}_{1:T} \right)}{2\eta} \,\square\, 2\eta V_T^* \left( \frac{\widetilde{\boldsymbol{g}}_{1:T}}{2\eta} \right),$$

where $f_T^*$ and $V_T^*$ are the Fenchel conjugates of $f_T$ and $V_T$ respectively, and $(f_1 \,\square\, f_2)$ denotes the *infimal convolution* [34, 19] of $f_1$ and $f_2$:

$$(f_1 \,\square\, f_2)(z) = \inf \left\{ f_1(y) + f_2(z - y) \right\}.$$

Thus, the variance/variability coupling guarantees observed in Equation (4) can be interpreted as achieving wealth lower-bounds for potential functions involving infimal convolution.

The above discussion provides a general characterization of variance/variability coupling guarantees, though it is admittedly less clear how difficult it is to design algorithms from this perspective due to the rather complicated potential function that appears. Nonetheless, we believe that this provides a valuable perspective and insight that could be of general interest. An important direction for future work is to develop useful tools for working with potential functions of this form.

## 6  Conclusion

In this paper, we have shown a way to reduce the problem of dynamic regret minimization to the static one. We proved a novel frontier of lower bounds showing a fundamental trade-off between penalties on the comparators and penalties on the variance of the gradients. In particular, we have shown that it is not possible to achieve a guarantee that scales with $\sqrt{\sum_{t=1}^{T-1} \|\boldsymbol{u}_t - \boldsymbol{u}_{t+1}\|^2}$ without incurring a variance penalty of $\mathcal{O}(GT)$. We developed a simple framework for achieving guarantees along the lower bound frontier, and used it to develop the first algorithm making a non-trivial variance/variability decoupling guarantee against arbitrary comparator sequences. Our framework is simple but powerful, allowing one to fully utilize the rich literature of static regret algorithms for online learning.

We conclude by noting some directions for future work. There is a lot of exciting potential to explore different measures of variability induced by different choices of the matrix $\boldsymbol{M}$, as well as going beyond weighted norms. As mentioned in Section 5, developing a useful toolset for potential functions involving infimal convolution is an important next-step for developing and understanding guarantees with a coupled variance/variability penalty, such as Equation (4). Also, our lower bound in Section 3 illustrates the variance-variability trade-off, but achieving the correct logarithmic dependencies proved to be very challenging — many of the standard tools for proving lower bounds in unconstrained settings revolve around anti-concentration results that do not readily extend to arbitrary weighted norms and higher-dimensions. We look forward to exciting development in these future directions.

## Acknowledgments

We thank Yu-Xiang Wang for the discussion on the function classes studied in non-parametric regression theory.

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

# A   Proofs for Section 3 (Lower bounds for unconstrained dynamic regret)

In this section, we provide proof of our main lower bound result from Section 3. We first introduce a technical tool from the literature on decoupling theory and a key lemma (Lemma 2). Proof of our main result is in Appendix A.1.

Consider a function $f : [-1, 1]^d \to \mathbb{R}$, defined as

$$f(\boldsymbol{x}) = \sum_{i,j} A_{i,j} x_i x_j \,,$$

Define $\boldsymbol{A}$ the matrix with elements $A_{i,j}$. In this section we will use the following notations for quantities related to a polynomial induced by the quadratic form $\boldsymbol{x} \mapsto \langle \boldsymbol{x}, \boldsymbol{A}\boldsymbol{x} \rangle$ (see page 6 of O'Donnell and Zhao [30])

$$\mathrm{Var}[f] = \sum_{i,j} A_{i,j}^2 = \|A\|_F^2,$$

$$\mathrm{Inf}_i[f] = \sum_{j=1}^{d} (A_{i,j}^2 + A_{j,i}^2) \,.$$

One of the key difficulties in deriving the lower bound is that squared weighted norms $\boldsymbol{x} \mapsto \langle \boldsymbol{x}, \boldsymbol{A}\boldsymbol{x} \rangle$ introduce dependencies between the coordinates of $\boldsymbol{x}$, which breaks the usual lower bound arguments which rely on anti-concentration of *independent* Rademacher random variables. Instead, we must leverage an anti-concentration result that holds for *polynomials* of random variables.

**Theorem 4** (Theorem 3 of Dinur et al. [10])**.** *There is a universal constant $C$ such that the following holds. Suppose $G : \{\pm 1\}^d \to \mathbb{R}$ is a polynomial of degree at most 2 and assume $\mathrm{Var}[g] = 1$. Let $t \geq 1$ and suppose that $\mathrm{Inf}_i[g] \leq C^{-2}t^{-2}$ for all $i \in [d]$. Then*

$$\mathbb{P}\{|g(x)| \geq t\} \geq \exp\left(-C^2 t^2 4 \log 2\right) \,.$$

Using this anti-concentration result, the following key lemma provides a general lower bound on the wealth obtainable by any algorithm, subject to the weighting imposed by a matrix $\boldsymbol{A}$.

**Lemma 2.** *Let $\mathcal{A}$ be an online learning algorithm, and suppose $\mathcal{A}$ guarantees $R_T(0) \leq G\epsilon_T$ for any sequence of linear losses $g_1, \ldots, g_T \in \mathbb{R}$ satisfying $|g_t| \leq G$. Let $\boldsymbol{A} \in \mathbb{R}^{T \times T}$ be any symmetric positive definite matrix, and let $\boldsymbol{B} = \boldsymbol{A} - Diag\,(\boldsymbol{A})$. Then, there is a universal constant $C > 0$ such that for any $1 \leq q \leq \frac{\|\boldsymbol{B}\|_F}{C\sqrt{2\max_i \sum_{j=1}^{T} B_{ij}^2}}$, there is a sequence of losses $g_1, \ldots, g_T \in \mathbb{R}$ such that*

$$\left\| \begin{pmatrix} g_1 \\ \vdots \\ g_T \end{pmatrix} \right\|_{\boldsymbol{A}}^2 \geq G^2 \left[\mathrm{Tr}(\boldsymbol{A}) + q \,\|\boldsymbol{A} - Diag\,(\boldsymbol{A})\|_F\right]$$

*and*

$$R_T(0) \geq G\epsilon_T \left[1 - 2^{4C^2 q^2}\right] \,.$$

*Proof.* Let $Y_1, \ldots, Y_T$ be independent Rademacher random variables and set $g_t = G Y_t$, so that $\mathbb{E}\,[R_T(0)] = \mathbb{E}\left[\sum_{t=1}^{T} g_t w_t\right] = 0$. Then, using the regret equivalence of Proposition 1 and conditioning on any event $\mathcal{E}$ with $\mathbb{P}\{\mathcal{E}\} > 0$, we have

$$0 = \mathbb{E}\,[R_T(0)]$$
$$= \mathbb{E}\left[R_T(0)\Big|\mathcal{E}\right]\mathbb{P}\{\mathcal{E}\} + \mathbb{E}\left[R_T(0)\Big|\mathcal{E}^c\right]\mathbb{P}\{\mathcal{E}^c\}$$
$$\leq \mathbb{E}\,[R_T(0)|\mathcal{E}]\,\mathbb{P}\{\mathcal{E}\} + G\,\epsilon_T\,(1 - \mathbb{P}\{\mathcal{E}\})\,,$$

where the last line uses the fact that $\mathcal{A}$ guarantees $R_T(0) \leq G\epsilon_T$ for any $g_1, \ldots, g_T$ satisfying $|g_t| \leq G$ for all $t$. Re-arranging, we have

$$\mathbb{E}\left[R_T(0)\Big|\mathcal{E}\right] \geq G\epsilon_T \left(1 - \frac{1}{\mathbb{P}\{\mathcal{E}\}}\right) \,. \tag{5}$$

Next, let $\widetilde{\boldsymbol{g}}_t = \mathbf{e}_t \otimes g_t$ for all $t$ and consider the event

$$\mathcal{E} = \left\{ \left\| \sum_{t=1}^{T} \widetilde{\boldsymbol{g}}_t \right\|_{\boldsymbol{A}}^2 = \left\| (g_1, \ldots, g_T)^\top \right\|_{\boldsymbol{A}}^2 \geq \mathrm{Tr}(\boldsymbol{A}) + q \left\| \boldsymbol{A} - \mathrm{Diag}\,(\boldsymbol{A}) \right\|_F \right\}$$

for some $q > 0$. We proceed by lower bounding the probability of this event.

Observe that

$$\left\| \sum_{t=1}^{T} \widetilde{\boldsymbol{g}}_t \right\|_{\boldsymbol{A}}^2 = G^2 \sum_{i,j} Y_i Y_j A_{ij} = G^2 \left[ \mathrm{Tr}(\boldsymbol{A}) + \sum_{i,j \neq i} Y_i Y_j A_{ij} \right].$$

Denote $\boldsymbol{B} = \boldsymbol{A} - \mathrm{Diag}\,(\boldsymbol{A})$ and note that $f(Y_1, \ldots, Y_T) = \sum_{i,j} Y_i Y_j B_{ij}$ is a polynomial of degree at most 2 and variance $\mathrm{Var}[f] = \sum_{i,j} B_{ij}^2 = \left\| \boldsymbol{A} - \mathrm{Diag}\,(\boldsymbol{A}) \right\|_F^2 = \left\| \boldsymbol{B} \right\|_F^2$. Moreover, since $\boldsymbol{A}$ is symmetric we have $\mathrm{Inf}_i[f] = \sum_{j=1}^{T} B_{ij}^2 + B_{ji}^2 = 2 \sum_{j=1}^{T} B_{ij}^2$ for any $i$. It follows that if we let $g(\boldsymbol{Y}) = \frac{f(\boldsymbol{Y})}{\sqrt{\|\boldsymbol{B}\|_F^2}} = \frac{f(\boldsymbol{Y})}{\|\boldsymbol{B}\|_F}$, then $g$ is a polynomial of degree at most 2, $\mathrm{Var}[g] = 1$, and for any $i \in [T]$ we have $\mathrm{Inf}_i[g] = \frac{2 \sum_{j=1}^{T} B_{ij}^2}{\|\boldsymbol{B}\|_F^2}$. Hence by Theorem 4, there is a universal constant $C$ such that for any $1 \leq q \leq \frac{\|\boldsymbol{B}\|_F}{C \sqrt{2 \max_i \sum_{j=1}^{T} B_{ij}^2}}$ it holds that

$$\mathbb{P}\left\{ f(\boldsymbol{Y}) \geq q \|\boldsymbol{B}\|_F \right\} = \mathbb{P}\left\{ g(\boldsymbol{Y}) \geq q \right\} \geq \exp\left( -4C^2 q^2 \log 2 \right) = 2^{-4C^2 q^2}.$$

Now observe that $\mathbb{P}\{\mathcal{E}\} = \mathbb{P}\{ f(\boldsymbol{Y}) \geq q \|\boldsymbol{B}\|_F \}$ by construction, so Equation (5) can be bound as

$$\mathbb{E}\left[ R_T(0) \big| \mathcal{E} \right] \geq G\epsilon_T \left( 1 - \frac{1}{\mathbb{P}\{\mathcal{E}\}} \right) = G\epsilon_T \left( 1 - 2^{4C^2 q^2} \right),$$

which implies the existence of a sequence $g_1, \ldots, g_T \in \mathbb{R}$ such that $R_T(0) \geq G\epsilon_T \left[ 1 - 2^{4C^2 q^2} \right]$ and

$$\left\| \sum_{t=1}^{T} \widetilde{\boldsymbol{g}}_t \right\|_{\boldsymbol{A}}^2 \geq G^2 \left[ \mathrm{Tr}(\boldsymbol{A}) + q \|\boldsymbol{B}\|_F \right] = G^2 \left[ \mathrm{Tr}(\boldsymbol{A}) + q \left\| \boldsymbol{A} - \mathrm{Diag}\,(\boldsymbol{A}) \right\|_F \right],$$

for any $1 \leq q \leq \frac{\|\boldsymbol{B}\|_F}{C \sqrt{2 \max_i \sum_{j=1}^{T} B_{ij}^2}}$. $\qquad \square$

### A.1 Proof of Theorem 1

In this section we prove our main lower bound.

**Theorem 1.** *Let the number of rounds $T \geq T_0$, where $T_0$ is a universal constant. Let $\mathcal{A}$ be an online learning algorithm, and suppose $\mathcal{A}$ guarantees $R_T(0) \leq G\epsilon_T$ for any sequence of linear losses $g_1, \ldots, g_T \in \mathbb{R}$ satisfying $|g_t| \leq G$. Let $\boldsymbol{M}^{-1} \in \mathbb{R}^{T \times T}$ be any symmetric positive definite matrix, denote $\widetilde{\boldsymbol{M}}^{-1} := \boldsymbol{M}^{-1} - \mathrm{Diag}\,(\boldsymbol{M}^{-1})$ and $V_T := \mathrm{Tr}(\boldsymbol{M}^{-1}) + \left\| \widetilde{\boldsymbol{M}}^{-1} \right\|_F$. Suppose that $\left\| \widetilde{\boldsymbol{M}}^{-1} \right\|_F^2 \geq \frac{T}{2} \max_i \sum_j (\widetilde{M}_{ij}^{-1})^2$. Then, for any $P$ satisfying $T_0 \leq \log_2 \frac{\sqrt{P V_T}}{2\epsilon_T} \leq T$, there is a sequence of losses $g_1, \ldots, g_T \in \mathbb{R}$, and $\widetilde{\boldsymbol{u}} = (u_1, \ldots, u_T)^\top \in \mathbb{R}^T$ satisfying $\|\widetilde{\boldsymbol{u}}\|_{\boldsymbol{M}} = \sqrt{P}$ such that we have*

$$R_T(u_1, \ldots, u_T) \geq \Omega \left( G\epsilon_T + G \sqrt{P \left[ \mathrm{Tr}(\boldsymbol{M}^{-1}) + \left\| \widetilde{\boldsymbol{M}}^{-1} \right\|_F \log_2^{\frac{1}{2}} \frac{\sqrt{P V_T}}{2\epsilon_T} \right]} \right).$$

*Proof.* Denote $\boldsymbol{A} = \boldsymbol{M}^{-1}$ and $\boldsymbol{B} = \boldsymbol{A} - \mathrm{Diag}\,(\boldsymbol{A})$. By Lemma 2, there is a universal constant $C$ and a sequence $g_1, \ldots, g_T \in \mathbb{R}$ such that for any $1 \leq q \leq \frac{\|\boldsymbol{B}\|_F}{C \sqrt{2 \max_i \sum_{j=1}^{T} B_{ij}^2}}$, it holds that

$$\left\| \sum_{t=1}^{T} \widetilde{\boldsymbol{g}}_t \right\|_{\boldsymbol{A}}^2 \geq G^2 \left[ \mathrm{Tr}(\boldsymbol{A}) + q \left\| \boldsymbol{A} - \mathrm{Diag}\,(\boldsymbol{A}) \right\|_F \right]$$

and

$$R_T(0) \geq G\epsilon_T \left[1 - 2^{4C^2 q^2}\right].$$

Hence, choosing comparator sequence $u_1, \ldots, u_T \in \mathbb{R}$ to satisfy and $\widetilde{u} = (u_1, \ldots, u_T)^\top = -\sqrt{P} \frac{A \sum_{t=1}^T \widetilde{g}_t}{\|\sum_{t=1}^T \widetilde{g}_t\|_A} \in \mathbb{R}^T$, we have $\|\widetilde{u}\|_{A^{-1}} = \|\widetilde{u}\|_M = \sqrt{P}$ and

$$
\begin{aligned}
R_T(u_1, \ldots, u_T) &= R_T(0) - \left\langle \sum_{t=1}^T \widetilde{g}_t, \widetilde{u} \right\rangle \\
&= G\sqrt{P} \left\| \sum_{t=1}^T \widetilde{g}_t \right\|_A + R_T(0) \\
&\geq G\sqrt{P\left[\mathrm{Tr}(A) + q\|A - \mathrm{Diag}(A)\|_F\right]} + R_T(0) \\
&\geq G\epsilon_T + G\sqrt{P\left[\mathrm{Tr}(A) + q\|A - \mathrm{Diag}(A)\|_F\right]} - G\epsilon_T 2^{4C^2 q^2}.
\end{aligned}
$$

Now, for $P$ satisfying $T_0 := 4C^2 \leq \log_2\left(\frac{\sqrt{P[\mathrm{Tr}(A) + \|B\|_F]}}{2\epsilon_T}\right) \leq T$ we may choose

$$q = \sqrt{\frac{\log_2\left(\frac{\sqrt{P[\mathrm{Tr}(A) + \|B\|_F]}}{2\epsilon_T}\right)}{4C^2}}.$$

Indeed, observe that this choice satisfies $1 \leq q \leq \frac{\|B\|_F}{C\sqrt{2\max_i \sum_{j=1}^T B_{ij}^2}}$ as required:

$$1 \leq q = \sqrt{\frac{\log_2\left(\frac{\sqrt{P[\mathrm{Tr}(A) + \|B\|_F]}}{2\epsilon_T}\right)}{4C^2}} \leq \sqrt{\frac{T}{4C^2}} \leq \frac{\|B\|_F}{C\sqrt{2\max_i \sum_{j=1}^T B_{ij}^2}},$$

where the final inequality uses the assumption $\|B\|_F^2 / 2\max_i \sum_{ij} B_{ij}^2 \geq \frac{T}{4}$. Hence, we have that

$$G\epsilon_T 2^{4C^2 q^2} \leq \frac{G}{2}\sqrt{P\left[\mathrm{Tr}(A) + \|B\|_F\right]} \leq \frac{G}{2}\sqrt{P\left[\mathrm{Tr}(A) + q\|B\|_F\right]},$$

so that the overall the regret can be lower-bounded as

$$
\begin{aligned}
R_T(u_1, \ldots, u_T) &\geq G\epsilon_T + \frac{1}{2}G\sqrt{P\left[\mathrm{Tr}(A) + q\|B\|_F\right]} \\
&= G\epsilon_T + \frac{G}{2}\sqrt{P\left[\mathrm{Tr}(A) + \|B\|_F \frac{\log^{\frac{1}{2}}\left(\sqrt{P[\mathrm{Tr}(A) + \|B\|_F]}/2\epsilon_T\right)}{\sqrt{T_0}}\right]}.
\end{aligned}
$$

$\square$

## A.2   Proof of Proposition 2

**Proposition 2.** *(Adapting to Squared Path-length Requires Superlinear Regret) Define the finite-difference operator $\Sigma \in \mathbb{R}^T$ as the matrix with entries*

$$\Sigma_{ij} = \begin{cases} 1 & \text{if } i = j \\ -1 & \text{if } i = j - 1 \\ 0 & \text{otherwise} \end{cases}.$$

*Let $\mathbf{S} = \Sigma^\top \Sigma$ and $M = \mathbf{S} \otimes I_d$. Then, $M$ satisfies the assumptions of Theorem 1 and*

$$\|\widetilde{u}\|_M^2 = \|u_T\|_2^2 + \sum_{t=1}^{T-1} \|u_t - u_{t+1}\|_2^2 \qquad \text{and} \qquad \mathrm{Tr}\left(M^{-1}\right) = \frac{T(T+1)}{2}.$$

*Proof.* We first show the properties that $\|\widetilde{\boldsymbol{u}}\|_F^2 = \|\boldsymbol{u}_T\|_2^2 + \sum_{t=1}^{T-1} \|\boldsymbol{u}_t - \boldsymbol{u}_{t+1}\|_2^2$ and $\mathrm{Tr}(\boldsymbol{\Sigma}^{-1}\boldsymbol{\Sigma}^{-\top}) = \sum_{t=1}^{T} \left[\boldsymbol{\Sigma}^{-1}\boldsymbol{\Sigma}^{-\top}\right]_{tt} = \sum_{t=1}^{T} T - t + 1 = \frac{T(T+1)}{2}$, and show that $\boldsymbol{M}$ satisfies the conditions of Theorem 1 at the end.

Observe that

$$(\boldsymbol{\Sigma} \otimes \boldsymbol{I}_d)\widetilde{\boldsymbol{u}} = \begin{pmatrix} \boldsymbol{I}_d & -\boldsymbol{I}_d & \boldsymbol{0} & \boldsymbol{0} & \cdots \\ \boldsymbol{0} & \boldsymbol{I}_d & -\boldsymbol{I}_d & \boldsymbol{0} & \cdots \\ \boldsymbol{0} & \boldsymbol{0} & \boldsymbol{I}_d & -\boldsymbol{I}_d & \cdots \\ \vdots & & & & \ddots \\ \boldsymbol{0} & \boldsymbol{0} & \boldsymbol{0} & \cdots & \boldsymbol{I}_d \end{pmatrix} \begin{pmatrix} \boldsymbol{u}_1 \\ \vdots \\ \boldsymbol{u}_T \end{pmatrix} = \begin{pmatrix} \boldsymbol{u}_1 - \boldsymbol{u}_2 \\ \boldsymbol{u}_2 - \boldsymbol{u}_3 \\ \vdots \\ \boldsymbol{u}_{T-1} - \boldsymbol{u}_T \\ \boldsymbol{u}_T \end{pmatrix},$$

and since $(\boldsymbol{\Sigma}^\top \otimes \boldsymbol{I}_d)(\boldsymbol{\Sigma} \otimes \boldsymbol{I}_d) = (\boldsymbol{\Sigma}^\top\boldsymbol{\Sigma}) \otimes \boldsymbol{I}_d = \boldsymbol{M}$, we have

$$\|\widetilde{\boldsymbol{u}}\|_{\boldsymbol{M}}^2 = \left\langle \widetilde{\boldsymbol{u}}, (\boldsymbol{\Sigma}^\top \otimes \boldsymbol{I}_d)(\boldsymbol{\Sigma} \otimes \boldsymbol{I}_d)\widetilde{\boldsymbol{u}} \right\rangle = \langle (\boldsymbol{\Sigma} \otimes \boldsymbol{I}_d)\widetilde{\boldsymbol{u}}, (\boldsymbol{\Sigma} \otimes \boldsymbol{I}_d)\widetilde{\boldsymbol{u}} \rangle$$

$$= \|\boldsymbol{u}_T\|_2^2 + \sum_{t=1}^{T-1} \|\boldsymbol{u}_t - \boldsymbol{u}_{t+1}\|_2^2 .$$

Using the inverse property of the Kronecker product, we also have

$$\boldsymbol{M}^{-1} = \left[\boldsymbol{\Sigma}^\top\boldsymbol{\Sigma} \otimes \boldsymbol{I}_d\right]^{-1} = \left[\boldsymbol{\Sigma}^\top\boldsymbol{\Sigma}\right]^{-1} \otimes \boldsymbol{I}_d = \boldsymbol{\Sigma}^{-1}\boldsymbol{\Sigma}^{-\top} \otimes \boldsymbol{I}_d,$$

and by Lemma 8 we have that $\boldsymbol{\Sigma}^{-1}$ is the upper-triangular matrix of all 1's, that is, the matrix with entries

$$\Sigma_{ij}^{-1} = \begin{cases} 1 & \text{if } i \leq j \\ 0 & \text{otherwise} \end{cases},$$

and likewise, $\boldsymbol{\Sigma}^{-\top}$ is a lower-triangular matrix of $1's$. In other words, for any $t$ we have

$$\left[\boldsymbol{\Sigma}^{-1}\boldsymbol{\Sigma}^{-\top}\right]_{tt} = \sum_{i=1}^{T} \Sigma_{ti}^{-1}\Sigma_{it}^{-\top} = \sum_{i \leq t} \Sigma_{ti}^{-1} = T - t + 1 .$$

So, summing over $t$ we have

$$\mathrm{Tr}(\boldsymbol{\Sigma}^{-1}\boldsymbol{\Sigma}^{-\top}) = \sum_{t=1}^{T} \left[\boldsymbol{\Sigma}^{-1}\boldsymbol{\Sigma}^{-\top}\right]_{tt} = \sum_{t=1}^{T} T - t + 1 = \frac{T(T+1)}{2} .$$

Now we show that $\boldsymbol{M}$ satisfies the conditions of Theorem 1. $\boldsymbol{M} = \mathbf{S} \otimes \boldsymbol{I}_d = [\boldsymbol{\Sigma}^\top\boldsymbol{\Sigma}] \otimes \boldsymbol{I}_d$ is clearly symmetric since it is the Kronecker product of two symmetric matrices. Observe that for any $\boldsymbol{x} \neq \boldsymbol{0} \in \mathbb{R}^T$ we have $\boldsymbol{\Sigma}\boldsymbol{x} \neq \boldsymbol{0}$ by positive definiteness of $\boldsymbol{\Sigma}$ (Lemma 8) and thus $\langle \boldsymbol{x}, \mathbf{S}\boldsymbol{x} \rangle = \langle \boldsymbol{\Sigma}\boldsymbol{x}, \boldsymbol{\Sigma}\boldsymbol{x} \rangle > 0$. Thus, $\boldsymbol{M} = \mathbf{S} \otimes \boldsymbol{I}_d$ is the Kronecker product of symmetric positive definite matrices, so $\boldsymbol{M}$ is symmetric positive definite [see, *e.g.*, 38, Chapter 2].

Lastly, let $\boldsymbol{B} = \boldsymbol{\Sigma}^{-1}\boldsymbol{\Sigma}^{-\top} - \mathrm{Diag}\left(\boldsymbol{\Sigma}^{-1}\boldsymbol{\Sigma}^{-\top}\right)$. We are to show that $\|\boldsymbol{B}\|_F \geq \frac{T}{2} \sum_j B_{ij}^2$ for any $i$. First observe that calculation of $[\boldsymbol{\Sigma}^{-1}\boldsymbol{\Sigma}^{-\top}]_{tt}$ is generalized to

$$\left[\boldsymbol{\Sigma}^{-1}\boldsymbol{\Sigma}^{-\top}\right]_{ij} = \sum_{k=1}^{T} \Sigma_{ik}^{-1}\Sigma_{kj}^{-\top} = \sum_{k=1}^{T} \Sigma_{ki}^{-\top}\Sigma_{kj}^{-\top} = \sum_{k=1}^{j \wedge i} 1 = T - \max\{j, i\} + 1,$$

for any $i, j$, and likewise $\boldsymbol{B}_{ij} = T - \max\{j, i\} + 1$ for $i \neq j$ and 0 otherwise, from which it is easily seen that $\max_i \sum_j B_{ij}^2 = \sum_j B_{1j}^2$, so for any $i$ we have

$$\sum_j B_{ij}^2 \leq \sum_j B_{1j}^2 = \sum_{j=2}^{T} (T - j + 1)^2 = \frac{1}{6}T(2T^2 - 3T + 1) .$$

On the other hand,

$$\|\boldsymbol{B}\|_F^2 = \sum_i \sum_j B_{ij}^2 = \frac{1}{6}T^2(T^2-1)$$

$$= \frac{T}{2}\frac{T}{6}(2T^2-2) = \frac{T}{2}\frac{T}{6}(2T^2-3T+3T-2) \geq \frac{T}{2}\frac{T}{6}(2T^2-3T+1)$$

$$\geq \frac{T}{2}\sum_j B_{ij}^2,$$

for any $i$, where the last line applies the inequality in the previous display. $\qquad\square$

## A.3 Sufficiency of Proposition 2

The choice of $\boldsymbol{M}$ in Proposition 2 uniquely exposes the squared path-length up to the constant offset term $\|\boldsymbol{u}_T\|^2$. In this section we demonstrate that the choice of offset term in Proposition 2 does not make any significant difference for the claim that adapting to the squared path-length requires incurring a $\mathrm{Tr}(\boldsymbol{M}^{-1}) \geq \Omega(T^2)$ penalty, and hence that Proposition 2 is sufficient to demonstrate that adapting to the squared path-length is not possible without incurring vacuous regret.

To expedite the discussion, we first introduce two technical lemmas, proven in Appendices A.3.1 and A.3.2 respectively.

**Lemma 3.** *Let $\boldsymbol{v} \in \mathbb{R}^T$ be an arbitrary non-zero vector and let $\boldsymbol{B} \in \mathbb{R}^{T\times T}$ be a symmetric matrix with eigenvalues $0 = \lambda_1(\boldsymbol{B}) < \lambda_2(\boldsymbol{B}) \leq \ldots \leq \lambda_T(\boldsymbol{B})$. Then*

$$\mathrm{Tr}((\boldsymbol{B}+\boldsymbol{v}\boldsymbol{v}^\top)^{-1}) \geq \|\boldsymbol{v}\|^2 + \sum_{t=2}^T \frac{1}{\lambda_t(\boldsymbol{B})}\,.$$

**Lemma 4.** *Let $\boldsymbol{\Sigma} \in \mathbb{R}^{T\times T}$ denote the finite-difference matrix defined in Proposition 2 and let $\boldsymbol{M} = \boldsymbol{\Sigma}^\top\boldsymbol{\Sigma}$. Then, for any $T > 1$, we have*

$$\lambda_{\max}(\boldsymbol{M}^{-1}) \leq \frac{9}{10}\,\mathrm{Tr}(\boldsymbol{M}^{-1}),$$

*where $\lambda_{\max}(\boldsymbol{M}^{-1})$ is the maximal eigenvalue of $\boldsymbol{M}^{-1}$.*

Now, Consider the 1-dimensional setting and note that for any positive definite $\boldsymbol{M}$ we can find a unique $\boldsymbol{\Sigma}$ such that $\boldsymbol{M} = \boldsymbol{\Sigma}^\top\boldsymbol{\Sigma}$. Hence,

$$\|\widetilde{\boldsymbol{u}}\|_{\boldsymbol{M}}^2 = \langle \widetilde{\boldsymbol{u}}, \boldsymbol{M}\widetilde{\boldsymbol{u}}\rangle = \langle \boldsymbol{\Sigma}\widetilde{\boldsymbol{u}}, \boldsymbol{\Sigma}\widetilde{\boldsymbol{u}}\rangle,$$

so without loss of generality we can focus on $\boldsymbol{\Sigma}$ for which

$$\langle \boldsymbol{\Sigma}\widetilde{\boldsymbol{u}}, \boldsymbol{\Sigma}\widetilde{\boldsymbol{u}}\rangle = \langle \boldsymbol{v}, \widetilde{\boldsymbol{u}}\rangle^2 + \sum_{t=2}^T \|u_t - u_{t-1}\|^2,$$

where $\boldsymbol{v} \neq \boldsymbol{0} \in \mathbb{R}^T$. [4] Note such a constant offset term is unavoidable: it is what captures the static regret guarantee in the case where $u_1 = \ldots = u_T = u$. Proposition 2 considers $\boldsymbol{v} = (0,\ldots,0,1)$ to get $\|\widetilde{\boldsymbol{u}}\|_{\boldsymbol{M}}^2 = \|u_T\|^2 + \sum_{t=2}^T \|u_t - u_{t-1}\|^2$, though below we will show that any vector $\boldsymbol{v}$ would still lead to $\mathrm{Tr}(\boldsymbol{M}^{-1}) = \Omega(T^2)$.

It is clear that the only way to construct expressions of the form above is via matrices $\boldsymbol{\Sigma}$ satisfying

$$\boldsymbol{\Sigma}\widetilde{\boldsymbol{u}} = c \begin{pmatrix} u_1 - u_2 \\ u_2 - u_3 \\ \vdots \\ u_{T-1} - u_T \\ \langle \boldsymbol{v}, \widetilde{\boldsymbol{u}}\rangle \end{pmatrix},$$

---

[4]Note that any such $\boldsymbol{M}$ is unique. Indeed, if there are positive definite matrices $\boldsymbol{M}_1 \in \mathbb{R}^{T\times T}$ and $\boldsymbol{M}_2 \in \mathbb{R}^{T\times T}$ such that $\|\widetilde{\boldsymbol{u}}\|_{\boldsymbol{M}_1}^2 = \langle \boldsymbol{v}, \widetilde{\boldsymbol{u}}\rangle^2 + P_T^{\|\cdot\|_2^2} = \|\widetilde{\boldsymbol{u}}\|_{\boldsymbol{M}_2}^2$ for all $\widetilde{\boldsymbol{u}} \in \mathbb{R}^T$, then $\langle \widetilde{\boldsymbol{u}}, (\boldsymbol{M}_1 - \boldsymbol{M}_2)\widetilde{\boldsymbol{u}}\rangle = 0$ and hence $\boldsymbol{M}_1 = \boldsymbol{M}_2$ since $\boldsymbol{M}_1$ and $\boldsymbol{M}_2$ are positive definite.

where $c \in \{-1, 1\}$ and the order of the rows indices of the vector can be permuted without loss of generality. In particular, the only matrices that can produce these expressions (again noting that the rows can be permuted without loss of generality) are of the form

$$\boldsymbol{\Sigma} = c \begin{pmatrix} 1 & -1 & 0 & 0 & \dots & 0 & 0 \\ 0 & 1 & -1 & 0 & \dots & 0 & 0 \\ 0 & 0 & 1 & -1 & \dots & 0 & 0 \\ \vdots & & & & \ddots & & \\ 0 & 0 & 0 & 0 & \dots & 1 & -1 \\ v_1 & v_2 & v_3 & v_4 & \dots & v_{T-1} & v_T \end{pmatrix} =: c \begin{pmatrix} \boldsymbol{\Delta} \\ \boldsymbol{v}^\top \end{pmatrix},$$

so $\boldsymbol{M} = \boldsymbol{\Sigma}^\top \boldsymbol{\Sigma} = \boldsymbol{\Delta}^\top \boldsymbol{\Delta} + \boldsymbol{v} \boldsymbol{v}^\top$. Moreover, $\boldsymbol{\Delta}^\top \boldsymbol{\Delta}$ is a symmetric matrix with a unique zero eigenvalue (corresponding to vectors in the span of $\boldsymbol{1} = (1, \dots, 1) \in \mathbb{R}^T$), so applying Lemma 3,

$$\mathrm{Tr}(\boldsymbol{M}^{-1}) = \mathrm{Tr}((\boldsymbol{\Delta}^\top \boldsymbol{\Delta} + \boldsymbol{v}\boldsymbol{v}^\top)^{-1}) \geq \|\boldsymbol{v}\|^2 + \sum_{t=2}^{T} \frac{1}{\lambda_t(\boldsymbol{\Delta}^\top \boldsymbol{\Delta})}.$$

Now, define $\boldsymbol{v}_0 = (0, \dots, 0, 1) \in \mathbb{R}^T$ and observe that $\boldsymbol{M}_0 := \boldsymbol{\Delta}^\top \boldsymbol{\Delta} + \boldsymbol{v}_0 \boldsymbol{v}_0^\top$ is precisely the matrix studied in Proposition 2. We have via the interlacing property of rank-1 updates to symmetric matrices that $\lambda_t(\boldsymbol{\Delta}^\top \boldsymbol{\Delta}) \leq \lambda_t(\boldsymbol{M}_0)$ [13, Theorem 8.1.8], so overall we have

$$\begin{aligned} \mathrm{Tr}(\boldsymbol{M}^{-1}) &\geq \sum_{t=2}^{T} \frac{1}{\lambda_t(\boldsymbol{M}_0)} = \sum_{t=2}^{T} \lambda_t(\boldsymbol{M}_0^{-1}) \\ &= \mathrm{Tr}(\boldsymbol{M}_0^{-1}) - \lambda_{\max}(\boldsymbol{M}_0^{-1}) \\ &\geq \mathrm{Tr}(\boldsymbol{M}_0^{-1}) - \frac{9}{10} \mathrm{Tr}(\boldsymbol{M}_0^{-1}) \\ &= \frac{1}{10} \mathrm{Tr}(\boldsymbol{M}_0^{-1}) = \frac{1}{10} \frac{T(T+1)}{2} \end{aligned}$$

where the last inequality applies Lemma 4 to bound $\lambda_{\max}(\boldsymbol{M}_0^{-1})$ and recalls $\mathrm{Tr}(\boldsymbol{M}_0^{-1}) = \frac{T(T+1)}{2}$ from Proposition 2.

Hence, the variance penalty will still be $\Omega(T^2)$ regardless of the choice of bias $\langle \boldsymbol{v}, \widetilde{\boldsymbol{u}} \rangle^2$ in the variability measure. Combined with our lower bound in Theorem 1, it follows that adapting to the squared path-length necessarily implies a variance penalty of $\mathrm{Tr}(\boldsymbol{M}^{-1}) \geq \Omega(T^2)$, leading to a vacuous regret upper bound.

### A.3.1 Proof of Lemma 3

**Lemma 3.** *Let $\boldsymbol{v} \in \mathbb{R}^T$ be an arbitrary non-zero vector and let $\boldsymbol{B} \in \mathbb{R}^{T \times T}$ be a symmetric matrix with eigenvalues $0 = \lambda_1(\boldsymbol{B}) < \lambda_2(\boldsymbol{B}) \leq \dots \leq \lambda_T(\boldsymbol{B})$. Then*

$$\mathrm{Tr}((\boldsymbol{B} + \boldsymbol{v}\boldsymbol{v}^\top)^{-1}) \geq \|\boldsymbol{v}\|^2 + \sum_{t=2}^{T} \frac{1}{\lambda_t(\boldsymbol{B})}.$$

*Proof.* Let $\boldsymbol{A} := \boldsymbol{B} + \boldsymbol{v}\boldsymbol{v}^\top$. Since $\boldsymbol{B}$ is symmetric, we have via the interlacing property that there is an $a_1, \dots, a_T \geq 0$ such that $\sum_{t=1}^{T} a_t = \|\boldsymbol{v}\|^2$ and $\lambda_t(\boldsymbol{A}) = \lambda_t(\boldsymbol{B}) + a_i$ [see, *e.g.*, Theorem 8.1.8 in 13]. Hence,

$$\begin{aligned} \mathrm{Tr}((\boldsymbol{B} + \boldsymbol{v}\boldsymbol{v}^\top)^{-1}) &= \mathrm{Tr}(\boldsymbol{A}^{-1}) = \sum_{t=1}^{T} \lambda_t(\boldsymbol{A}^{-1}) = \sum_{t=1}^{T} \frac{1}{\lambda_t(\boldsymbol{A})} \\ &= \sum_{t=1}^{T} \frac{1}{\lambda_t(\boldsymbol{B}) + a_i} \geq \min_{\substack{a_1, \dots, a_T \geq 0 \\ \sum_{t=1}^{T} a_t = \|\boldsymbol{v}\|^2}} \sum_{t=1}^{T} \frac{1}{\lambda_t(\boldsymbol{B}) + a_i}. \end{aligned}$$

To analyze the constrained optimization in the last line, let $\alpha_1, \ldots, \alpha_T \geq 0$, $\beta \in \mathbb{R}$, and define the Lagrangian

$$L(a_1, \ldots, a_T, \alpha_1, \ldots, \alpha_T, \beta) = \sum_{t=1}^{T} \frac{1}{\lambda_t(\boldsymbol{B}) + a_t} - \sum_{t=1}^{T} \alpha_t a_t + \beta \left( \sum_{t=1}^{T} a_t - \|\boldsymbol{v}\|^2 \right).$$

For any $t$, we have

$$\frac{\partial L}{\partial a_t} = \frac{-1}{(\lambda_t(\boldsymbol{B}) + a_t)^2} - \alpha_t + \beta = 0 \iff a_t = \frac{1}{\sqrt{\beta - \alpha_t}} - \lambda_t(\boldsymbol{B}).$$

Plugging this into the dual $D(\alpha_1, \ldots, \alpha_T, \beta) = \min_{a_1, \ldots, a_T} L(a_1, \ldots, a_T, \alpha_1, \ldots, \alpha_T, \beta)$ we have

$$\begin{aligned}
D(\alpha_1, \ldots, \alpha_T, \beta) &= \sum_{t=1}^{T} \sqrt{\beta - \alpha_t} - \sum_{t=1}^{T} \alpha_t \left( \frac{1}{\sqrt{\beta - \alpha_t}} - \lambda_t(\boldsymbol{B}) \right) + \beta \left( \sum_{t=1}^{T} \frac{1}{\sqrt{\beta - \alpha_t}} - \lambda_t(\boldsymbol{B}) - \|\boldsymbol{v}\|^2 \right) \\
&= \sum_{t=1}^{T} \sqrt{\beta - \alpha_t} + \sum_{t=1}^{T} (\beta - \alpha_t) \frac{1}{\sqrt{\beta - \alpha_t}} + \sum_{t=1}^{T} (\alpha_t - \beta) \lambda_t(\boldsymbol{B}) - \beta \|\boldsymbol{v}\|^2 \\
&= 2 \sum_{t=1}^{T} \sqrt{\beta - \alpha_t} + \sum_{t=1}^{T} (\alpha_t - \beta) \lambda_t(\boldsymbol{B}) - \beta \|\boldsymbol{v}\|^2 .
\end{aligned}$$

The derivatives of the dual *w.r.t* $\alpha_t$ are

$$\frac{\partial D}{\partial \alpha_t} = \frac{-1}{\sqrt{\beta - \alpha_t}} + \lambda_t(\boldsymbol{B}).$$

Observe that for $\lambda_1(\boldsymbol{B}) = 0$, we have $\frac{\partial D}{\partial \alpha_t} = -\frac{1}{\sqrt{\beta - \alpha_t}} \leq 0$, so $D$ is decreasing in $\alpha_1$, so the dual is maximized when $\alpha_1 = 0$. Using the relation $a_1 = \frac{1}{\sqrt{\beta - \alpha_1}} - \lambda_1(\boldsymbol{B})$ above we have $a_1 = \frac{1}{\sqrt{\beta - \alpha_1}} - \lambda_1(\boldsymbol{B}) = \frac{1}{\sqrt{\beta}}$. Equating the other derivatives for $t > 1$ to zero we have

$$\frac{1}{\sqrt{\beta - \alpha_t}} = \lambda_t(\boldsymbol{B}) \implies \lambda_t(\boldsymbol{B}) + a_t = \lambda_t(\boldsymbol{B})$$
$$\implies a_t = 0 \quad \forall t > 1$$

where we used the relationship $a_t = \frac{1}{\sqrt{\beta - \alpha_t}} - \lambda_t(\boldsymbol{B})$ from above. Finally, the optimal $\beta$ is such that $\sum_{t=1}^{T} a_t = \frac{1}{\sqrt{\beta}} = \|\boldsymbol{v}\|^2$, so overall we have

$$\min_{\substack{a_1, \ldots, a_T \geq 0 \\ \sum_{t=1}^{T} a_t = \|\boldsymbol{v}\|^2}} \sum_{t=1}^{T} \frac{1}{\lambda(\boldsymbol{B}) + a_i} = \frac{1}{\sqrt{\beta - \alpha_1}} + \sum_{t=2}^{T} \frac{1}{\lambda_t(\boldsymbol{B}) + a_i} = \frac{1}{\sqrt{\beta}} + \sum_{t=2}^{T} \frac{1}{\lambda_t(\boldsymbol{B})}$$

$$= \|\boldsymbol{v}\|^2 + \sum_{t=2}^{T} \frac{1}{\lambda_t(\boldsymbol{B})} . \qquad \square$$

### A.3.2 Proof of Lemma 4

**Lemma 4.** *Let $\boldsymbol{\Sigma} \in \mathbb{R}^{T \times T}$ denote the finite-difference matrix defined in Proposition 2 and let $\boldsymbol{M} = \boldsymbol{\Sigma}^\top \boldsymbol{\Sigma}$. Then, for any $T > 1$, we have*

$$\lambda_{\max}(\boldsymbol{M}^{-1}) \leq \frac{9}{10} \operatorname{Tr}(\boldsymbol{M}^{-1}),$$

*where $\lambda_{\max}(\boldsymbol{M}^{-1})$ is the maximal eigenvalue of $\boldsymbol{M}^{-1}$.*

*Proof.* The matrix $\boldsymbol{M}^{-1} = \boldsymbol{\Sigma}^{-1} \boldsymbol{\Sigma}^{-\top}$ is symmetric and positive definite, hence has real eigenvalues. The eigenvalues of $\boldsymbol{M}^{-1}$ can be bound in terms of its trace as follows (see, *e.g.*, Theorem 2.1 Wolkowicz and Styan [43], provided for convenience in Theorem 5):

$$\lambda_{\max}(\boldsymbol{M}^{-1}) \leq \frac{\operatorname{Tr}(\boldsymbol{M}^{-1})}{T} + \sqrt{(T-1) \left[ \frac{\operatorname{Tr}(\boldsymbol{M}^{-\top} \boldsymbol{M}^{-1})}{T} - \left( \frac{\operatorname{Tr}(\boldsymbol{M}^{-1})}{T} \right)^2 \right]} .$$

Next, observe that by Lemma 8, matrix $\mathbf{\Sigma}^{-1}$ is an upper-triangular matrix of all 1's, so that

$$[\boldsymbol{M}^{-1}]_{ij} = \left[\mathbf{\Sigma}^{-1}\mathbf{\Sigma}^{-\top}\right]_{ij} = \sum_{k\in[t]}\Sigma_{ik}^{-1}\Sigma_{kj}^{-\top} = \sum_{k\in[T]}\Sigma_{ik}^{-1}\Sigma_{jk}^{-1}$$

$$= \sum_{k\in[T]}\mathbf{1}\left\{k\geq i\right\}\mathbf{1}\left\{k\geq j\right\} = T - \max\left\{i,j\right\} + 1,$$

Hence,

$$\operatorname{Tr}\left(\boldsymbol{M}^{-1}\right) = \sum_{t=1}^{T}[\boldsymbol{M}^{-1}]_{ii} = \sum_{t=1}^{T}(T-t+1) = \sum_{t=1}^{T}t = \frac{T(T+1)}{2}.$$

Moreover,

$$\operatorname{Tr}\left(\boldsymbol{M}^{-\top}\boldsymbol{M}^{-1}\right) = \sum_{t=1}^{T}[\boldsymbol{M}^{-\top}\boldsymbol{M}^{-1}]_{tt} = \sum_{t=1}^{T}\sum_{k=1}^{t}M_{tk}^{-\top}M_{kt}^{-1}$$

$$= \sum_{t=1}^{T}\sum_{k=1}^{t}(M_{kt}^{-1})^2 = \sum_{t=1}^{T}\sum_{k=1}^{t}(T-\max\left\{t,k\right\}+1)^2$$

$$= \frac{T(T+1)^2(T+2)}{12}.$$

Thus,

$$\left[\frac{\operatorname{Tr}\left(\boldsymbol{M}^{-\top}\boldsymbol{M}^{-1}\right)}{T} - \left(\frac{\operatorname{Tr}\left(\boldsymbol{M}^{-1}\right)}{T}\right)^2\right] = \frac{(T+1)^2(T+2)}{12} - \frac{(T+1)^2}{4}$$

$$= \frac{(T+1)^2}{4}\left[\frac{T+2}{3}-1\right]$$

$$= \frac{(T+1)^2}{4}\frac{T-1}{3}.$$

Overall, $\lambda_{\max}(\boldsymbol{M}^{-1})$ is bounded by

$$\lambda_{\max}(\boldsymbol{M}^{-1}) \leq \frac{\operatorname{Tr}\left(\boldsymbol{M}^{-1}\right)}{T} + \sqrt{(T-1)\frac{(T+1)^2}{4}\frac{T-1}{3}}$$

$$= \frac{T+1}{2} + \frac{(T+1)(T-1)}{2\sqrt{3}}$$

$$= \frac{T(T+1)}{2\sqrt{3}} + \frac{T+1}{2}\left[1-\frac{1}{\sqrt{3}}\right]$$

$$\leq \frac{T(T+1)}{2\sqrt{3}} + \frac{T(T+1)}{2}\frac{1}{4} \leq \frac{9}{10}\frac{T(T+1)}{2} = \frac{9}{10}\operatorname{Tr}(\boldsymbol{M}^{-1}),$$

where the last line observes that $1-\frac{1}{\sqrt{3}} \leq \frac{1}{2} \leq \frac{T}{4}$ for $T\geq 2$ and the fact that $\frac{1}{\sqrt{3}}+\frac{1}{4} \approx 0.83 \leq \frac{9}{10}$. $\qquad\square$

## B  Proofs for Section 4 (Dynamic regret for unconstrained OLO via weighted norms)

### B.1  Details on the 1-Dimensional Reduction

In this section, for completeness we provide the details of the 1-dimensional reduction of Cutkosky and Orabona [9], specialized to dual weighted-norm pairs $(\|\cdot\|_{\boldsymbol{M}}, \|\cdot\|_{\boldsymbol{M}^{-1}})$ as well as its regret guarantee.

For concreteness, we choose adaptive FTRL with AdaGrad-norm stepsizes [40] as the direction learner. For simplicity we use the scale-free version of [33], so that the direction learner's update is slightly simpler, not requiring prior knowledge of the Lipschitz constant $\mathfrak{G} \geq \|\widetilde{g}_t\|_{M^{-1}}$.

Using Cutkosky and Orabona [9, Theorem 2], we have that the regret of Algorithm 2 is equal to

$$R_T(\widetilde{u}) = R_T^{\mathcal{A}}(\|\widetilde{u}\|_M) + \|u\|_M R_T^{\text{direction}}\left(\frac{\widetilde{u}}{\|\widetilde{u}\|_M}\right), \quad \forall \widetilde{u} \in \mathbb{R}^{dT},$$

where $R_T^{\mathcal{A}}$ is the regret of $\mathcal{A}$ over a sequence of $G$-Lipschitz linear losses and $R_T^{\text{direction}}$ is the regret of (scale-free) adaptive FTRL with a feasible set equal to the unitary ball defined by $\|\cdot\|_M$.

Choosing the algorithm $\mathcal{A}$ to be [20, Algorithm 1], we have

$$R_T^{\mathcal{A}}(\|\widetilde{u}\|_M) \leq \mathcal{O}\left(\mathfrak{G}\epsilon + \|\widetilde{u}\|_M\left[\sqrt{V_T \log\left(\frac{\|\widetilde{u}\|_M \sqrt{V_T}\Lambda_T}{\mathfrak{G}\epsilon} + 1\right)} \vee \mathfrak{G}\log\left(\frac{\|\widetilde{u}\|_M \sqrt{V_T}\Lambda_T}{\epsilon\mathfrak{G}}\right)\right]\right),$$

where $V_T = \sum_{t=1}^T \|\widetilde{g}_t\|_{M^{-1}}^2$ and $\Lambda_T = \log^2(\sum_{t=1}^T \|\widetilde{g}_t\|_{M^{-1}}^2 / \mathfrak{G}^2) \leq \mathcal{O}(\log^2 T)$.

Focusing now on the regret of the direction learner, define the distance generating function $\psi(\widetilde{x}) = \frac{1}{2}\|\widetilde{x}^2\|_M$. Using [31, Theorem 4.3], we have that $\psi$ is 1-strongly convex *w.r.t* $\|\cdot\|_M$. Hence, using the regret guarantee of Scale-free FTRL, *i.e.*, Theorem 1 of Orabona and Pál [33], for any $\widetilde{v} \in \mathbb{R}^{dT}$ such that $\|\widetilde{v}\|_M \leq 1$ the regret of the direction learner is

$$R_T^{\text{Direction}}(\widetilde{v}) \leq \left[\frac{1}{2}\|\widetilde{v}\|_M^2 + 2.75\right]\sqrt{\sum_{t=1}^T \|\widetilde{g}_t\|_{M^{-1}}^2} + 3.5\max_{t\leq T}\|\widetilde{g}_t\|_{M^{-1}} \leq \mathcal{O}\left(\sqrt{\sum_{t=1}^T \|\widetilde{g}_t\|_{M^{-1}}^2}\right).$$

Applying this with $\widetilde{v} = \frac{\widetilde{u}}{\|\widetilde{u}\|_M}$ and combining with the previous two displays leads to the bound stated in the proof of Theorem 2.

## B.2 The Haar Matrices and their Properties

In this section we provide some useful supporting lemmas related to the Haar matrices $H_n$. We first introduce the *Haar basis vectors*, which make up the columns of the matrix $H_n$. Throughout this section we assume for simplicity that $T$ is a power of 2.

**Definition 1.** *For any $\tau \in \{2^i : i = 1 : \log_2(T)\}$ and $i \in [T/\tau]$, the Haar basis vector at timescale $\tau$ and location $i$ is the vector in $\mathbb{R}^T$ with entries*

$$[h_i^{(\tau)}]_t = \begin{cases} 1 & \text{if } t \in [\frac{1}{2}\tau(i-1)+1, \frac{1}{2}\tau i] \\ -1 & \text{if } t \in [\frac{1}{2}\tau i + 1, \tau i] \\ 0 & \text{otherwise} \end{cases} \tag{6}$$

The Haar basis vectors are often arranged into the columns of a matrix as follows:

$$\mathbf{H}_n = \begin{pmatrix} h_0 & h_1^{(T)} & h_1^{(T/2)} & h_2^{(T/2)} & h_1^{(T/4)} & h_2^{(T/4)} & h_3^{(T/4)} & h_4^{(T/4)} & \cdots & h_{T/2}^{(2)} \end{pmatrix},$$

where $h_0 = (1, 1, \ldots, 1)^\top \in \mathbb{R}^T$. This matrix is referred to as the (unnormalized) Haar basis matrix of order $n = \log_2(T)$. It is well-known that $H_n$ has the following equivalent recursive form [38, 11, 37]:

$$\mathbf{H}_0 = (1),$$

$$\mathbf{H}_n = \left(\mathbf{H}_{n-1} \otimes \begin{pmatrix} 1 \\ 1 \end{pmatrix} \quad \mathbf{I}_{2^{n-1}} \otimes \begin{pmatrix} 1 \\ -1 \end{pmatrix}\right). \tag{7}$$

So, for instance, we have

$$\mathbf{H}_1 = \left(\mathbf{H}_0 \otimes \begin{pmatrix} 1 \\ 1 \end{pmatrix} \quad \mathbf{I}_{2^0} \otimes \begin{pmatrix} 1 \\ -1 \end{pmatrix}\right) = \left((1) \otimes \begin{pmatrix} 1 \\ 1 \end{pmatrix} \quad (1) \otimes \begin{pmatrix} 1 \\ -1 \end{pmatrix}\right) = \begin{pmatrix} 1 & 1 \\ 1 & -1 \end{pmatrix},$$

$$\mathbf{H}_2 = \left(\begin{pmatrix} 1 & 1 \\ 1 & -1 \end{pmatrix} \otimes \begin{pmatrix} 1 \\ 1 \end{pmatrix} \quad \begin{pmatrix} 1 & 0 \\ 0 & 1 \end{pmatrix} \otimes \begin{pmatrix} 1 \\ -1 \end{pmatrix}\right) = \begin{pmatrix} 1 & 1 & 1 & 0 \\ 1 & 1 & -1 & 0 \\ 1 & -1 & 0 & 1 \\ 1 & -1 & 0 & -1 \end{pmatrix},$$

and so on. For our purposes, we will primaly work in terms of the matrices $\boldsymbol{H}_n$ rather than the basis vectors $\boldsymbol{h}_i^{(\tau)}$. The main utility of defining the basis vectors $\boldsymbol{h}_i^{(\tau)}$ is that their definition easily implies the following useful result, which states that the Haar basis vectors are *sparsely supported w.r.t* time.

**Proposition 5.** *Let $n = \log_2 T$ and let $\boldsymbol{H}_n \in \mathbb{R}^{T \times T}$ be the unnormalized Haar basis matrix of order $n$. Then for any $t \in [T]$, there at most $1 + \log T$ indices $i$ for which $[\boldsymbol{H}_n]_{t,i} \neq 0$.*

The proof follows immediately from Definition 1 (*i.e.*, any $t$ can fall into only one of the intervals covered at each of the $\log_2(T)$ time-scales) and accounting for the additional column $\boldsymbol{h}_0$ of all 1's.

In what follows, we will also use the following well-known relationship between the vec operator and the Kronecker product (see, *e.g.*, Steeb and Shi [38, Chapter 2.11]).

**Proposition 6.** *Let $\boldsymbol{A}$, $\boldsymbol{B}$, and $\boldsymbol{C}$ be matrices of appropriate dimensions such that the product $\boldsymbol{ABC}$ exists. Then, $vec(\boldsymbol{ABC}) = (\boldsymbol{C}^\top \otimes \boldsymbol{A})vec(\boldsymbol{B})$.*

The following three lemmas will be used to prove the guarantees of the algorithm characterized in Section 4.1 (Propositions 3 and 4).

**Lemma 5.** *Let $n = \log_2(T)$, $\boldsymbol{v} = (v_1, \ldots, v_T)^\top \in \mathbb{R}^T$, and let $\mathbf{H}_n$ be the unnormalized Haar basis matrix of order $n$. Then*

$$\mathbf{H}_n^T \boldsymbol{v} = \begin{pmatrix} \mathbf{H}_{n-1}^\top \boldsymbol{v}_+ \\ I_{2^{n-1}} \boldsymbol{v}_- \end{pmatrix},$$

*where*

$$\boldsymbol{v}_+ = \begin{pmatrix} v_1 + v_2 \\ v_3 + v_4 \\ \vdots \\ v_{T-1} + v_T \end{pmatrix}, \quad \boldsymbol{v}_- = \begin{pmatrix} v_1 - v_2 \\ v_3 - v_4 \\ \vdots \\ v_{T-1} - v_T \end{pmatrix}.$$

*Proof.* From Equation (7), we have that

$$\mathbf{H}_n^\top \boldsymbol{v} = \left( \mathbf{H}_{n-1} \otimes \begin{pmatrix} 1 \\ 1 \end{pmatrix} \quad I_{2^{n-1}} \otimes \begin{pmatrix} 1 \\ -1 \end{pmatrix} \right)^\top \boldsymbol{v} = \begin{pmatrix} \mathbf{H}_{n-1}^\top \otimes (1 \quad 1) \\ I_{2^{n-1}} \otimes (1 \quad -1) \end{pmatrix} \boldsymbol{v}$$

$$= \begin{pmatrix} \left[ \mathbf{H}_{n-1}^\top \otimes (1 \quad 1) \right] \boldsymbol{v} \\ \left[ I_{2^{n-1}} \otimes (1 \quad -1) \right] \boldsymbol{v} \end{pmatrix}.$$

Moreover, leveraging Proposition 6 we have

$$\mathbf{H}_n^\top \boldsymbol{v} = \begin{pmatrix} \text{vec} \left( (1 \quad 1) \begin{pmatrix} v_1 & v_3 & \cdots & v_{T-1} \\ v_2 & v_4 & \cdots & v_T \end{pmatrix} \mathbf{H}_{n-1} \right) \\ \text{vec} \left( (1 \quad -1) \begin{pmatrix} v_1 & v_3 & \cdots & v_{T-1} \\ v_2 & v_4 & \cdots & v_T \end{pmatrix} \mathbf{H}_{n-1} \right) \end{pmatrix}$$

$$= \begin{pmatrix} \text{vec} \left( \overbrace{(v_1 + v_2 \quad v_3 + v_4 \quad \cdots \quad v_{T-1} + v_T)}^{= \boldsymbol{v}_+^\top} \mathbf{H}_{n-1} \right) \\ \text{vec} \left( \underbrace{(v_1 - v_2 \quad v_3 - v_4 \quad \cdots \quad v_{T-1} - v_T)}_{= \boldsymbol{v}_-^\top} I_{2^{n-1}} \right) \end{pmatrix}$$

$$= \begin{pmatrix} \mathbf{H}_{n-1}^\top \boldsymbol{v}_+ \\ I_{2^{n-1}} \boldsymbol{v}_- \end{pmatrix}. \qquad \square$$

**Lemma 6.** *Let $\mathbf{H}_n$ be the unnormalized Haar basis matrix of order $n$. Then, $\mathbf{H}_n \mathbf{H}_n^\top$ satisfies*

$$\mathbf{H}_n \mathbf{H}_n^\top = \mathbf{H}_{n-1} \mathbf{H}_{n-1}^\top \otimes \begin{pmatrix} 1 & 1 \\ 1 & 1 \end{pmatrix} + I_{2^{n-1}} \otimes \begin{pmatrix} 1 & -1 \\ -1 & 1 \end{pmatrix} \tag{8}$$

$$= \begin{pmatrix} \mathbf{H}_{n-1} \mathbf{H}_{n-1}^\top + \mathbf{1}_{2^{n-1}} & \mathbf{0}_{2^{n-1}} \\ \mathbf{0}_{2^{n-1}} & \mathbf{H}_{n-1} \mathbf{H}_{n-1}^\top + \mathbf{1}_{2^{n-1}}, \end{pmatrix}, \tag{9}$$

*where $\mathbf{1}_{2^{n-1}}$ and $\mathbf{0}_{2^{n-1}}$ are $2^{n-1} \times 2^{n-1}$ matrices of 1's and 0's respectively.*

*Proof.* For brevity, let us denote $\boldsymbol{B}_n = \mathbf{H}_n \mathbf{H}_n^\top$. The first equality follows from elementary properties of block matrices and the Kronecker product: using the recursive form of $\mathbf{H}_n$, we have

$$\boldsymbol{B}_n = \mathbf{H}_n \mathbf{H}_n^\top$$

$$= \left( \mathbf{H}_{n-1} \otimes \begin{pmatrix} 1 \\ 1 \end{pmatrix} \quad \boldsymbol{I}_{2^{n-1}} \otimes \begin{pmatrix} 1 \\ -1 \end{pmatrix} \right) \begin{pmatrix} \mathbf{H}_{n-1}^\top \otimes (1 \quad 1) \\ \boldsymbol{I}_{2^{n-1}} \otimes (1 \quad -1) \end{pmatrix}$$

$$= \mathbf{H}_{n-1} \otimes \begin{pmatrix} 1 \\ 1 \end{pmatrix} \mathbf{H}_{n-1}^\top \otimes (1 \quad 1) + \boldsymbol{I}_{2^{n-1}} \otimes \begin{pmatrix} 1 \\ -1 \end{pmatrix} \boldsymbol{I}_{2^{n-1}} \otimes (1 \quad -1)$$

$$= \mathbf{H}_{n-1} \mathbf{H}_{n-1}^\top \otimes \begin{pmatrix} 1 \\ 1 \end{pmatrix} (1 \quad 1) + \boldsymbol{I}_{2^{n-1}} \otimes \begin{pmatrix} 1 \\ -1 \end{pmatrix} (1 \quad -1)$$

$$= \mathbf{H}_{n-1} \mathbf{H}_{n-1}^\top \otimes \begin{pmatrix} 1 & 1 \\ 1 & 1 \end{pmatrix} + \boldsymbol{I}_{2^{n-1}} \otimes \begin{pmatrix} 1 & -1 \\ -1 & 1 \end{pmatrix}$$

$$= \boldsymbol{B}_{n-1} \otimes \begin{pmatrix} 1 & 1 \\ 1 & 1 \end{pmatrix} + \boldsymbol{I}_{2^{n-1}} \otimes \begin{pmatrix} 1 & -1 \\ -1 & 1 \end{pmatrix} .$$

To get the second expression, let us proceed by induction. We have $\boldsymbol{B}_0 = (1)$ and

$$\boldsymbol{B}_1 = \mathbf{H}_1 \mathbf{H}_1^\top = \begin{pmatrix} 1 & 1 \\ 1 & -1 \end{pmatrix} \begin{pmatrix} 1 & 1 \\ 1 & -1 \end{pmatrix}^\top = \begin{pmatrix} 2 & 0 \\ 0 & 2 \end{pmatrix} = \begin{pmatrix} \boldsymbol{B}_0 + \mathbf{1}_1 & \mathbf{0}_1 \\ \mathbf{0}_1 & \boldsymbol{B}_0 + \mathbf{1}_1 \end{pmatrix} .$$

Next, let us assume that $\boldsymbol{B}_n$ satisfies

$$\boldsymbol{B}_n = \begin{pmatrix} \boldsymbol{B}_{n-1} + \mathbf{1}_{2^{n-1}} & \mathbf{0}_{2^{n-1}} \\ \mathbf{0}_{2^{n-1}} & \boldsymbol{B}_{n-1} + \mathbf{1}_{2^{n-1}} \end{pmatrix} .$$

Then, applying the recursive form Equation (8) for $\boldsymbol{B}_{n+1}$, we have

$$\boldsymbol{B}_{n+1} = \boldsymbol{B}_n \otimes \begin{pmatrix} 1 & 1 \\ 1 & 1 \end{pmatrix} + \boldsymbol{I}_{2^n} \otimes \begin{pmatrix} 1 & -1 \\ -1 & 1 \end{pmatrix}$$

$$= \begin{pmatrix} \boldsymbol{B}_{n-1} + \mathbf{1}_{2^{n-1}} & \mathbf{0}_{2^{n-1}} \\ \mathbf{0}_{2^{n-1}} & \boldsymbol{B}_{n-1} + \mathbf{1}_{2^{n-1}} \end{pmatrix} \otimes \begin{pmatrix} 1 & 1 \\ 1 & 1 \end{pmatrix} + \boldsymbol{I}_{2^n} \otimes \begin{pmatrix} 1 & -1 \\ -1 & 1 \end{pmatrix}$$

$$= \begin{pmatrix} \boldsymbol{B}_{n-1} \otimes \begin{pmatrix} 1 & 1 \\ 1 & 1 \end{pmatrix} + \mathbf{1}_{2^{n-1}} \otimes \begin{pmatrix} 1 & 1 \\ 1 & 1 \end{pmatrix} & \mathbf{0}_{2^n} \\ \mathbf{0}_{2^n} & \boldsymbol{B}_{n-1} \otimes \begin{pmatrix} 1 & 1 \\ 1 & 1 \end{pmatrix} + \mathbf{1}_{2^{n-1}} \otimes \begin{pmatrix} 1 & 1 \\ 1 & 1 \end{pmatrix} \end{pmatrix}$$

$$+ \begin{pmatrix} \boldsymbol{I}_{2^{n-1}} \otimes \begin{pmatrix} 1 & -1 \\ -1 & 1 \end{pmatrix} & \mathbf{0}_{2^n} \\ \mathbf{0}_{2^n} & \boldsymbol{I}_{2^n} \otimes \begin{pmatrix} 1 & -1 \\ -1 & 1 \end{pmatrix} \end{pmatrix}$$

$$= \begin{pmatrix} \boldsymbol{B}_n + \mathbf{1}_{2^n} & \mathbf{0}_{2^n} \\ \mathbf{0}_{2^n} & \boldsymbol{B}_n + \mathbf{1}_{2^n,} \end{pmatrix},$$

where the last line observes that $\mathbf{1}_{2^{n-1}} \otimes \begin{pmatrix} 1 & 1 \\ 1 & 1 \end{pmatrix} = \mathbf{1}_{2^n}$ and that after adding the two block matrices, the top left and bottom right blocks are both

$$\boldsymbol{B}_{n-1} \otimes \begin{pmatrix} 1 & 1 \\ 1 & 1 \end{pmatrix} + \boldsymbol{I}_{2^{n-1}} \otimes \begin{pmatrix} 1 & -1 \\ -1 & 1 \end{pmatrix} + \mathbf{1}_{2^n} = \boldsymbol{B}_n + \mathbf{1}_{2^n},$$

via Equation (8). Hence, the stated result follows by induction. $\square$

Now using this, we have the following bound on the norm of the high-dimensional surrogate losses.

**Lemma 7.** *Let $n = \log_2(T)$, $\mathbf{e}_t$ be the $t^{th}$ standard basis vector of $\mathbb{R}^T$, and for $\boldsymbol{g}_t \in \mathbb{R}^d$ let $\widetilde{\boldsymbol{g}}_t = \mathbf{e}_t \otimes \boldsymbol{g}_t \in \mathbb{R}^{dT}$. Let $\mathbf{H}_n$ be a Haar matrix of order $n$ and let $\boldsymbol{B} = \mathbf{H}_n \otimes \boldsymbol{I}_d$ be it's block extension to sequence in $\mathbb{R}^d$. Then, we have*

$$\|\widetilde{\boldsymbol{g}}_t\|_{\boldsymbol{B}\boldsymbol{B}^\top}^2 = (\log T + 1) \|\boldsymbol{g}_t\|_2^2 .$$

*Proof.* Using Lemma 1, we have that

$$\|\widetilde{\boldsymbol{g}}_t\|_{\boldsymbol{BB}^\top}^2 = \left[\boldsymbol{H}_n\boldsymbol{H}_n^\top\right]_{tt}\|\boldsymbol{g}_t\|^2 \ .$$

Moreover, using Equation (9) it can easily be seen that the diagonal entries of $\boldsymbol{H}_n\boldsymbol{H}_n^\top$ are $\log_2 T + 1$, so we have

$$\|\widetilde{\boldsymbol{g}}_t\|_{\boldsymbol{BB}^\top}^2 \leq (1 + \log_2 T)\|\boldsymbol{g}_t\|_2^2 \ . \qquad \square$$

### B.3 Proof of Proposition 3

**Proposition 3.** *Let $n = \log_2 T$ and $\mathbf{H}_n$ be the unnormalized Haar basis matrix of order $n$. For any $\tau \in \{2^i : i = 0, \dots, \log_2 T\}$, let $N_\tau = T/\tau$ and let $\mathcal{I}_1^{(\tau)}, \dots, \mathcal{I}_{N_\tau}^{(\tau)}$ be a partition of $[T]$ into intervals of length $\tau$. Define the average comparator in interval $\mathcal{I}_i^{(\tau)}$ to be $\bar{\boldsymbol{u}}_i^{(\tau)} = \frac{1}{\tau}\sum_{t\in\mathcal{I}_i^{(\tau)}} \boldsymbol{u}_t$, and define the* squared path-length at time-scale $\tau < T$ *to be*

$$\bar{P}(\vec{\boldsymbol{u}}, \tau) := \sum_{i=1}^{N_\tau/2} \left\|\bar{\boldsymbol{u}}_{2i-1}^{(\tau)} - \bar{\boldsymbol{u}}_{2i}^{(\tau)}\right\|_2^2,$$

*and $\bar{P}(\vec{\boldsymbol{u}}, T) = \left\|\bar{\boldsymbol{u}}_1^{(T)}\right\|_2^2 = \|\bar{\boldsymbol{u}}\|_2^2$. Then, setting $\mathbf{S} = [\mathbf{H}_n\mathbf{H}_n^\top]^{-1}$ and $\boldsymbol{M} = \mathbf{S}\otimes\boldsymbol{I}_d$, we have*

$$\|\widetilde{\boldsymbol{u}}\|_{\boldsymbol{M}}^2 \leq \|\bar{\boldsymbol{u}}\|_2^2 + \frac{1}{4}\sum_{i=0}^{\log_2(T)} \bar{P}(\vec{\boldsymbol{u}}, 2^i) \leq \|\bar{\boldsymbol{u}}\|_2^2 + \frac{1}{4}\log(T)\max_\tau \bar{P}(\vec{\boldsymbol{u}}, \tau),$$

$$\|\widetilde{\boldsymbol{g}}_t\|_{\boldsymbol{M}^{-1}}^2 = \|\boldsymbol{g}_t\|_2^2(1 + \log T) \ .$$

*Proof.* The proof of the claim $\|\widetilde{\boldsymbol{g}}_t\|_{\boldsymbol{M}^{-1}}^2 = \|\widetilde{\boldsymbol{g}}_t\|_{\mathbf{HH}^\top}^2 = \|\boldsymbol{g}_t\|_2^2[\log_2(T)+1]$ is provided in Lemma 7.

To see the form of $\|\widetilde{\boldsymbol{u}}\|_{\boldsymbol{M}}^2$, let us first write

$$\|\widetilde{\boldsymbol{u}}\|_{\boldsymbol{M}}^2 = \left\langle\widetilde{\boldsymbol{u}}, [\boldsymbol{HH}^\top]^{-1}\widetilde{\boldsymbol{u}}\right\rangle = \left\langle\boldsymbol{H}^{-1}\widetilde{\boldsymbol{u}}, \boldsymbol{H}^{-1}\widetilde{\boldsymbol{u}}\right\rangle = \left\|\boldsymbol{H}^{-1}\widetilde{\boldsymbol{u}}\right\|_2^2 \ .$$

The result then follows by showing that

$$\boldsymbol{H}^{-1}\widetilde{\boldsymbol{u}} = \frac{1}{2}\begin{pmatrix} 2\bar{\boldsymbol{u}} \\ \bar{\boldsymbol{u}}_1^{(T/2)} - \bar{\boldsymbol{u}}_2^{(T/2)} \\ \bar{\boldsymbol{u}}_1^{(T/4)} - \bar{\boldsymbol{u}}_2^{(T/4)} \\ \bar{\boldsymbol{u}}_3^{(T/4)} - \bar{\boldsymbol{u}}_4^{(T/4)} \\ \vdots \\ \boldsymbol{u}_1 - \boldsymbol{u}_2 \\ \boldsymbol{u}_3 - \boldsymbol{u}_4 \\ \vdots \\ \boldsymbol{u}_{T-1} - \boldsymbol{u}_T \end{pmatrix}, \qquad (10)$$

so that

$$\left\|\boldsymbol{H}^{-1}\widetilde{\boldsymbol{u}}\right\|_2^2 = \underbrace{\|\bar{\boldsymbol{u}}\|_2^2}_{\bar{P}(T)} + \underbrace{\frac{1}{4}\left\|\bar{\boldsymbol{u}}_1^{(T/2)} - \bar{\boldsymbol{u}}_2^{(T/2)}\right\|_2^2}_{\bar{P}(T/2)} + \underbrace{\frac{1}{4}\left\|\bar{\boldsymbol{u}}_1^{(T/4)} - \bar{\boldsymbol{u}}_2^{(T/4)}\right\|_2^2 + \frac{1}{4}\left\|\bar{\boldsymbol{u}}_3^{(T/4)} - \bar{\boldsymbol{u}}_4^{(T/4)}\right\|_2^2}_{\bar{P}(T/4)}$$

$$+ \dots + \underbrace{\frac{1}{4}\|\boldsymbol{u}_1 - \boldsymbol{u}_2\|_2^2 + \frac{1}{4}\|\boldsymbol{u}_3 - \boldsymbol{u}_4\|_2^2 + \dots + \frac{1}{4}\|\boldsymbol{u}_{T-1} - \boldsymbol{u}_T\|_2^2}_{=\bar{P}(1)},$$

where for brevity we have dropped the argument $\vec{\boldsymbol{u}}$ on $\bar{P}(\vec{\boldsymbol{u}}, \tau)$.

Equation (10) is best shown via example; the general case is mostly a tedius exercise which we provide at the end. Assume $T = 4$, then the Haar matrix of order $n = \log_2(T) = 2$ is

$$\mathbf{H}_2 = \begin{pmatrix} 1 & 1 & 1 & 0 \\ 1 & 1 & -1 & 0 \\ 1 & -1 & 0 & 1 \\ 1 & -1 & 0 & -1 \end{pmatrix} = \underbrace{\begin{pmatrix} \frac{1}{2} & \frac{1}{2} & \frac{1}{\sqrt{2}} & 0 \\ \frac{1}{2} & \frac{1}{2} & \frac{-1}{\sqrt{2}} & 0 \\ \frac{1}{2} & -\frac{1}{2} & 0 & \frac{1}{\sqrt{2}} \\ \frac{1}{2} & -\frac{1}{2} & 0 & \frac{-1}{\sqrt{2}} \end{pmatrix}}_{=: \widetilde{\mathbf{H}}_2} \underbrace{\begin{pmatrix} 2 & 0 & 0 & 0 \\ 0 & 2 & 0 & 0 \\ 0 & 0 & \sqrt{2} & 0 \\ 0 & 0 & 0 & \sqrt{2} \end{pmatrix}}_{=: \boldsymbol{D}_2}.$$

It is well-known that for any $T$ the columns of $\mathbf{H}_{\log_2(T)}$ form an orthogonal basis of $\mathbb{R}^T$ [41, Chapter 6.1.1], which implies that $\widetilde{\mathbf{H}}_2$ is orthonormal. So, $\widetilde{\mathbf{H}}_2^{-1} = \widetilde{\mathbf{H}}_2^\top$ and

$$\mathbf{H}_2^{-1} = (\widetilde{\mathbf{H}}_2 \boldsymbol{D}_2)^{-1} = \boldsymbol{D}_2^{-1} \widetilde{\mathbf{H}}_2^{-1} = \boldsymbol{D}_2^{-1} \widetilde{\mathbf{H}}_2^\top$$

$$= \begin{pmatrix} \frac{1}{2} & 0 & 0 & 0 \\ 0 & \frac{1}{2} & 0 & 0 \\ 0 & 0 & \frac{1}{\sqrt{2}} & 0 \\ 0 & 0 & 0 & \frac{1}{\sqrt{2}} \end{pmatrix} \begin{pmatrix} \frac{1}{2} & \frac{1}{2} & \frac{1}{2} & \frac{1}{2} \\ \frac{1}{2} & \frac{1}{2} & -\frac{1}{2} & -\frac{1}{2} \\ \frac{1}{\sqrt{2}} & -\frac{1}{\sqrt{2}} & 0 & 0 \\ 0 & 0 & \frac{1}{\sqrt{2}} & \frac{-1}{\sqrt{2}} \end{pmatrix} = \begin{pmatrix} \frac{1}{4} & \frac{1}{4} & \frac{1}{4} & \frac{1}{4} \\ \frac{1}{4} & \frac{1}{4} & -\frac{1}{4} & -\frac{1}{4} \\ \frac{1}{2} & -\frac{1}{2} & 0 & 0 \\ 0 & 0 & \frac{1}{2} & \frac{-1}{2} \end{pmatrix},$$

which leads to Equation (10) after applying the Kronecker product:

$$H^{-1}\widetilde{\boldsymbol{u}} = \begin{pmatrix} \frac{\boldsymbol{I}_d}{4} & \frac{\boldsymbol{I}_d}{4} & \frac{\boldsymbol{I}_d}{4} & \frac{\boldsymbol{I}_d}{4} \\ \frac{\boldsymbol{I}_d}{4} & \frac{\boldsymbol{I}_d}{4} & -\frac{\boldsymbol{I}_d}{4} & -\frac{\boldsymbol{I}_d}{4} \\ \frac{\boldsymbol{I}_d}{2} & -\frac{\boldsymbol{I}_d}{2} & \boldsymbol{0} & \boldsymbol{0} \\ \boldsymbol{0} & \boldsymbol{0} & \frac{\boldsymbol{I}_d}{2} & \frac{-\boldsymbol{I}_d}{2} \end{pmatrix} \begin{pmatrix} \boldsymbol{u}_1 \\ \vdots \\ \boldsymbol{u}_T \end{pmatrix} = \begin{pmatrix} \frac{\boldsymbol{u}_1 + \boldsymbol{u}_2 + \boldsymbol{u}_3 + \boldsymbol{u}_4}{4} \\ \frac{\boldsymbol{u}_1 + \boldsymbol{u}_2 - \boldsymbol{u}_3 - \boldsymbol{u}_4}{4} \\ \frac{\boldsymbol{u}_1 - \boldsymbol{u}_2}{2} \\ \frac{\boldsymbol{u}_3 - \boldsymbol{u}_4}{2} \end{pmatrix} = \frac{1}{2} \begin{pmatrix} 2\bar{\boldsymbol{u}} \\ \bar{\boldsymbol{u}}_1^{(T/2)} - \bar{\boldsymbol{u}}_2^{(T/2)} \\ \boldsymbol{u}_1 - \boldsymbol{u}_2 \\ \boldsymbol{u}_3 - \boldsymbol{u}_4 \end{pmatrix}.$$

More generally, start with $d = 1$ begin again by factoring

$$\mathbf{H}_n^{-1} = \boldsymbol{D}_n^{-1} \widetilde{\mathbf{H}}_n^\top = \boldsymbol{D}_n^{-2} \mathbf{H}_n^\top,$$

where now $\widetilde{\mathbf{H}}_n$ is the normalized Haar basis matrix of order $n = \log_2(T)$ and

$$\boldsymbol{D}_n = \mathrm{Diag}\left( \sqrt{T}, \underbrace{\sqrt{T}}_{2^0}, \underbrace{\sqrt{T/2}, \sqrt{T/2}}_{2^1}, \underbrace{\sqrt{T/4}, \ldots, \sqrt{T/4}}_{2^2}, \ldots, \underbrace{\sqrt{2}, \ldots, \sqrt{2}}_{2^{n-1}} \right).$$

The result is then attained by unrolling the recursion for $\mathbf{H}_n^\top \widetilde{\boldsymbol{u}}$ given by Lemma 5 and factoring in the normalization factors $\boldsymbol{D}_n^{-2}$. The result for $d > 1$ is then immediately implied by observing that the block matrix $\mathbf{H}_n^{-1} \otimes \boldsymbol{I}_d$ will act upon the vector components of $\widetilde{\boldsymbol{u}} \in \mathbb{R}^{dT}$ in an identical way to how $\mathbf{H}_n^{-1}$ acts upon a vector of scalars. $\square$

### B.4   Proof of Proposition 4

**Proposition 4.** *The algorithm characterized by applying Theorem 2 with $\boldsymbol{S} = [\boldsymbol{H}_n \boldsymbol{H}_n^\top]^{-1}$ can be implemented with $\mathcal{O}(d \log T)$ per-round computation.*

*Proof.* Note that the losses passed to the 1-dimensional parameter-free algorithm are $\langle \widetilde{\boldsymbol{v}}_t, \widetilde{\boldsymbol{g}}_t \rangle = \langle \widetilde{\boldsymbol{v}}_t, \boldsymbol{e}_t \otimes \boldsymbol{g}_t \rangle$, and since $\boldsymbol{e}_t \otimes \boldsymbol{g}_t$ has only $d$ active indices we can compute the 1-dimensional learner's losses in $\mathcal{O}(d)$. As such, the 1-dimensional learner can be implemented in $\mathcal{O}(d)$ per-round computation.

For the direction learner, we are to show that each of the relevant variables can be maintained using only $\mathcal{O}(d \log T)$ per-round computation.

Using Proposition 3, we immediately have $V_{t+1} = V_t + \|\widetilde{\boldsymbol{g}}_t\|_{\boldsymbol{M}^{-1}}^2 = V_t + (\log T + 1) \|\boldsymbol{g}_t\|^2$, so $V_{t+1}$ can be maintained using only $\mathcal{O}(d)$ per-round computation (*i.e.*, to compute $\|\boldsymbol{g}_t\|^2$).

For the scaling factor $\left\|\widetilde{\boldsymbol{\theta}}_{t+1}\right\|_{\boldsymbol{M}^{-1}}$, observe that

$$\left\|\widetilde{\boldsymbol{\theta}}_{t+1}\right\|_{\boldsymbol{M}^{-1}}^2 = \|\widetilde{\boldsymbol{g}}_t\|_{\boldsymbol{M}^{-1}}^2 + \left\|\widetilde{\boldsymbol{\theta}}_t\right\|_{\boldsymbol{M}^{-1}}^2 + 2\left\langle \widetilde{\boldsymbol{\theta}}_t, \boldsymbol{M}^{-1} \widetilde{\boldsymbol{g}}_t \right\rangle.$$

Hence, we again have $\mathcal{O}(d)$ per-round computation to compute $\|\widetilde{\boldsymbol{g}}_t\|^2_{\boldsymbol{M}^{-1}}$, and letting $\boldsymbol{h}_t = \boldsymbol{H}_n^\top \boldsymbol{e}_t$ we can decompose the last term as

$$
\begin{aligned}
\left\langle \widetilde{\boldsymbol{\theta}}_t, \left(\boldsymbol{H}_n \boldsymbol{H}_n^\top \otimes \boldsymbol{I}_d\right)(\mathbf{e}_t \otimes \boldsymbol{g}_t)\right\rangle &= \left\langle \widetilde{\boldsymbol{\theta}}_t, \left(\boldsymbol{H}_n \boldsymbol{H}_n^\top \mathbf{e}_t \otimes \boldsymbol{g}_t\right)\right\rangle \\
&= \left\langle \sum_{i=1}^{t-1} \boldsymbol{e}_i \otimes \boldsymbol{g}_i, \boldsymbol{H}_n \boldsymbol{h}_t \otimes \boldsymbol{g}_t\right\rangle \\
&= \sum_{i=1}^{t-1} (\boldsymbol{e}_i^\top \otimes \boldsymbol{g}_i^\top)(\boldsymbol{H}_n \boldsymbol{h}_t \otimes \boldsymbol{g}_t) \\
&= \sum_{i=1}^{t-1} \boldsymbol{e}_i^\top \boldsymbol{H}_n \boldsymbol{h}_t \otimes \langle \boldsymbol{g}_i, \boldsymbol{g}_t\rangle \\
&= \sum_{i=1}^{t-1} \langle \boldsymbol{h}_i, \boldsymbol{h}_t\rangle \langle \boldsymbol{g}_i, \boldsymbol{g}_t\rangle \\
&= \left\langle \sum_{i=1}^{t-1} \boldsymbol{h}_i \langle \boldsymbol{g}_i, \boldsymbol{g}_t\rangle, \boldsymbol{h}_t\right\rangle \\
&= \left\langle \sum_{i=1}^{t-1} \boldsymbol{h}_i \boldsymbol{g}_i^\top \boldsymbol{g}_t, \boldsymbol{h}_t\right\rangle \\
&= \left\langle \boldsymbol{g}_t, \underbrace{\left[\sum_{i=1}^{t-1} \boldsymbol{g}_i \boldsymbol{h}_i^\top\right]}_{=:\boldsymbol{\Lambda}_t} \boldsymbol{h}_t\right\rangle.
\end{aligned}
$$

From Proposition 5, for any $t$ the vector $\boldsymbol{h}_t = \boldsymbol{H}_n^\top \boldsymbol{e}_t$ has only $\log T + 1$ active non-zero elements by construction of the Haar basis, so given $\boldsymbol{\Lambda}_t$, the product $\boldsymbol{\Lambda}_t \boldsymbol{h}_t$ takes a linear combination of $\log T + 1$ vectors in $\mathbb{R}^d$, for $\mathcal{O}(d \log T)$ operations. Note that the variable $\boldsymbol{\Lambda}_t$ can also be maintained with $\mathcal{O}(d \log T)$ operations since each term is $\boldsymbol{g}_t \boldsymbol{h}_t^\top$, which involves updating $\mathcal{O}(\log T)$ columns of $\boldsymbol{\Lambda}_{t-1} \in \mathbb{R}^{d \times T}$. Hence overall we can maintain $\left\|\widetilde{\boldsymbol{\theta}}_{t+1}\right\|_{\boldsymbol{M}^{-1}}$ using $\mathcal{O}(d \log T)$ per-round computation.

Lastly, consider the variable $\widetilde{\boldsymbol{\theta}}_{t+1}$. Observe that we can maintain a variable $\widehat{\boldsymbol{\theta}}_{t+1} = -\left(\boldsymbol{H}_n^\top \otimes \boldsymbol{I}_d\right)\sum_{s=1}^t \widetilde{\boldsymbol{g}}_s$ using $\mathcal{O}(d \log T)$ computation:

$$
\widehat{\boldsymbol{\theta}}_{t+1} = -\left(\boldsymbol{H}_n^\top \otimes \boldsymbol{I}_d\right)\sum_{s=1}^t \widetilde{\boldsymbol{g}}_s = \widehat{\boldsymbol{\theta}}_t - \left(\boldsymbol{H}_n^\top \boldsymbol{e}_t \otimes \boldsymbol{g}_t\right) = \widehat{\boldsymbol{\theta}}_t - (\boldsymbol{h}_t \otimes \boldsymbol{g}_t),
$$

since $\boldsymbol{h}_t \otimes \boldsymbol{g}_t$ is a block vector containing $\log(T) + 1$ non-zeros blocks of length $d$. Hence,

$$
\begin{aligned}
\widetilde{\boldsymbol{\theta}}_{t+1} &= (\boldsymbol{H}_n \boldsymbol{H}_n^\top \otimes \boldsymbol{I}_n)\sum_{s=1}^t \boldsymbol{g}_s = (\boldsymbol{H}_n \otimes \boldsymbol{I}_n)(\boldsymbol{H}_n^\top \otimes \boldsymbol{I}_n)\sum_{s=1}^t \widetilde{\boldsymbol{g}}_s \\
&= (\boldsymbol{H}_n \otimes \boldsymbol{I}_n)\widehat{\boldsymbol{\theta}}_{t+1},
\end{aligned}
$$

and again via the construction of the Haar basis, each row of $\boldsymbol{H}_n$ (i.e., each column of $\boldsymbol{H}_n^\top$) has only $\log T + 1$ non-zero entries, we can compute each $d \times 1$ block of $\widetilde{\boldsymbol{\theta}}_{t+1}$ using $\mathcal{O}(d \log T)$ computation. Finally, observe that in order to implement the direction learner, we need only compute the $t^{\text{th}}$ $d \times 1$ block of $\widetilde{\boldsymbol{\theta}}_t$. Indeed, since for each $t$, the vector $\widetilde{\boldsymbol{g}}_t = \boldsymbol{e}_t \otimes \boldsymbol{g}_t$ has only $d$ non-zero indices, it suffices to retrieve the corresponding indices of $\widetilde{\boldsymbol{v}}_t$ to implement direction learner. $\square$

We note that the memory overhead of maintaining each of these variables can also likely be reduced by more careful bookkeeping, and acknowledging the fact that the algorithm only really needs to retrieve the $t^{\text{th}}$ block of $\widetilde{\boldsymbol{w}}_t$, since the losses are $\widetilde{\boldsymbol{g}}_t = \boldsymbol{e}_t \otimes \boldsymbol{g}_t$. We omit these considerations here for brevity.

## C  Proofs for Section 5 (Recovering Variance-Variability Coupling Guarantees)

### C.1  Proof of Theorem 3

**Theorem 3.** *Let* $\mathrm{Wealth}_T := -\sum_{t=1}^{T} \langle \widetilde{g}_t, \widetilde{w}_t \rangle$ *denote the "wealth" of an algorithm* $\mathcal{A}$ *and let* $(f, f^*)$ *be a Fenchel conjugate pair. Then* $\mathcal{A}$ *guarantees* $\mathrm{Wealth}_T \geq f_T^*\big(-\sum_{t=1}^{T} \widetilde{g}_t\big)$ *for any sequence* $\widetilde{g}_1, \ldots, \widetilde{g}_T$ *if and only if* $R_T(\vec{u}) \leq f_T(\widetilde{u})$ *for any sequence* $\vec{u} = (u_1, \ldots, u_T)$ *in* $\mathcal{W}$, *where* $\widetilde{u} = (u_1^\top, \ldots, u_T^\top)^\top$ *is the concatenation of the sequence* $\vec{u}$ *into a vector.*

*Proof.* Thanks to Proposition 1, the proof is essentially the same as the usual one. We provide the argument here for completeness.

From Proposition 1, $R_T(\vec{u}) = R_T^{\mathrm{Seq}}(\widetilde{u}) = \sum_{t=1}^{T} \langle \widetilde{g}_t, \widetilde{w}_t - \widetilde{u} \rangle$ for $\widetilde{g}_t = \mathbf{e}_t \otimes g_t$ and $\widetilde{u} = \sum_{t=1}^{T} \mathbf{e}_t \otimes u_t$. Hence, recalling the definition of the Fenchel conjugate, we have

$$R_T(\vec{u}) = \sum_{t=1}^{T} \langle g_t, w_t - u_t \rangle = \sum_{t=1}^{T} \langle \widetilde{g}_t, \widetilde{w}_t - \widetilde{u} \rangle = -\mathrm{Wealth}_T - \sum_{t=1}^{T} \langle \widetilde{g}_t, \widetilde{u} \rangle$$

$$\leq \Big\langle -\sum_{t=1}^{T} \widetilde{g}_t, \widetilde{u} \Big\rangle - f_T^*\Big(-\sum_{t=1}^{T} \widetilde{g}_t\Big) \leq \sup_{\boldsymbol{\theta}} \langle \boldsymbol{\theta}, \widetilde{u} \rangle - f_T^*(\boldsymbol{\theta}) = f_T(\widetilde{u}) \,.$$

Similarly, for the other direction, suppose we have $R_T(\vec{u}) = R_T^{\mathrm{Seq}}(\widetilde{u}) \leq f_T(\widetilde{u})$ for any $\widetilde{u}$. Then re-arranging, we have $\mathrm{Wealth}_T \geq \Big\langle -\sum_{t=1}^{T} \widetilde{g}_t, \widetilde{u} \Big\rangle - f_T(\widetilde{u})$, and since this holds for any $\widetilde{u}$, we can choose the one that tightens the bound to get $\mathrm{Wealth}_T \geq \sup_{\widetilde{u}} \Big\langle -\sum_{t=1}^{T} \widetilde{g}_t, \widetilde{u} \Big\rangle - f_T(\widetilde{u}) = f_T^*(-\sum_{t=1}^{T} \widetilde{g}_t)$. $\qquad\square$

## D  Supporting Lemmas

**Lemma 8.** *Let* $\boldsymbol{\Sigma} \in \mathbb{R}^{T \times T}$ *be the finite-difference operator, having entries*

$$\Sigma_{ij} = \begin{cases} 1 & \text{if } i = j \\ -1 & \text{if } j = i+1 \\ 0 & \text{otherwise} \end{cases} .$$

*Then,*

1. *The inverse of* $\boldsymbol{\Sigma}$ *the upper-triangular matrix of* 1*'s:*

$$\Sigma_{ij}^{-1} = \begin{cases} 1 & \text{if } j \geq i \\ 0 & \text{otherwise} \end{cases}, \quad \forall i, j \,.$$

2. *The eigenvalues of* $\boldsymbol{\Sigma}$ *and* $\boldsymbol{\Sigma}^{-1}$ *are* $\lambda_i = 1$ *for all* $i \in [T]$.

3. $x \mapsto x^\top \boldsymbol{\Sigma} x$ *is positive definite.*

*Moreover, the analogous properties hold for the block matrix* $\boldsymbol{\Sigma} \otimes \boldsymbol{I}_d \in \mathbb{R}^{dT \times dT}$.

*Proof.* The inverse of $\boldsymbol{\Sigma}$ is the upper-triangular matrix $\boldsymbol{\Delta}$ characterized by entries

$$\Delta_{ij} = \begin{cases} 1 & \text{if } j \geq i \\ 0 & \text{otherwise} \end{cases} .$$

To see why, observe that we have $\Sigma_{T,T} \Delta_{T,T} = 1$ and for $i < T$ we have

$$[\boldsymbol{\Sigma}\boldsymbol{\Delta}]_{ij} = \sum_{i,j} \Sigma_{ik} \Delta_{kj} = \Delta_{ij} - \Delta_{i+1,j} = \begin{cases} 1 & \text{if } i = j \\ 0 & \text{otherwise} \end{cases},$$

and likewise for $[\mathbf{\Delta\Sigma}]_{ij}$. Hence $\mathbf{\Sigma\Delta} = \mathbf{\Delta\Sigma} = I$ and $M^{-1} = \mathbf{\Delta}$.

Next, since $\mathbf{\Sigma}$ and $\mathbf{\Sigma}^{-1}$ are upper-triangular, their eigenvalues are equal to their diagonal entries, and hence both have eigenvalues $\lambda_i = 1$ for all $i$.

To see that the asymmetric matrix $\mathbf{\Sigma}$ is positive definite, it suffices to show that the symmetric part of $\mathbf{\Sigma}$, *i.e.*, the matrix $\mathbf{\Sigma}_S = (\mathbf{\Sigma} + \mathbf{\Sigma}^\top)/2$, is positive definite [23]. Luckily, $\mathbf{\Sigma}_S$ is also a well-known variation of the discrete difference operator and is known to be positive definite [see, *e.g.*, Theorem 7.4.7 in 39].

For the block matrix $\boldsymbol{B} = \mathbf{\Sigma} \otimes \boldsymbol{I}_d$, the inverse is given immediately by the inverse property of the Kronecker product: $\boldsymbol{B}^{-1} = (\mathbf{\Sigma} \otimes \boldsymbol{I}_d)^{-1} = \mathbf{\Sigma}^{-1} \otimes \boldsymbol{I}_d$. We also have that $\boldsymbol{B} = \mathbf{\Sigma} \otimes \boldsymbol{I}_d$ and $\boldsymbol{B}^{-1}$ have eigenvalues $\lambda_i = 1$ for all $i \in [dT]$, since both are again upper-triangular with 1's on their main diagonal. Finally, we have positive definiteness of $\boldsymbol{B}$ using the fact that the symmetric part of $\boldsymbol{B} = \mathbf{\Sigma} \otimes \boldsymbol{I}_d$ is $\frac{1}{2}(\boldsymbol{B} + \boldsymbol{B}^\top) = \frac{1}{2}(\mathbf{\Sigma} \otimes \boldsymbol{I}_d + \mathbf{\Sigma}^\top \otimes \boldsymbol{I}_d) = \frac{1}{2}(\mathbf{\Sigma} + \mathbf{\Sigma}^\top) \otimes \boldsymbol{I}_d$ by the distributive property, hence $\boldsymbol{B}$ is the Kronecker product of two symmetric positive definite matrices, so $\boldsymbol{B}$ is positive definite [38, Chapter 2]. $\qquad\square$

We borrow the following eigenvalue bound from [43].

**Theorem 5.** *(Wolkowicz and Styan [43, Theorem 2.1]) Let $\boldsymbol{A}$ be a symmetric $n \times n$ matrix with eigenvalues $\lambda_1(\boldsymbol{A}) \leq \ldots \leq \lambda_n(\boldsymbol{A})$. Then*

$$\lambda_{\max}(\boldsymbol{A}) \leq \frac{\mathrm{Tr}(\boldsymbol{A})}{n} + \sqrt{(n-1)\left[\frac{\mathrm{Tr}(\boldsymbol{A}^\top \boldsymbol{A})}{n} - \left(\frac{\mathrm{Tr}(\boldsymbol{A})}{n}\right)^2\right]}.$$

