# OpenReview forum: "An Equivalence Between Static and Dynamic Regret Minimization"
_NeurIPS.cc/2024/Conference — NeurIPS 2024 poster_

### Official Review · Reviewer_bfmX · 2024-07-10

**Soundness:** 4
**Presentation:** 2
**Contribution:** 4
**Rating:** 6
**Confidence:** 3

**Summary:**

This paper demonstrates that dynamic regret minimization can be reduced to static regret minimization by embedding the comparator sequence into a higher-dimensional space, allowing the use of static regret algorithms for dynamic settings. It establishes a trade-off between the penalties associated with loss variance and comparator sequence variability, proving that achieving regret bounds purely based on the squared path-length without additional penalties is infeasible. Additionally, the paper introduces a new measure of variability based on the locally-smoothed squared path-length, providing a more practical approach to managing dynamic regret.

**Strengths:**

- The paper establishes a novel equivalence between static and dynamic regret minimization. This theoretical contribution provides a unified framework for analyzing and designing algorithms for dynamic regret minimization, leveraging the well-established techniques for static regret.

- By establishing the trade-off between penalties due to the variance of the losses and the variability of the comparator sequence, the paper provides deep insights into the inherent limitations and trade-offs in dynamic regret minimization. This understanding is crucial for developing more effective algorithms and sets a foundation for future research in the area.

- The introduction of the locally-smoothed squared path-length reduces variance penalties, balances adaptability by smoothing out noise, and facilitates the decoupling of regret terms, leading to more practical and robust dynamic regret bounds.

**Weaknesses:**

- My major concern is the main titile could be somewhat overclaimed, as the proposed reduciton is restricted in OLO (or reduce OCO to OLO). Though I admit that analysis in non-convex problems (in bandit feedback) could be extremely hard, dynamic regret and static regret is still discussed in these problems. So may be ''DR is equal to SR in OCO'' may be more suitable.

- The approach involves embedding the comparator sequence into a higher-dimensional space, which could significantly increase computational complexity. Related concerns should be discussed, particularly in terms of implementing the proposed methods in large-scale or real-time applications.

- The paper does not provide intuitive examples or case studies to illustrate how to obtain dynamic regret in practical problems. I believe a simple example of existing problems (for example, non-stationary OCO) allows us to compare dynamic regret obtained by this reduction with existing studies, making it easier to understand.

- I'm not fully convinced by purely theoretical results. May be adding some simple evaluations to demonstrate the dynamic regret is indeed same as static regret by the proposed reduction can provide empirical evidences.

- The reasons for choosing this locally-smoothed square path-length is not clear. If I get it right, path-lengthes sharing similar local-smoothness would all fullfill the requirements. Meanwhile, this locally-smoothed square path-length is too detailed such that it may loose empirical intuitions or physical meanings to some extent.

**Questions:**

1. Does the high-dimensional embedding reduction also hold for adpative regret (to static regret)?
2. Can we directly design a unified algorithm (best-of-both-worlds) without knowing the types of regret based on this reduction?

**Limitations:**

\

---

> ### Author Rebuttal · Authors · 2024-08-07
>
> > My major concern is the main titile could be somewhat overclaimed, as
> > the proposed reduciton is restricted in OLO
>
> We actually believe that our title is carefully worded to avoid
> overclaiming: we do indeed present *an* equivalence between static and
> dynamic regret --- it is one which holds in a particular context, and
> our abstract and introduction state very clearly the scope of this work.
> Note also that, as discussed in [this comment](https://openreview.net/forum?id=hD8Et4uZ1o&noteId=CJxEDImAbf), the equivalence
> can be applied more generally for OCO.
>
> That said, we do not think this is a big issue either way: we can add
> "for Linear Losses" to the title if it solves the major concern of the
> reviewer.
>
> > The approach involves embedding the comparator sequence into a
> > higher-dimensional space, which could significantly increase
> > computational complexity. Related concerns should be discussed,
> > particularly in terms of implementing the proposed methods in
> > large-scale or real-time applications.
>
> Computational complexity is indeed one of the main considerations when
> choosing the dual norm pair $(\\|\cdot\\|,\\|\cdot\\|_{*})$ (i.e., choosing
> the matrix $M$ in the context of Theorem 2), and we do discuss it on
> page 4, line 136. Moreover, the computational complexity of the
> algorithm achieving the smoothed square path-length bound (characterized
> in Proposition 3) is discussed on page 8.
>
> We can add some additional discussion regarding the related
> implementation concerns. Note that these concerns are shared by all
> other dynamic regret algorithms achieving a non-trivial dependence on
> the comparator variability, as our algorithm can be implemented with the
> *same* computational $O(d\log T)$ per-round complexity as prior works,
> as discussed on page 8.
>
> > Does the high-dimensional embedding reduction also hold for adpative
> > regret (to static regret)?
>
> No, adaptive regret involves making guarantees over all sub-intervals
> simultaneously, while the proposed reduction explicitly considers the
> total loss over the entire interval $[1,T]$. These are quite separate
> notions of performance and there is no known general equivalence between
> them.
>
> > Can we directly design a unified algorithm (best-of-both-worlds)
> > without knowing the types of regret based on this reduction?
>
> If by types of regret you are referring to static vs. dynamic regret,
> the answer is *yes*: in fact, our guarantee in Theorem 2 captures static
> regret as a special case. If by types of regret your are referring to
> adapting to various different measures of variability (e.g., Theorem 2
> with different choices of $M$), *the answer is also yes*: see the last
> part of our [global response here](https://openreview.net/forum?id=hD8Et4uZ1o&noteId=2OutnXLWT0) for details.
>
> > The reasons for choosing this locally-smoothed square path-length is
> > not clear. If I get it right, path-lengthes sharing similar
> > local-smoothness would all fullfill the requirements. Meanwhile, this
> > locally-smoothed square path-length is too detailed such that it may
> > loose empirical intuitions or physical meanings to some extent.
>
> Due to space constraints, please see our [global response here](https://openreview.net/forum?id=hD8Et4uZ1o&noteId=2OutnXLWT0)
> for a detailed discussion of these concerns.

---

> > ### Comment · Reviewer_bfmX · 2024-08-13
> >
> > Thanks for your detailed explainations, all my concerns are correctly addressed.

---

### Official Review · Reviewer_yAJf · 2024-07-12

**Soundness:** 3
**Presentation:** 3
**Contribution:** 3
**Rating:** 8
**Confidence:** 3

**Summary:**

The paper addresses the problem of minimizing dynamic regret. The main goal is to provide a unified perspective on the problems of minimizing both dynamic and static regret. The first key contribution, presented in Proposition 1, is an interesting and straightforward observation demonstrating a general reduction from dynamic to static regret, showing the equivalence between these two notions.

In Theorem 1, the paper establishes a lower bound, indicating that squared path-length incurs a linear penalty, making adaptation to squared path-length impossible.

Furthermore, using a parameter-free algorithm in Algorithm 2, the paper achieves an upper bound that matches the previously derived lower bound for any weighted norm
$\|\cdot\|_{M}$.

In Section 4.1, the authors discuss the suitable choice for $M$. By setting $M=(HH^{\top}) ^{−1}$, where
$H$ is the unnormalized Haar basis matrix, they derive in Proposition 3 an upper bound for the regret that fully decouples comparators and gradient terms.

**Strengths:**

The message of the paper is clear, and the theoretical results are solid, consisting of a collection of small steps and observations that lead to important and surprising conclusions. The paper is extremely well-written and a pleasure to read.

**Weaknesses:**

I do not see any particular weaknesses in the paper.

Minor point: There are typos in lines 164 to 167. $V$ should be $V_T$ and $\epsilon$ should be $\epsilon_T$.

**Questions:**

I would like to know which measures of variability the authors expect to obtain with different choices of $M$ that yield reasonable and significant regret bounds.

**Limitations:**

The authors addressed the limitations of their work.

---

> ### Author Rebuttal · Authors · 2024-08-07
>
> Thank you for the positive review, we are glad you enjoyed reading our
> paper!
>
> > I would like to know which measures of variability the authors expect
> > to obtain with different choices of that yield reasonable and
> > significant regret bounds.
>
> So far the literature mostly revolves around the (unsquared) path-length
> as the measure of variability. One of the exciting things about this
> work is that it opens the door for many unexplored directions in terms
> of the variability measure, and provides a convenient framework for
> studying new notions of variability and their trade-offs. In this work,
> we focused on achieving results related to squared path-length, but it
> may be possible to achieve results scaling with more general distance
> metrics, for instance by leveraging the notion of group norms. The
> variance-variability coupling guarantees in Section 5 are also a very
> open and almost entirely unexplored direction that we would like to
> investigate further.

---

> > ### Comment · Reviewer_yAJf · 2024-08-08
> > **Rebuttal acknowledgment**
> >
> > I would like to thank the authors for the rebuttal. For now, I will maintain my current score. I plan to discuss the paper with the other reviewers and look forward to the author's discussions with them as well. I will update my score accordingly.

---

### Official Review · Reviewer_QTaB · 2024-07-12

**Soundness:** 2
**Presentation:** 2
**Contribution:** 3
**Rating:** 5
**Confidence:** 3

**Summary:**

This paper studied dynamic regret minimisation in convex optimisation. First, they proved some interesting equivalence between dynamic regret and static regret on extended decision space. Based on this observation, they showed a lower bound which implying that the hypothesis that optimal dynamic regret scales with square path length is impossible. The authors then provided an algorithm that incur regret scales with new notion of path length, which match the novel lower bound.

**Strengths:**

The main contribution of this paper is demonstrating the equivalence between dynamic regret and static regret in extended space, which is quite interesting. This allows the authors to present a new lower bound and prove a new upper bound for dynamic regret. Although I am not very familiar with the literature on dynamic regret, I believe the result is significant if it is correct, especially in comparison to previous results stated in this paper.

**Weaknesses:**

I found the writing of this paper quite difficult to follow, especially in the main sections (3 and 4). The comparison between the lower bound and upper bound could be better explained.

My question: The lower bound is $\tilde \Omega(\max_M G\sqrt{||\tilde{u}||_M \mathrm{Tr}(M) })$, and the upper bound essentially is $\tilde O(\min_M G\sqrt{||\tilde{u}||_M \mathrm{Tr}(M) })$. However, without strong duality on regret, there still seems to be a gap between the lower bound and upper bound. Correct me if I misunderstood, but I cannot find the reason why the lower bound and upper bound are tight as claimed in the paper.

**Questions:**

See weaknesses

**Limitations:**

Authors did discuss the limitation and potential future works.

---

> ### Author Rebuttal · Authors · 2024-08-07
>
> > My question: The lower bound is
> > $\tilde\Omega (\max\_{M}G\\|\tilde u\\|\_{M^{-1}}\sqrt{\mathrm{Tr}(M)})$,
> > and the upper bound essentially is
> > $\tilde O(\min\_{M}G\\|\tilde u\\|\_{M^{-1}}\sqrt{\mathrm{Tr}(M)})$.
>
> There seems to be a misunderstanding of the quantification of $M$ in the
> lower and upper bounds. The lower bound holds for *any* $M$ satisfying
> the conditions, not just for the worst-case $M$. Similarly our upper
> bound holds for any valid choice of $M$ rather than just the best one.
> So the upper and lower bounds do indeed match.
>
> To clarify the ideas, it might be useful to compare it, for example,
> with online mirror descent with $p$-norms: there exist upper and lower
> bounds that both hold for any $p$, not just with the best/worst ones.

---

> > ### Comment · Reviewer_QTaB · 2024-08-11
> >
> > Many thanks for your clarification.

---

### Official Review · Reviewer_Rnvn · 2024-07-13

**Soundness:** 3
**Presentation:** 2
**Contribution:** 2
**Rating:** 6
**Confidence:** 3

**Summary:**

This paper presents a reduction from the dynamic regret minimization for Online Convex Optimization (OCO) to a static regret minimization problem over an extended domain. Using this reduction, the authors establish a lower bound that highlights the trade-off between the variation of the comparator and the variance of the gradient norm. This general lower bound indicates that a squared path-length bound is impossible. Furthermore, the authors provide a general upper bound for dynamic regret based on techniques from parameter-free online learning. By specifying $M$, the paper demonstrates that a new type of squared dynamic regret bound is achievable.

**Strengths:**

- This paper provides a simple yet effective reduction from the dynamic regret minimization problem to the static regret minimization problem. Subsequently, the techniques developed for comparator-adaptive methods can be applied to the dynamic regret minimization problem.
- The paper offers a new analysis of the lower bound for dynamic regret minimization, revealing the intrinsic trade-off between the variability of the comparator sequence and the gradient norm.
- A new type of squared dynamic regret bound is introduced in this paper, based on this novel perspective.

**Weaknesses:**

- One of my main concerns about the paper is that the results appear to be somewhat overstated. The abstract claims that "we show that dynamic regret minimization is equivalent to static regret minimization in an extended decision space." This claim is somewhat misleading to me, as the proposed reduction only applies to the online linear optimization problem (or OCO with linearized surrogate loss). It is unclear if the proposed reduction holds for a broader range of dynamic regret minimization problems, such as those involving exp-concave or strongly convex loss functions. If not, it would be more appropriate to explicitly mention this limitation in the title and abstract and to provide a more detailed discussion on these limitations in the main paper.

- The high-level idea of this paper is quite similar to that of Zhang et al., 2023. Both approaches convert the dynamic regret minimization problem into a static regret minimization problem in another domain, using comparator-adaptive online learning methods to minimize extended static regret. While the authors discuss this in the related work section, it might be beneficial to explicitly state that previous work has already somewhat demonstrated equivalence between the dynamic regret minimization and static regret minimization problems.

-  As mentioned earlier, a similar reduction was presented in Section 2.2 of Zhang et al., 2023. It would be helpful if the authors provided a more detailed comparison of the two reductions. In the related work section, the paper states, "We take a similar but slightly more general approach." A more detailed explanation of how Proposition 1 generalizes previous work would be beneficial. Specifically, it would be useful to clarify whether the proposition can be derived by selecting a specific dictionary in Zhang et al., 2023. A more detailed discussion in Section 2 of this paper would be advantageous.

Overall, I find the paper interesting, but I am concerned about the overstated contributions and the insufficient comparison with previous work. I would be happy to raise my score if these issues are properly addressed.

Reference: Zhiyu Zhang, Ashok Cutkosky, and Ioannis Ch. Paschalidis. Unconstrained dynamic regret via sparse coding, 2023.

===post-rebuttal===
My primary concern was the significance of the proposed framework compared with Zhang et al. (2023) and the overstatement regarding the generality of the framework. After several discussions with the authors, I am convinced that the paper offers a more flexible and general framework for handling the dynamic regret minimization problem beyond the work of Zhang et al. (2023). To reflect this, I have updated my score to 6. Nevertheless, I still encourage the authors to clarify the limitations and provide a more detailed discussion on the contributions of the proposed framework in the revision.

**Questions:**

Please refer to the weakness part.

**Limitations:**

This paper is theoretical in nature, and I do not identify any potential negative societal impact.

---

> ### Author Rebuttal · Authors · 2024-08-07
>
> > The abstract claims that \"we show that dynamic regret minimization is
> > equivalent to static regret minimization in an extended decision
> > space.\" This claim is somewhat misleading to me, as the proposed
> > reduction only applies to the online linear optimization problem (or
> > OCO with linearized surrogate loss). It is unclear if the proposed
> > reduction holds for a broader range of dynamic regret minimization
> > problems, such as those involving exp-concave or strongly convex loss
> > functions.
>
> The proposed reduction can be applied to any convex losses, including
> strongly convex and exp-concave losses, by bounding
> $\sum_{t=1}^{T}\ell_{t}(w_{t})-\ell_{t}(u_{t})\le \sum_{t=1}^{T}\langle g_{t},w_{t}-u_{t}\rangle= R_{T}^{\text{Seq}}(\tilde u)$
> via convexity.
>
> We can also be more precise, obtaining an **equality for convex
> losses**, by observing that $$\begin{aligned}
>   \sum_{t=1}^{T}\ell_{t}(w_{t})-\ell_{t}(u_{t})=\sum_{t=1}^{T} \langle g_{t},w_{t}-u_{t}\rangle-D_{\ell_{t}}(u_{t}\|w_{t})\end{aligned}$$
> by definition of Bregman divergences. Since this is an equality, any
> possible improvements that one might hope to leverage in settings with
> curvature are reflected in the terms $-D_{\ell_{t}}(u_{t}\|w_{t})$, while
> our reduction can be applied to the first term,
> $\sum_{t=1}^{T}\langle{g_{t},w_{t}-u_{t}}\rangle=R_{T}^{\text{Seq}}(\tilde u)$.
> As is usually the case, one should design their algorithm in such a way
> that these curvature terms $-D_{\ell_{t}}(u_{t}\|w_{t})$ are properly
> leveraged in the end. For instance, for strongly-convex losses in the
> static regret setting, Algorithm 6 of Cutkosky & Orabona (2018) provides
> a reduction for OLO which makes a guarantee that implies logarithmic
> regret when the curvature terms satisfy
> $-D_{\ell_{t}}(u\|w_{t})\le -\frac{\alpha}{2}\\|u-w_{t}\\|^{2}$ (that is,
> when the losses are $\alpha$-strongly convex). A similar reduction could
> potentially be applied here and would be an interesting direction for
> future investigation.
>
> > A similar reduction was presented in Section 2.2 of Zhang et al.,
> > 2023. It would be helpful if the authors provided a more detailed
> > comparison of the two reductions. In the related work section, the
> > paper states, \"We take a similar but slightly more general
> > approach.\" A more detailed explanation of how Proposition 1
> > generalizes previous work would be beneficial. Specifically, it would
> > be useful to clarify whether the proposition can be derived by
> > selecting a specific dictionary in Zhang et al., 2023. A more detailed
> > discussion in Section 2 of this paper would be advantageous.
> >
> > \[\...\] it might be beneficial to explicitly state that previous work
> > has already somewhat demonstrated equivalence between the dynamic
> > regret minimization and static regret minimization problems.
>
> The key difference is that we provide a general equivalence between
> static and dynamic regret that *any strategy* can be plugged into, while
> Zhang et al. (2023) provide a reduction which prescribes a particular
> strategy which allows the user to leverage static regret minimizers for
> dynamic regret. That is, our Proposition 1 holds regardless of what
> strategy you end up choosing to set $w_{t}$, and simply shows that
> writing the dynamic regret in a different way leads to static regret in
> a different decision space. Because there is no restriction on how
> $w_{t}$ is set, our proposition actually captures the approach of Zhang
> et al. (2023) as a special case. Note that this also demonstrates that
> there is no choice of dictionary in the framework of Zhang et al. (2023)
> from which Proposition 1 can be derived --- Proposition 1 is strictly
> more general.
>
> In contrast, the approach proposed by Zhang et al. (2023) requires that
> the designer commit to choosing $w_{t}=\mathcal{H}\_{t}x\_{t}$ for some
> matrix $\mathcal{H}\_{t}$ and where the coordinates of $x_{t}$ are
> separate 1-dimensional learning algorithms. Because their approach
> prescribes a strategy that the user must design around, their approach
> *does not* imply an equivalence between dynamic and static regret, but
> rather provides a reduction that allows one to leverage a particular
> class of static regret algorithms to design dynamic regret algorithms.
> This potentially limits the types of guarantees that can be achieved
> using the approach; for instance, the algorithm achieving the smoothed
> path-length result in Section 4.1 can not be represented in their
> framework, since it does not use separate coordinate-wise 1-dimensional
> updates.

---

> > ### Comment · Reviewer_Rnvn · 2024-08-13
> >
> > Thank you for the detailed response. However, I remain concerned about the generality of the proposed reduction as claimed by the authors.
> >
> > Regarding the first point, it is still unclear to me how the proposed reduction can be applied to minimize the dynamic regret bound for exp-concave or strongly convex functions. While one could indeed introduce an additional Bregman divergence term, the subsequent analysis required to achieve improved dynamic regret for exp-concave functions (e.g. the rate in [1]) is far from straightforward to me. In this regard, the proposed reduction does not necessarily simplify the analysis for minimizing dynamic regret but rather shifts it to an analysis focused on the comparator-adaptive static regret bound, a perspective first introduced by Zhang et al., 2023.
> >
> > As for the comparison with Zhang et al., 2023, I appreciate the authors' feedback on the generality of the proposed reduction. However, the significance of this generality remains unclear. Both in the main paper and the feedback, the analysis of the squared path-length bound is presented as a prominent example demonstrating the advantage of the proposed framework. As other reviewers have pointed out, the advantage and significance of the proposed squared path-length are still not entirely evident.
> >
> > [1] Baby and Wang. Optimal Dynamic Regret in Exp-Concave Online Learning. COLT 2021.

---

> ### Author Response · Authors · 2024-08-13
>
> > it is still unclear to me how the proposed reduction can be applied to minimize the dynamic regret bound for exp-concave or strongly convex functions.
>
> We are genuinely puzzled by this comment and we kindly invite the reviewer to reconsider it.
> We showed above that our equivalence *can* be extended to other losses as well *just because the reviewer wondered ``if the proposed reduction holds for a broader range of dynamic regret minimization problems''*, but this was never a claim in our paper!
> It seems very unfair to be penalized for something that is completely orthogonal to the contribution of our paper. It should be very clear that we focus on linear losses and linearized convex losses, but we are happy to emphasize it more in the text, to avoid any misunderstanding.
> That said, focusing on linear losses is actually very common in this literature, even in the recent paper by Zhang et al. (2023), which also applies to linear losses and has no straightforward way to achieve the optimal bound for exp-concave losses.
>
>
>
> > Both in the main paper and the feedback, the analysis of the squared path-length bound is presented as a prominent example demonstrating the advantage of the proposed framework.
>
> We respectfully but firmly disagree with this mischaracterization of our results: as we just agreed with Reviewer YDMs, the averaged squared path length is just an example, not the main contribution. Instead, as we wrote above, our main contribution is about *a framework that allows to reason on the trade-offs of measures of dynamic regret, both in terms of upper and lower bounds.* We also agreed with Reviewer YDMs that our text did not clearly convey this message: We apologize for it and we will improve it. Clearly, none of these trade-offs are present in the beautiful work of Zhang et al. (2023), that instead focuses on orthogonal aspects. This is evident from the fact that their framework does not enable one to prove lower bounds.

---

> > ### Comment · Reviewer_Rnvn · 2024-08-13
> >
> > I agree that applying the work to exp-concave losses and strongly convex functions should not be a critical point, but it does reveal that the paper shares the same limitations as Zhang et al. (2023), rather than being a truly "general framework." Some aspects of the presentation are confusing to me. For instance, the abstract claims, "we show that dynamic regret minimization is equivalent to static regret minimization in an extended decision space." without any statement on the condition. Similarly, the response states, "The key difference is that we provide a *general equivalence* between static and dynamic regret that *any strategy* can be plugged into." (I believe it is still necessary to use comparator-adaptive algorithms to achieve meaningful dynamic regret.) Furthermore, line 30 asserts, "Note that OCO problems can *always* be reduced to OLO via the well-known inequality," and lines 121-123 claim, "Proposition 1 is a regret equivalence—we *lose nothing* by taking this perspective, yet it allows us to immediately apply *all the usual techniques* and approaches from the static regret setting."
> >
> > In my view, a more concise contribution would be to present a general framework to reduce minimizing dynamic regret to obtain comparator-adaptive static regret bound for OLO case. I strongly recommend that the authors clarify this point, particularly in the abstract, introduction, and reduction sections.

---

> > > ### Author Response · Authors · 2024-08-13
> > >
> > > First of all, we thank the reviewer for reconsidering their criticism about exp-concave losses and strongly convex functions, we appreciate the intellectual honesty.
> > >
> > > > the paper shares the same limitations as Zhang et al. (2023), rather than being a truly "general framework"
> > >
> > > We might be nitpicking on the definition of ``general framework''. Does our framework recover any optimal rate for any kind of convex loss used in the dynamic regret literature? Clearly not, but it was never our aim. However, I think we can agree that we are strictly more general than Zhang et al. (2023), for the reasons we listed above, and this generality is non-trivial (for instance, it allows us to prove lower bounds that the framework of Zhang et al (2023) could not). Also, our paper and Zhang et al. (2023) focus on different aspects of the dynamic regret problem, so we do not really see why these two approaches ought to be considered as being in conflict; they both make unique and novel contributions to the literature.
> > >
> > >
> > > > Some aspects of the presentation are confusing to me.
> > >
> > > While in our eyes our scope and limitations were clear, it is possible we might overlooked this aspect and we appreciate the feedback to make the message of our paper sharper. As already discussed with Reviewer YDMs and bfmX, we are very happy to change the text to emphasize that we focus on linear and linearized losses in all the relevant parts of the paper, and clearly specify that we do not explicitly leverage the curvature of the losses, as well as specify that our main contribution is about a way to analyze trade-offs in measures of dynamic regret. Future work might explore the aspect of curvature, possibly following the approach we sketched above.
> > >
> > > > I believe it is still necessary to use comparator-adaptive algorithms to achieve meaningful dynamic regret.
> > >
> > > This is not necessarily the case; the use of comparator-adaptive algorithms is just a convenient way to achieve adaptivity to an unknown comparator norm (and thus to the variability of the comparator sequence via our reduction), but it is not necessarily the only way to achieve meaningful adaptivity to the comparator sequence. For instance, the Metagrad algorithm of van Erven \& Koolen (2016) achieves a guarantee which adapts to an unknown comparator in the static bounded domain setting but it is even stronger than the usual comparator-adaptive algorithms which are leveraged by this work and by Zhang et al. (2023). It may be possible to apply such an algorithm in our framework to get new results which would not otherwise be possible using the framework of Zhang et al (2023). Of course, this would entail additional computational considerations, but solving all these details is beyond the scope of this rebuttal and would be an interesting direction for future work.

---

> ### Comment · Reviewer_Rnvn · 2024-08-13
>
> I think there may be some misunderstanding from the authors regarding my previous response. My intention was never to criticise the paper for not achieving improved dynamic regret for exp-concave functions. Rather, it is taken as an example to illustrate that some parts of the paper are overstated, as evidenced by the claim I referenced earlier. Nevertheless, I appreciate the authors' efforts in making the revisions.
>
> > The use of comparator-adaptive algorithms is just a convenient way to achieve adaptivity to an unknown comparator norm
>
> I understand that one can use other methods to achieve adaptivity to the comparator sequence, but it is clear that not just any strategy can be applied. Achieving meaningful dynamic results still requires specific techniques to minimize the converted static regret. While the range of algorithms that can be used might extend beyond those in Zhang et al., 2023, the earlier statement, "The key difference is that we provide a general equivalence between static and dynamic regret that any strategy can be plugged into," along with the claim, "yet it allows us to immediately apply all the usual techniques and approaches from the static regret setting," is overstated to me.

---

> > ### Author Response · Authors · 2024-08-14
> >
> > Thank you again for taking the time to honestly engage in the discussion with us. We understand your perspective and will incorporate all the insights gained from our discussion here to improve the clarity of the main text.
> >
> > > I understand that one can use other methods to achieve adaptivity to the comparator sequence, but it is clear that not just any strategy can be applied. Achieving meaningful dynamic results still requires specific techniques to minimize the converted static regret.
> >
> > As a brief final note, would like to point out that the two referenced statements referenced do not contradict. While it is true that only certain choices will lead to meaningful guarantees, this is true of *any* general framework and is not unique to ours. For instance, we hope we can agree that FTRL and mirror descent can be considered ``general frameworks'' for static regret minimization, yet not all sequences of convex regularizers lead to meaningful guarantees. This does not make these frameworks less valuable, and indeed many of the interesting questions in online learning revolve around *which choices* of regularizer lead to meaningful guarantees under which assumptions. This is analogous to the choice of strategy to be applied in our framework.

---

### Official Review · Reviewer_28Fn · 2024-07-15

**Soundness:** 3
**Presentation:** 3
**Contribution:** 3
**Rating:** 7
**Confidence:** 3

**Summary:**

The main contribution of the paper is showing that, by `lifting' an online linear optimization (OLO) problem to a higher dimensional space, the dynamic regret in the original setting is equal to the $\textit{static}$ regret achieved in the higher dimensional setting. While simple, this clean reduction allows easier and more elegant reasoning about how to develop and analyze dynamic regret algorithms since static regret algorithms and analyses are so well studied and understood.

They then employ this reduction to prove the existence of a trade-off between comparator variability penalty in terms of a $M$ matrix norm and $G^2 Tr(M^{-1})$, where $G$ is the Lipschitz parameter of the losses. By choosing $M$ in such a way that the comparator variability term becomes the squared path-length, they are able to show that adapting a dynamic regret rate to the squared path-length implies a supra-linear (i.e., vacuous) guarantee.

They complement the above general trade-off by providing an algorithm that is able to match the lower bounds along the frontier implied by different specifications of $M$. They instantiate this with a particular matrix setting and attain a corresponding positive result which says it is possible to instead adapt to a modified definition of squared path-length (more specifically, a locally smoothed counterpart).

**Strengths:**

While simple, their reduction is quite elegant and insightful. This conceptual contribution is also nicely complemented by the authors employing it to achieve a new lower-bound (that can be matched via existing methods). The presentation is clear and intuitive.

**Weaknesses:**

Some weaknesses of the paper are that the focus on matrix norms in Section 3 may be a bit limited in scope and that the statement of Theorem 1 is quite convoluted. More broadly, having additional implications of the equivalence would be desirable (e.g., having a more thorough and completed version of Section 5 and/or a more comprehensive list of observations in a similar vein).

**Questions:**

- Do you believe there are any implications of your reduction that are particularly useful from a practical/deployment perspective, or do you see the contribution as being a strictly conceptual one?
- Do you have any additional intuition for the locally-smoothed path-length quantity? For example, how smoothed would one generally expect this quantity to be?

**Limitations:**

The authors have addressed limitations of their work.

---

> ### Author Rebuttal · Authors · 2024-08-07
>
> > Do you believe there are any implications of your reduction that are
> > particularly useful from a practical/deployment perspective, or do you
> > see the contribution as being a strictly conceptual one?
>
> We believe that the most important implication is that the reduction
> allows us to use any present and, more importantly, future result from
> static online learning, to solve dynamic regret problems. For example,
> the algorithm that we show in Section 4.1 has the same computational
> complexity as previous ones (such as Zhang et al. (2018), Jacobsen &
> Cutkosky (2022), and Zhang et al. (2023), to name a few) so it has the
> same applicability as these works.
>
> Practically speaking, in many real-world problems the "solution" to the
> problem (represented, e.g., by the ideal model parameters) can change
> over time, due to unpredictable changes in environment or underlying
> data-generating process. These changes to the solution are captured by
> the time-varying comparator in dynamic regret. A major benefit of these
> comparator-adaptive dynamic regret guarantees is that the algorithm
> *automatically adapts* to this changing comparator sequence, without
> requiring *any* prior knowledge whatsoever about this sequence or how it
> is changing. We believe this is an exceptionally powerful property which
> is indeed useful from a practical/deployment perspective.
>
> > Do you have any additional intuition for the locally-smoothed
> > path-length quantity? For example, how smoothed would one generally
> > expect this quantity to be?
>
> Due to space constraints, please see our [global response here](https://openreview.net/forum?id=hD8Et4uZ1o&noteId=2OutnXLWT0) for a
> detailed discussion of these concerns.

---

> > ### Comment · Reviewer_28Fn · 2024-08-13
> > **Acknowledgement of response**
> >
> > Thank you for clarifying the practical implications! I maintain my positive score (7).

---

### Official Review · Reviewer_YDMs · 2024-07-16

**Soundness:** 3
**Presentation:** 3
**Contribution:** 2
**Rating:** 4
**Confidence:** 4

**Summary:**

This paper investigates the problem of dynamic regret minimization in unbounded domains. The authors propose a novel lossless reduction from dynamic regret minimization to static regret minimization by treating the changing comparators in dynamic regret as a fixed one in another decision space with higher dimensions. The key in the reduced static regret minimization is to choose an appropriate matrix norm which is used to transform the static regret bound in lifted domains into a meaningful dynamic regret in the original decision space. Denoting by $M$ the norm-related matrix, this work proposes a novel lower bound regarding the variability penalty $\\|\tilde{u}\\|_M$ and a variance penalty $G^2 \text{Tr} (M^{-1})$. Consequently, the authors proved that a dynamic regret depending on the squared path length is impossible. By contrast, by choosing a specific matrix norm defined by Haar basis matrices, it suffices to achieve a dynamic regret bound with the squared path length defined on some averaged comparators.

**Strengths:**

The dynamic-to-static regret is pretty impressive. Although it is a little conceptual and less practical, it provides us with a novel way for dynamic regret minimization.

The lower bound in Theorem 1 is also meaningful by depicting the influences of the variability penalty $\\|\tilde{u}\\|_M$ and the variance penalty $G^2 \text{Tr} (M^{-1})$, providing guidance for consequent algorithm design.

**Weaknesses:**

I have some major questions for the results regarding the squared path length and I hope that the authors can answer the questions to solve my concerns.

1. I believe that the lower bound in Theorem 1 is correct, which lower-bounds by the dynamic regret with $\\|\tilde{u}\\|_M$ and $G^2 \text{Tr} (M^{-1})$. However, I am doubtful about Proposition 2 when translating this lower bound to a dynamic regret measured by the squared path length. Proposition 2 only shows that choosing a specific $M$ does not suffice to achieve a valid squared-path-length dynamic regret. However, it does not rule out the possibility of other choices of $M$. Or, is there some one-to-one mapping between the choice of $M$ and the squared path length? In my point of view, discussions of this issue are missing and I hope that the authors could revise the paper according to it in the revised version.
2. As for upper bounds, the obtained bound (in Section 4) is less favorable than I originally expected. The squared path length is defined on some averaged comparators, which are averaged on the intervals induced by the Harr OLR algorithm. The relationship between the current averaged squared length and the desired squared length is not discussed. On the top of Page 8, the authors show that their result can recover the previous dynamic regret defined with $P_T$ (non-squared path length). However, on Page 2, the authors stated that $P_T$ based dynamic regret is not favorable enough with an overly pessimistic diameter quantity. As a result, implying $P_T$ based dynamic regret is not convincing enough, at least for me, to validate the significance of the obtained result.
3. The obtained regret also suffers from additional $\log T$ terms, which is not favorable enough in the full information setup. Do previous works with similar setups also suffer from the same issue?
4. The dynamic-to-static reduction is novel. However, when designing algorithms, the computational complexity is not discussed at all, which is pretty important because the dimension is related to time horizon $T$ in the reduced problem. I suggest that the authors could add discussions about computational complexity in the revised version.

Overall, my main concern is that despite the novelty of the dynamic-to-static reduction, to what extent this reduction is significant is not fully discussed. I believe that this reduction is of interest to the community. However, the currently obtained results do not suffice to validate its significance.

**Questions:**

A minor type maybe: In Line 24, a missing inverse operator in the matrix $S$? (i.e., $[H_n H_n^\top]^{-1}$ but not $H_n H_n^\top$?)

**Limitations:**

The authors have adequately addressed the limitations and, if applicable, potential negative societal impact of their work.

---

> ### Author Rebuttal · Authors · 2024-08-07
>
> > The authors stated that $P_T$ based dynamic regret is not favorable enough with
> > an overly pessimistic diameter quantity.
> >
> > The relationship between the current averaged squared length and the
> > desired squared length is not discussed
>
> Given that the smoothed version of squared path-length is a new notion
> of variability, it is natural to show that it is always at least as good
> as the standard notion of variability. However, it should be clear this
> is only a sanity check and this notion can often be much better since it
> does not incur the pessimistic diameter penalty.
> Please see [our global response](https://openreview.net/forum?id=hD8Et4uZ1o&noteId=2OutnXLWT0) for a detailed
> discussion.
>
> > The obtained regret also suffers from additional terms, which is not
> > favorable enough in the full information setup. Do previous works?
>
> All prior works which achieve dynamic regret guarantees in unbounded
> domains do indeed incur additional logarithmic penalties, even in the
> full information setting. See Jacobsen & Cutkosky (2022,2023) and Zhang et al. (2023).
> Similar to the static regret
> setting, such logarithmic penalties appear to be the price paid for
> achieving adaptivity to an arbitrary comparator sequence. Such penalties
> are known to be **necessary** in online learning with unbounded domains,
> see the lower bound in Orabona (2013), and recently proved in the
> stochastic setting as well by Carmon & Hinder (2024).
> Works which assume a bounded domain, such as Zhang et al (2018) can
> avoid this additional term because they are leveraging prior knowledge
> about the comparator sequence, ie, that $\\|u_{t}-w_{1}\\|\le D$ for all
> t. In this work, we focus on achieving novel guarantees in the
> unbounded setting, though our reduction in Proposition 1 is applicable
> in the bounded setting as well.
>
> > the computational complexity of the reduction is not discussed
>
> The computational complexity is related to the choice of $M$, and is one
> of the main considerations that one must consider when choosing $M$.
> This is discussed on page 4, lines 136-138. In fact, one of the key
> properties of the $M$ we choose in Proposition 3 is that it supports
> sparse updates, requiring $O(d\log T)$ per-round computation to update
> the algorithm, matching the computational complexity enjoyed by
> prior works in this setting. This is discussed on page 8, starting on
> line 243.
>
> > proposition 2 only shows that choosing a specific M does not suffice
>
> This is a very good point, thanks for raising it. The choice of $M$
> uniquely exposes the squared path length up to a constant offset term,
> but it is easy to see that the offset term does not make any difference
> for our claim. To see why, consider the 1-dimensional setting and note
> that for any positive definite $M$ there is a unique $\Sigma$ such that
> $M=\Sigma^{\top}\Sigma$. Hence, without loss of generality we can focus
> on $\Sigma$ satisfying
> $$
>   \\|\tilde u\\|\_{M}^{2}=\langle\tilde u,M\tilde u\rangle=\langle \Sigma\tilde u, \Sigma\tilde u\rangle = \langle v,\tilde u\rangle^{2} + \sum_{t=2}^{T}\\|u_{t}-u_{t-1}\\|^{2},
> $$
> for some $v\in\mathbb{R}^{T}$ and such that $M=\Sigma^{\top}\Sigma$ is
> positive definite. Note that the constant offset term is unavoidable: it
> is what captures the static regret guarantee in the case where
> $u_{1}=\ldots=u_{T}=u$. Proposition 2 considers $v=(0,\ldots,0,1)$
> to
> get
> $\\|\tilde u\\|^{2}\_{M}=\\|u_{T}\\|^{2}+\sum\_{t=2}^{T}\\|u\_{t}-u\_{t-1}\\|^{2}$,
> though below we will show that any vector $v$ would still lead to
> $Tr(M^{-1})=\Omega(T^{2})$.
>
> It is clear that the only way to construct expressions of the form above
> is via matrices $\Sigma$ satisfying
> $$\Sigma\tilde u=c\begin{pmatrix}u\_{1}-u\_{2}\\\\ u\_{2}-u\_{3}\\\\\vdots\\\\u\_{T-1}-u\_{T}\\\\ \langle v,\tilde u\rangle\end{pmatrix},$$
> where $c\in\{-1,1\}$ and the order of the rows indices of the vector can
> be permuted without loss of generality. In particular, the only matrices
> that can produce these expressions (again noting that the rows can be
> permuted without loss of generality) are of the form
> $$
> \Sigma = c\begin{pmatrix}1&-1&0&0&\dots&0&0\\\\
>             0&1&-1&0&\dots&0&0\\\\
>             0&0&1&-1&\dots&0&0\\\\
>             \vdots& & &\ddots&&&\\\\
>             0&0&0&0&\dots&1&-1\\\\
>             \hline\\\\
>             v\_{1}&v\_{2}&v\_{3}&v\_{4}&\ldots&v\_{T-1}&v\_{T}
>           \end{pmatrix}
>   =:
>   c\begin{pmatrix}
>     \Delta\\\\
>     \hline\\\\
>     v^{\top}
>   \end{pmatrix},
> $$
> so $M=\Delta^\top\Delta + v v^\top$.
> Adding a rank-1 update to $\Delta^{\top}\Delta$ will increase its
> eigenvalues $\lambda_{i}=\lambda_i(\Delta^{\top}\Delta)$ by some
> $a_{i}\ge 0$, such that $\sum_{i}a_{i}=\\|v\\|^2$. Moreover, there is a
> unique zero eigenvalue of $\Delta^{\top}\Delta$, corresponding to the
> eigenvector $(1,\ldots,1)/T$. Therefore,
> $Tr(M^{-1})=1/a_{1}+\sum_{i>1}\frac{1}{\lambda_{i}+a_{i}}$. This implies
> that the worst-case choice of $a_{1},\dots,a_{T}$ is given by a convex
> constrained optimization problem which yields $a_{i}=0$ for $i>1$ and
> $a_{1}=\\|v\\|^2$ via the KKT conditions. Hence, the worst-case choice of
> $v$ is the one which increases only the eigenvalue corresponding to the
> uniform eigenvector, so the worst case choice of $v$ is proportional to
> the uniform eigenvector and
> $Tr(M^{-1})\ge \Omega(\frac{1}{\\|v\\|^2}+\sum_{i>1}\frac{1}{\lambda_{i}})\ge \Omega(\sum_{i>1}\frac{1}{\lambda_{i}(M_{0})})$,
> where $M_{0}$ is the matrix from Proposition 2, obtained by setting
> $v=(0,\dots,0,1)$. Notably, from the proof of Proposition 2 it can be
> seen that even if we dropped the largest eigenvalue of $M_{0}^{-1}$, we
> would still have $Tr(M^{-1}_0)\ge \Omega(T^{2})$. Hence, any of the
> matrices which produce the squared path-length dependence must incur the
> same $\Omega(T^{2})$ variance penalty up to constant factors. We will
> expand this reasoning into a more formal proof in the paper and we can
> provide additional details if needed, but the above claims can be easily
> verified numerically.

---

> > ### Comment · Reviewer_YDMs · 2024-08-12
> > **Comments to Rebuttal**
> >
> > Thanks to the authors for the detailed reply. I still have some follow-up questions about the authors' rebuttal, and hope that the authors could resolve my concerns.
> >
> > 1. I have read the global response about the motivation of the averaged squared path length. The authors said, "It turns out that in the worst case, this is only a $\log T$ factor more than the square path length at the worst scale; see first inequality in line 239". Could the authors provide some further explanations or derivations on this statement? Does it mean that the averaged path length can be upper-bounded by the desired squared path length only by a $\log T$ factor overhead?
> >
> > 2. I do not agree with the first motivation for the squared path length, which shows that it is never worse (up to polylog factors) than the usual path length $P_T^{\\|\cdot\\|}$. When transforming the averaged path length into $P_T^{\\|\cdot\\|}$, there is an additional term of $\bar{D}$ (see the equation in line 239). However, as the authors have said in the introduction part, this diameter quantity is overly pessimistic and thus cannot be used to validate the significance of the obtained results.
> >
> > 3. Being able to aggregate multiple algorithms, as stated at the end of the global response, is an advantage. However, given that we cannot identify a good enough choice of $M$ here (please refer to the next point below), how good the aggregated guarantee will be cannot be explicitly illuminated. I acknowledge that this is an advantage of the obtained result. However, this advantage seems to not be able to bring some explicit improvements compared with the current bound.
> >
> > 4. For my first weakness in the original review, the authors responded that "Hence, without loss of generality we can focus on
> > $\Sigma$ satisfying...". I do not understand why this choice of $M$ is "without loss of generality" since this choice seems to rule out a large number of possible Ms. Could the authors provide more explanations on this point?

---

> > > ### Author Response · Authors · 2024-08-12
> > >
> > > Thanks for engaging with us, we are happy to discuss these points
> > > further.
> > >
> > > However, before the specific points, we would like to stress once again the averaged squared path length is only an example of the generality of the framework and not our main contribution. Focusing too much on it might hide the fact that the main message of our paper is *a framework that allows to reason on the trade-offs of measures of dynamic regret, both in terms of upper and lower bounds*. As an illustrative analogy, the seminal work of Nemirovski & Yuudin (1983) introduces the mirror descent framework and its general guarantee, and then proceeds to show specific examples such as $p$-norms. The example is not the main contribution; the framework and the existence of a trade-off are the key contributions upon which a large literature of work has been developed by the community.
> > >
> > > >\"this is only a $\log$ factor more than the square path length at the worst scale" [...] Could the authors provide some further explanations or derivations on this statement?
> > >
> > > From its definition, the average squared path length is the sum of the squared path length at *all possible time scales*. Given that there are $O(\log T)$ time scales (line 230), in the *worst case* the averaged path length can be at most $O(\log T)$ times the path length at the worst scale (i.e., $\sum_{\tau=1}^{N}\bar{P}(\tau,\vec{u})\le N\max_{\tau}\bar{P}(\tau,\vec{u})$ where $\bar{P}(\tau,\vec{u})$ denotes the averaged square path-length at timescale $\tau$), as detailed in the global response. Note that the standard squared path length corresponds to scale $\tau=1$.
> > >
> > > >I do not agree with the first motivation for the squared path length, which shows that it is never worse (up to polylog factors) than the usual path length
> > >
> > > We are not sure how this could be controversial: any new measure should be at least as good (up to polylog factors) as the measure used in prior works. However, this is not a proof of superiority, as the reviewer seems to imply; this is a sanity check showing that *in the worst case*, the new variability penalty is at least as good as the one in prior works. However, there are cases where it can be significantly better: in our global response, we provide an example in which the averaged square path-length is significantly smaller precisely thanks to the diameter: the averaged squared path length is equal to the squared path length and it is $T (a-b)^2$, while the $M P_T^{\\|\cdot\\|}$ dependence, where $M$ is the diameter term, is $O(T |a-b| \max(b,a)),$ which can be arbitrarily worse.
> > >
> > > >Being able to aggregate multiple algorithms...
> > >
> > > Our results show there are trade-offs and the best choice of $M$ will be data-dependent, so trying to quantify gains is hopeless in general. Instead, we can easily aggregate algorithms and compete with the best matrix among a number of different ones. Is it important? We very much believe so! Once again, we would like to point out the similarity with $p$-norms: Theorem 2 in Cutkosky (2019) used the same idea to have a static regret with respect to all $p$-norm: no explicit gain is showed as well, but it should be clear the importance of his result. We show how to do this for *different measures of dynamic regret* and we are not aware of anything similar in this literature.
> > >
> > > >I do not understand why this choice of is \"without loss of generality\" since this choice seems to rule out a large number of possible Ms. Could the authors provide more explanations on this point?
> > >
> > > We are not completely sure we understand your question.
> > >
> > > In case you are wondering how it is possible to decompose $M$ into a product of a matrix and its transpose, this is always true because $M$ is symmetric and positive definite. Note that only the matrix $M$ matters and not its representation through $\Sigma$.
> > >
> > > In case you are wondering why we restrict our attention only to the matrices $M$ that result in the term $\langle v, \tilde u\rangle^{2} + \sum_{t=2}^{T}\\|u_{t}-u_{t-1}\\|^{2}$ and not to some other expression. Indeed, we proved that adding another terms to the above two terms can kill the $\Omega(T^2)$ factor: this is exactly the case of the averaged squared path length that we propose. However, any such addition would result in a measure that it is not the squared path length anymore. Hence, our claim is correct: The squared path length cannot be achieved without incurring the stated penalty.
> > >
> > > Finally, in case you are wondering if it is possible to find two different matrices that result in $\langle v, \tilde u\rangle^{2} + \sum_{t=2}^{T}\\|u_{t}-u_{t-1}\\|^{2}$, this is possible only up to permutations of the rows of $\Sigma$, as we explained in the global response.
> > >
> > > Please let us know if you meant something else.

---

> > > > ### Comment · Reviewer_YDMs · 2024-08-13
> > > > **Thanks for the feedback**
> > > >
> > > > Thanks for the detailed feedback. I have the following comments and questions.
> > > >
> > > > 1. **About the main contribution**: The authors said that the results of the (averaged) squared path length are not the main contribution of this work. However, if this is true, from my point of view, the introduction part should be reorganized to emphasize more about this kind of trade-off. The current version of the introduction part starts from the standard path length, shows that it is less favorable in unbounded domains, and introduces the desired quantity of squared path length. This would make readers certainly feel that this work aims to achieve some kind of dynamic regret in terms of the squared path length, as also summarized as the second contribution of this work on Page 2. I suggest that the authors could revise the paper to make the main contribution that you wanna claim clearer.
> > > >
> > > > 2. **About the relationship of the averaged path length with the standard squared one**: The authors' explanations have solved part of my concerns. From your response, I understand that Line 239 actually provides $\sum_{\tau=1}^N \bar{P}(\tau, u) \le N \max_{\tau} \bar{P}(\tau, u)$. However, there is a $\max$ operator on the RHS. While I understand that $\bar{P}(\tau=1, u)$ is similar to the desired path length, the $\max$ operator makes it hard to provide a clear relationship between the averaged path length and the squared one. Besides, when $\tau=1$, $\bar{P}(\tau=1, u)$ looks like $\\|u_1 - u_2\\|^2 + \\|u_3 - u_4\\|^2 + \cdots$. This quantity seems to be better than the squared path length by omitting terms like $\\|u_2 - u_3\\|^2$, $\\|u_4 - u_5\\|^2$, etc? Could the authors provide some further explanations about the relationship between $\bar{P}(\tau=1, u)$ and the standard squared path length? To conclude, in this comment, I understand that there is no direct relationship between the averaged path length and the standard squared one due to the $\max$ operator. Am I right?
> > > >
> > > > 3. **About being able to aggregate multiple algorithms**: Thanks for the reply. I agree with the authors now.
> > > >
> > > > 4. **About the without-loss-of-generality choice of $M$**: Sorry for the confusion. I was referring to the second point you said. I am still confused about whether this choice of $M$ is ***the only*** way to get the squared path length. If not, you cannot prove that it is impossible to achieve the squared path length dynamic regret. Some explanations of the authors are mainly heuristic but without rigorous proofs. It means that it is still possible to achieve the squared path length by choosing another unknown $M$. The authors still cannot rule out this kind of possibility, do you?
> > > >
> > > > At last, thanks again for the very detailed responses.

---

> > > > > ### Author Response · Authors · 2024-08-13
> > > > >
> > > > > Thanks for your comments and further questions, this discussion is very useful for us.
> > > > >
> > > > > > About the main contribution
> > > > >
> > > > > The reviewer is very right and we will definitely follow their suggestion.
> > > > >
> > > > > > About the relationship of the averaged path length with the standard squared one
> > > > >
> > > > > Yes, this is a good catch: you are correct that $\bar P(1,\vec{u})$ is actually
> > > > > slightly better than the squared path-length on its own. Our apologies for the
> > > > > confusion regarding this detail.
> > > > > This does not significantly change our discussion so far; with this correction
> > > > > in mind, our example
> > > > > shows a case where our measure is actually better than the squared path-length. Moreover, we still have the property that the averaged square path-length is
> > > > > proportional to $T(b-a)^{2}$ while the usual $MP_{T}^{\\|\cdot\\|}$ dependence is proportional to $T\\|b-a\\|\max(b,a)$, and hence can also be arbitrarily better
> > > > > than the dependencies appearing in prior works.
> > > > >
> > > > > We agree with the reviewer: the $\max$ over time-scales does make it difficult to precisely
> > > > > quantify the difference to the regular squared path-length.
> > > > > However, this result is the first squared-path-length-like result *of any kind* which has been
> > > > > achieved for arbitrary comparator sequences.
> > > > > As stated earlier, we believe these are reasonable compromises to make while still achieving a result which captures a similar essence to the ideal, but impossible to achieve, result.
> > > > >
> > > > >
> > > > >
> > > > > > About the without-loss-of-generality choice of $M$
> > > > >
> > > > > Suppose that there are two matrices $M_1$ and $M_2$ such that for any $\tilde u$ we have
> > > > > $$
> > > > > \\|\tilde u\\|\_{M_1}^{2} = \langle v,\tilde u\rangle^{2} + \sum_{t=2}^{T}\\|u_{t}-u_{t-1}\\|^{2},
> > > > > $$
> > > > > and
> > > > > $$
> > > > > \\|\tilde u\\|\_{M_2}^{2} = \langle v,\tilde u\rangle^{2} + \sum_{t=2}^{T}\\|u_{t}-u_{t-1}\\|^{2}~.
> > > > > $$
> > > > > Given the definition of the weighted norm, this implies that $\tilde{u}^\top (M_1-M_2) \tilde{u}=0$ for any $\tilde{u}$. Given that this holds for any $\tilde{u}$, this implies that $M_1-M_2$ is the zero matrix, that is, $M_1=M_2$ (see, e.g., https://math.stackexchange.com/questions/3314271/when-does-a-quadratic-form-being-equal-to-zero-implies-the-underlying-matrix-is)

---

> > > > > > ### Comment · Reviewer_YDMs · 2024-08-13
> > > > > > **Thanks for the response**
> > > > > >
> > > > > > Thanks for the timely feedback. I am happy that the authors and I have reached a certain consensus. I have only one remaining question about the choice of $M$. I do not understand why $M$ has to be in the specific form of  $\\|\tilde{u}\\|_{M}^{2} = \langle v,  \tilde{u}\rangle^2 + \sum\_{t=2}^T \\|u_t - u\_{t-1}\\|^2$? Are there other possibility of $M$ such that the squared path length can still be derived?

---

> > > > > > > ### Author Response · Authors · 2024-08-13
> > > > > > >
> > > > > > > We can agree that the squared path-length must appear in the expression,
> > > > > > > so what remains is why the leading term $\langle v,\tilde u\rangle^2$
> > > > > > > must appear. In the case of static regret, wherein $u_1=\dots=u_T=u$,
> > > > > > > there must be a term which reduces to $\\|u\\|^2$, since otherwise the
> > > > > > > guarantee would violate the existing $\tilde\Omega(\\|u\\|\sqrt{T})$ lower
> > > > > > > bounds for static regret. For any $v\ne 0$, the term
> > > > > > > $\langle v,\tilde u\rangle^2$ is a strict generalization of the
> > > > > > > $\\|u\\|^2$ term that must appear in the static case, but also captures
> > > > > > > any possible combination of the $u_t$'s (such as $\bar{u}=1/T\sum_tu_t$)
> > > > > > > which could appear using a weighted norm.
> > > > > > >
> > > > > > > One can surely consider other choices of matrices that result in more
> > > > > > > terms *in addition* to the two terms above. One might wonder if only
> > > > > > > non-positive terms can be added, but it is easy to see that this would
> > > > > > > make the $\Omega(T^2)$ penalty even bigger (because the matrix $Q$
> > > > > > > corresponding to these terms would be negative semi-definite and the
> > > > > > > eigenvalues of $M+Q$ would be smaller than the one of $M$ by Weyl's
> > > > > > > inequality), so it is not a possibility. Thus, one can only add terms
> > > > > > > that are non-negative or whose sign depends on the $u_i$. But, as we
> > > > > > > wrote above, this would result in something that is not the squared path
> > > > > > > length anymore! For example, the average squared path norm almost
> > > > > > > contains the squared path norm, and it would be easy to make it even
> > > > > > > bigger so that is exactly contains it, but it is clearly not the squared
> > > > > > > path norm anymore.

---

### Author Rebuttal · Authors · 2024-08-07

We thank the reviewers for their thoughtful comments and taking the time to carefully read our paper. Below, we address
some of their common questions.

A common point contention in the reviews was the significance of the
average squared path-length dependence guarantee achieved in Proposition
3. We provide additional discussion about these concerns below, but we
would first like to stress that **the power of our approach lies in its
generality**. Indeed, our lower bound shows more generally that *there
exists a fundamental trade-off between the comparator variability
penalty and the gradient variance penalty*, so there is no one measure
of variability that is strictly better across all problem instances.
This is similar to the trade-off existing in terms of dual norms in
online mirror descent. We are not aware of any other paper on dynamic
regret pointing out these fundamental trade-offs, and this result is due
to our simple equivalence with static regret. Our general result in
*Theorem 2 provides a framework for achieving any of these trade-offs*,
simply by changing the choice of weighting matrix $M$. In Proposition 3,
we provided as an example one instance of the more general result
because we believe it is a reasonable compromise to the square
path-length, but we stress that our result is much more general than
this one example. Achieving more favorable trade-offs via cleverly
crafted weighting matrices $M$ is an important direction for future
work. Moreover, as we show below, given multiple different choices of
$M$ we can always easily ensure the best of their guarantees.

**Motivation of averaged squared length/intuition.** It is important to
understand that we proved that it is impossible to adapt to the squared
path-length $P_{T}^{\\|\cdot\\|^{2}}$ without making the regret guarantee
vacuous. More fundamentally, the lower bound says that this is the wrong
quantity to consider, so we must look for some achievable compromise.
The average squared path-length is one such reasonable compromise for
the following reasons:

1\) It is never be worse (up to polylog factors) than the usual
path-length $P_{T}^{\\|\cdot\\|}$;

2\) The algorithm which achieves adaptivity to the averaged square
path-length can be implemented in the same $O(d\log T )$ per-round
complexity enjoyed by prior works;

3\) It shares a similar physical meaning to the squared path length.

It should be clear why points 1 and 2 are important, so let's now focus
on point 3. The averaged squared path length is nothing else than the
sum of the squared path length at different time scales, see line 232.
In this view, the averaged squared path length includes the squared path
length, but it seems to inflate it just enough to avoid the vacuous
result given by our lower bound (i.e., Proposition 2). Now, one might
wonder by how much this is inflated, and it turns out that in the worst
case this is only a $\log T$ factor more than the square path length at
the worst scale, see first inequality in line 239. So, ignoring the log
factor, we are essentially substituting the squared path length at time
scale 1, with the maximum over $\tau$ of the squared path length at time
scale $\tau$. Thus, the original squared path length measures the drift
only at high frequency, while the averaged path length measures the
drift in all frequencies, like a more holistic measure of drift.
Moreover, this measure can be very close (or even equal!) to the squared
path-length in some instances. Indeed, consider a comparator sequence
which alternates between $a$ and $b$: on each of the timescales
$\tau=2^{i}$, the averages on each of the intervals of length $\tau$
will be $\bar u_{i}=(a+b)/2$. Hence the average square path-length at
timescale $\tau$ is $0$ for any $\tau$ except for $\tau=1$ so the two
measures are *exactly* equivalent here. However, it is impossible
compare favorably with the square path-length across *all* possible
problem instances due to the fundamental trade-off discussed above.
Nevertheless, we show below that one can easily combine the guarantees
achieved by different choices of $M$.

Finally, we would also like to point out that Proposition 3 is the
*first square path-length bound of any kind* that holds for arbitrary
comparator sequences, so we believe the result is of interest to the
community despite the fairly reasonable compromises discussed above.

**Best-of-Both Worlds Guarantees.** As discussed above, our lower bound
demonstrates a trade-off such that no $M$ can lead to uniformly better
guarantees across all problem instances. Luckily, given any two
instances of the algorithm in Theorem 2, call them $A$ and $B$, with
difference choices of the matrix $M$, we can always achieve the better
of their two guarantees using the standard iterate-adding argument of
Cutkosky (2018). The idea is simple: observe that for any choice of $M$,
Theorem 2 provides an algorithm guaranteeing
$R_{T}(0,\ldots,0)\le O(\epsilon)$. Hence, letting $w_{t}^{A}$ and
$w_{t}^{B}$ denote the outputs of algorithms $A$ and $B$ on round $t$,
suppose we play $w_{t}=w_{t}^{A}+w_{t}^{B}$. Then our dynamic regret is
$$\begin{aligned}
  R_{T}(\vec{u})= \sum_{t=1}^{T}\langle g_{t},w_{t}^{A}+w_{t}^{B}-u_{t}\rangle = R_{T}^{A}(\vec{u})+R_{T}^{B}(0,\ldots,0)\le O(R_{T}^{A}(\vec{u})+\epsilon),\end{aligned}$$
and similarly, we have
$R_{T}(\vec{u})\le O(\epsilon + R_{T}^{B}(\vec{u}))$. Since both of
these hold, overall we have
$R_{T}(\vec{u})\le O(\epsilon + \min\{R_{T}^{A}(\vec{u}),R_{T}^{B}(\vec{u})\})$.
The argument immediately extends to combining the guarantees of $N$
algorithms at the cost of only an $O(\epsilon N)$ penalty. Moreover, the
parameter $\epsilon$ in Theorem 2 appears only in $O(\epsilon)$ and
$\log(1/\epsilon)$ terms, so we can safely scale it by $1/N$ so that
combining $N$ algorithms requires only an $\log N$ penalty.

---

### Decision · Program_Chairs · 2024-09-25

**Decision:**

Accept (poster)

**Comment:**

This paper studies online convex optimization and investigates dynamic regret minimization by linking this measure with classic static regret. The primary motivation is to consider the unbounded online optimization setup and replace the classic path length with the squared one, which can be much smaller and more favored. The authors show that this objective seems challenging by building the reduction from dynamic regret to static regret over the extended domain. In fact, the authors provide a lower-bound argument to demonstrate the impossibility of achieving this. Instead, they propose novel algorithms to achieve dynamic regret scaling with a locally smooth squared path length, which is a new contribution to the community.

During the review stage, the paper received mixed evaluations. After extensive internal discussion among the reviewers as well as my personal check of the results, I feel those results are interesting and novel. However, as argued by several reviewers, there are indeed several critical issues that need addressing and clarifying before publication:

(1) Potential overstatement of generality: While reviewers agree that the work provides a more general reduction from dynamic regret to comparator-adaptive static bounds compared to Zhang et al. (2023), some paper claims may be overstated. For instance, the title states "an equivalence between dynamic regret and static regret," but the abstract (also elsewhere) suggests that "dynamic regret minimization is equivalent to static regret minimization in an extended decision space," which sounds like the authors have established a universal reduction from dynamic regret to static regret. Evidently, this is inaccurate as the paper does not consider other important OCO setups like exp-concave, strongly convex cases, or bandits settings. The authors should carefully revise these statements throughout the paper to prevent any misleading interpretations. Additionally, as suggested by two reviewers, the paper's title should be more rigorous -- it should reflect that the reduction targets OLO (PS: even for OLO, if more refined problem-dependent regret bounds are considered, I'm not sure if this reduction continues to perform effectively).

(2) Technical soundness of impossible of general squared path length bound: this is an interesting result, as the instance-dependent lower bound is always valuable in dynamic regret research. However, several technical aspects remain unclear, as one reviewer raised. The authors should carefully review and revise these arguments to make them more precise.

Despite these issues, the paper presents valuable and interesting results worthy of knowing to the community. Although the conference venue does not have a mandatory revision system, the authors have committed to addressing the concerns in their rebuttal document. I believe the authors will incorporate these improvements in an updated version and make the paper more accurate. Therefore, I recommend accepting this paper! The authors are requested to revise the paper by incorporating the reviewers' suggestions.